# Analytic Characterization of the Hessian in Shallow ReLU Models: A Tale of Symmetry

**Yossi Arjevani**
NYU
yossi.arjevani@gmail.com

**Michael Field**
UCSB
mikefield@gmail.com

## Abstract

We consider the optimization problem associated with fitting two-layers ReLU networks with respect to the squared loss, where labels are generated by a target network. We leverage the rich symmetry structure to analytically characterize the Hessian at various families of spurious minima in the natural regime where the number of inputs $d$ and the number of hidden neurons $k$ is finite. In particular, we prove that for $d \geq k$ standard Gaussian inputs: (a) of the $dk$ eigenvalues of the Hessian, $dk - O(d)$ concentrate near zero, (b) $\Omega(d)$ of the eigenvalues grow linearly with $k$. Although this phenomenon of extremely skewed spectrum has been observed many times before, to our knowledge, this is the first time it has been established rigorously. Our analytic approach uses techniques, new to the field, from symmetry breaking and representation theory, and carries important implications for our ability to argue about statistical generalization through local curvature.

## 1 Introduction

Much of the current effort in understanding the empirical success of artificial neural networks is concerned with the geometry of the associated nonconvex optimization landscapes. Of particular importance is the Hessian spectrum which characterizes the local curvature of the loss at different points in the space. This, in turn, allows one to closely examine the dynamics of stochastic first order methods [1, 2], design potentially better optimization methods [3, 4], and argue about various challenging aspects of the network generalization capabilities [5, 6, 7]. Unfortunately, the excessively high cost involved in an exact computation of the Hessian spectrum renders this task prohibitive already for moderate-sized problems.

Existing approaches for addressing this computational barrier use numerical methods for approximating the Hessian spectrum [2, 8], study the limiting spectral density of shallow models w.r.t. *randomly* drawn weights [9, 10, 11], or employ various simplified indirect curvature metrics [12, 5, 7, 13, 14]. Notably, none of these techniques is able to yield an analytic characterization of the Hessian at critical points in high-dimensional spaces.

In this paper, we develop a novel approach for studying the Hessian in a class of student-teacher (ST) models. Concretely, we focus on the squared loss of fitting the ReLU network $\boldsymbol{x} \mapsto \mathbf{1}_k^\top \phi(\boldsymbol{W}\boldsymbol{x})$,

$$\mathcal{L}(\boldsymbol{W}) \doteq \frac{1}{2}\mathbb{E}_{\boldsymbol{x} \sim \mathcal{N}(\mathbf{0}, I_d)}\big[(\mathbf{1}_k^\top \phi(\boldsymbol{W}\boldsymbol{x}) - \mathbf{1}_k^\top \phi(\boldsymbol{V}\boldsymbol{x}))^2\big], \quad \boldsymbol{W} \in M(k, d), \tag{1.1}$$

where $\phi(z) \doteq \max\{0, z\}$ is the ReLU activation acting coordinate-wise, $\mathbf{1}_k$ is the $k$-dimensional vector of all ones, $M(k, d)$ denotes the space of all $k \times d$ matrices, and $\boldsymbol{V} \in M(k, d)$ denotes the weight matrix of the target network. The ST framework offers a clean venue for analyzing optimization- and generalization-related aspects of neural network models, and has consequently enjoyed a surge of interest in recent years, e.g., [15, 16, 17, 18, 19, 20, 21, 22], to name a few.

Perhaps surprisingly, already for this simple model, the rich and perplexing geometry of the induced nonconvex optimization landscape seems to be out of reach of existing analytic methods.

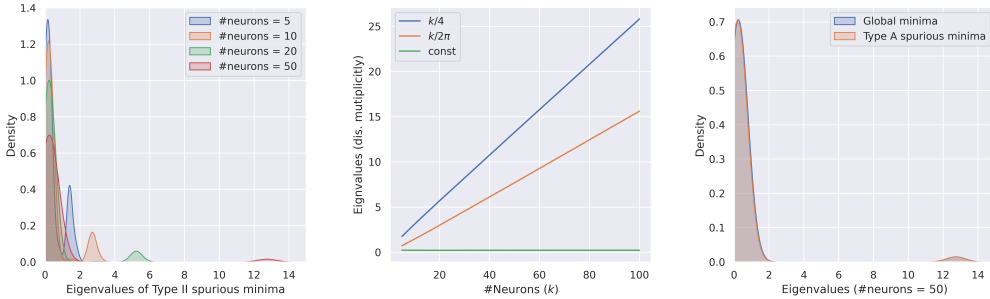

Figure 1: (Left) in congruence with Theorem 2, $1 - \Theta(1/k)$ fraction of the spectral density at type II spurious minima concentrates around $1/4 \pm 1/2\pi$ as the number of neurons $k$ grows simultaneously with the number of inputs. The remaining $\Theta(1/k)$ fraction consists of outliers. (Middle) examining the spectrum of type II minima (disregarding multiplicity) as $k$ grows confirms the existence of $k + 1$ outlier eigenvalues, of which $k$ grow at a rate of $k/4$ and one at a rate of $k/2\pi$. (Right) the spectra of global minima and type A spurious minima are almost indistinguishable already for $k = 50$, thus challenging the flat minima conjecture.

The starting point of our approach is the following simple observation: for any permutation matrices $\boldsymbol{P} \in M(k, k)$, $\boldsymbol{Q} \in M(d, d)$, it holds that $\mathcal{L}(\boldsymbol{P}\boldsymbol{W}\boldsymbol{Q}^\top) = \mathcal{L}(\boldsymbol{W})$, for all $\boldsymbol{W} \in M(k, d)$ [23, Section 4.1]. It is natural to ask how the critical points of $\mathcal{L}$ reflect this symmetry. This question was answered in [23] where it was shown that critical points detected by stochastic gradient descent (SGD) remain unchanged under transformations of the form $\boldsymbol{W} \mapsto \boldsymbol{P}\boldsymbol{W}\boldsymbol{Q}^\top$ for large groups of pairs of permutation matrices $(\boldsymbol{P}, \boldsymbol{Q})$. Using these invariance properties, families of critical points of $\mathcal{L}$ were expressed as power series in $1/\sqrt{k}$ leading to, for example, a precise formula for the decay rate of $\mathcal{L}$ [24]. Building on this, we show in this paper how the rich symmetry structure can be used to derive an analytic description of the Hessian spectral density of $\mathcal{L}$, for arbitrarily large, yet finite, values of $k$. Having this access to precise high-dimensional spectral densities, we revisit a number of hypotheses in the machine learning literature pertaining to curvature, optimization and generalization, and establish or refute them *rigorously* for the first time.

The paper is organized as follows. In Section 2 we state our main results and provide discussions aimed at interpreting our findings in the light of existing literature. Section 3 and Section 4 are devoted to describing our representation theory-based approach; all proofs are deferred to the appendix. Lastly, detailed empirical corroborations of our analysis are given in Section F.

## 2 Main results and related work

A formal discussion of our main results requires some familiarity with group and representation theory. Here, we provide a high-level description of our contributions, and defer more detailed statements to later sections after the relevant notions have been introduced.

**Symmetry-based analysis framework.** Utilizing the rich symmetry exhibited by neural network models, we develop a novel framework for analytically characterizing the second-order information of shallow ST ReLU models. In its general form, our main result can be stated as follows.

**Theorem 1** (Informal). *Assuming a $k \times k$ orthogonal target matrix $\boldsymbol{V}$ ($d = k$), the spectrum of local minima of $\mathcal{L}$ consists of a fixed number of distinct eigenvalues (ranging between 6 and 22 for high symmetry minima)—independent of the number of neurons $k$. Moreover, the spectral distribution is massively concentrated in a small number of eigenvalues (ranging between 2 and 4 for high symmetry minima) which accounts for $k^2 - O(k)$ of the spectrum. Similar results hold if $d > k$.*

The theorem is a consequence of the unique *isotypic* decomposition of the Hessian that derives from the invariance properties of $\mathcal{L}$ (see Theorem 4). Using stability arguments, it follows that upon

convergence, the spectral density is expected to accumulate in clusters whose number does not depend on $k$. This is confirmed by empirical results which we provide in section F.1.

Next, we instantiate our framework to the global minima and three families of spurious local minima introduced in [24], referred to as types A, I and II (type II corresponds to the spurious minima described for $6 \leq k \leq 20$ in [22]). A complete description of the minima is provided in Lemma 5 (type II) and in Section D.2 (type A and I).

**Theorem 2.** *Assuming a $k \times k$ orthogonal target matrix $\boldsymbol{V}$, and $k \geq 6$,*

1. $\nabla^2 \mathcal{L}$ *at $\boldsymbol{W} = \boldsymbol{V}$ has 6 distinct strictly positive eigenvalues:*

   (a) $\frac{1}{4} - \frac{1}{2\pi}$ *of multiplicity $\frac{k(k-1)}{2}$.*

   (b) $\frac{1}{4} + \frac{1}{2\pi}$ *of multiplicity $\frac{k(k-3)}{2}$.*

   (c) $\frac{k+1}{4} + O(k^{-1})$ *and $\frac{1}{4} + O(k^{-1})$ of multiplicity $k - 1$.*

   (d) $\approx -0.3471 + \frac{k}{2\pi} + O(k^{-1})$ *and $\approx 0.8471 + \frac{k}{4} + O(k^{-1})$ of multiplicity one.*

   (e) *The objective value is $0$.*

2. $\nabla^2 \mathcal{L}$ *at type A spurious local minima has 7 distinct strictly positive eigenvalues:*

   (a) $\frac{1}{4} - \frac{1}{2\pi} - \frac{1}{\pi\sqrt{k}} + O(k^{-1})$ *of multiplicity $\frac{(k-1)(k-2)}{2}$.*

   (b) $\frac{1}{4} + \frac{1}{2\pi} - \frac{1}{\pi\sqrt{k}} + O(k^{-1})$ *of multiplicity $\frac{k(k-3)}{2}$.*

   (c) *3 eigenvalues, $\frac{k+1}{4} + O(k^{-1/2})$, $\frac{1}{4} + O(k^{-1/2})$ and $\frac{1}{4} - \frac{1}{2\pi} + O(k^{-1/2})$ of multiplicity $k - 1$.*

   (d) *2 eigenvalues: $c_1 + \frac{k}{4} + O(k^{-1/2})$ and $c_2 + \frac{k}{2\pi} + O(k^{-1/2})$ of multiplicity one, $c_1, c_2 > 0$.*

   (e) *The objective value is $(\frac{1}{2} - \frac{1}{\pi}) + O(k^{-1/2})$ [24].*

3. $\nabla^2 \mathcal{L}$ *at type II spurious local minima has 12 distinct strictly positive eigenvalues:*

   (a) $\frac{1}{4} - \frac{1}{2\pi} - \frac{1}{\pi k} + O(k^{-3/2})$ *of multiplicity $\frac{(k-2)(k-3)}{2}$.*

   (b) $\frac{1}{4} + \frac{1}{2\pi} - \frac{1}{\pi k} + O(k^{-3/2})$ *of multiplicity $\frac{(k-1)(k-4)}{2}$.*

   (c) *5 Eigenvalues of multiplicity $k - 2$, of which one grows at a rate of $\frac{k+1}{4} + O(k^{-1})$, and the rest converge to small constants.*

   (d) *5 Eigenvalues of multiplicity 1, of which 2 grow at a rate of $c_3 + \frac{k}{4} + O(k^{-1})$, one grows at a rate of $c_4 + \frac{k}{2\pi} + O(k^{-1})$, $c_3, c_4 > 0$, and the rest converge to small constants.*

   (e) *The objective value is $(\frac{1}{2} - \frac{2}{\pi^2})k^{-1} + O(k^{-3/2})$ [24].*

   *If $d > k$, there will be 2 (resp. 3) additional strictly positive eigenvalues for type A (resp. I or II) minima with total multiplicity $(d - k)k$. The full description, together with that for type I eigenvalues, is given in Section A.*

We note that methods for establishing the existence of spurious local minima for $\mathcal{L}$ are computer-aided and applicable only for small-scale problems [22]. Our method establishes the existence of spurious local minimum analytically and for arbitrarily large $k$ and $d$ (assuming $k \leq d$). An additional consequence of Theorem 2 is that *not all local minima are alike*. Below, we discuss the implications of the similarities and the differences between families of minima of $\mathcal{L}$.

**Positively-skewed Hessian spectral density.** Although first reported nearly 30 years ago [25], to the best of our knowledge, this is the first time that this phenomenon of extremely skewed spectral density has been established *rigorously* for high-dimensional problems (see Figure 1). Early empirical studies of the Hessian spectrum [25] revealed that local minima tend to be extremely ill-conditioned. This intriguing observation was corroborated and further refined in a series of works [26, 27, 28] which studied how the spectrum evolves along the training process. It was noticed that, upon convergence, the spectral density decomposes into two parts: a bulk of eigenvalues concentrated around zero, and a small set of positive outliers located away from zero.

Due to the high computational cost of an exact computation of the Hessian spectrum ($O(k^3 d^3)$ for a $k \times d$ weight matrix), this phenomenon of extremely skewed spectral densities has only been

confirmed for small-scale networks. Other methods for extracting second-order information in large-scale problems roughly fall into two general categories. The first class of methods approximate the Hessian spectral density by employing various numerical estimation techniques, most notably stochastic Lanczos method (e.g., [2, 8]). These methods have provided various numerical evidences that indicate that a similar skewed spectrum phenomenon also occurs in full-scale modern neural networks. The second class of techniques builds on tools from random matrix theory. This approach yields an exact computation of the limiting spectral distribution (i.e., the number of neurons is taken to infinity), assuming the inputs, as well as the model weights are drawn at random [9, 10, 11]. In contrast, our method gives an exact description of the spectral density for essentially any (finite) number of neurons, and at critical points rather than randomly drawn weight matrices.

**The flat minima conjecture and implicit bias.** It has long been debated whether some notion of local curvature can be used to explain the remarkable generalization capabilities of modern neural networks [3, 5, 6, 29, 30, 4, 7]. One intriguing hypothesis suggests that minima with wider basins of attraction tend to generalize better. An intuitive possible explanation is that flat minima promote statistical and numerical stability; together with low empirical loss, these ingredients are widely-used to achieve good generalization, cf. [31].

Perhaps surprisingly, our analysis shows that the spectra of global minima and the spurious minima considered in Theorem 2 agree on $k^2 - O(k)$ out of $k^2$ eigenvalues to within $O(k^{-1/2})$-accuracy ($d = k$). Thus, only the remaining $O(k)$ can potentially account for any essential difference in the local curvature. However, for type A spurious minima, even the remaining $O(k)$ eigenvalues are $O(k^{-1/2})$-far from the spectrum of the global minima. Consequently, in our settings, local second-order curvature *cannot* be used to separate global minima from spurious minima, thus ruling out notions of 'flatness' which rely exclusively on the Hessian spectrum. Of course, other metrics of a 'wideness of basins' may well apply.

Despite being a striking counter-example for a spectral-based notion of flatness, we note that, empirically, under Xavier initialization [32], type A spurious minima are rarely detected by SGD [24]. This stands in sharp contrast to type II minima to which SGD converges with a substantial empirical probability. Thus, for reasons which are yet to be understood, the bias induced by Xavier initialization seems to favor the class of global and type II minima at which the objective value decays with $k$ to zero, rather than type A and type I minima whose objective value converges to strictly positive constants, cf., [33, 34]. We leave further study of this phenomenon, as well as other families of spurious minima, to future work.

**Proof technique.** Conceptually, the derivation of the eigenvalue estimate in Theorem 2 is based on ideas originating in symmetry-breaking, equivariant bifurcation theory and representation theory. Group invariance properties of the loss function (1.1) imply that the Hessian at symmetric points (under a proper notion of symmetry) must exhibit a certain block structure, and this makes possible an explicit computation of the Hessian spectrum. Empirically, and somewhat miraculously, spurious minima of (1.1) tend to be highly symmetric. As a consequence, their Hessian can be simplified using the same symmetry-based methods. The reminder of the paper is devoted to a formal and more detailed exposition of this approach.

## 3 The method: a symmetry-based analysis of the Hessian

In order to avoid a long preliminaries section, key ideas and concepts are introduced and organized so as to illuminate our strategy for analyzing the Hessian. We illustrate with reference to the case of global minima where $d = k$ and the target weight matrix $\boldsymbol{V}$ is the identity $\boldsymbol{I}_k$.

### 3.1 Studying invariance properties via group action

We first review background material on group actions and fix notations (see [35, Chapters 1, 2] for a more complete account). Elementary concepts from group theory are assumed known. We start with two examples that are used later.

**Examples 1.** (1) The *symmetric group* $S_d$, $d \in \mathbb{N}$, is the group of permutations of $[d] \doteq \{1, \ldots, d\}$. (2) Let $\mathrm{GL}(d, \mathbb{R})$ denote the space of invertible linear maps on $\mathbb{R}^d$. Under composition, $\mathrm{GL}(d, \mathbb{R})$ has the structure of a group. The *orthogonal group* $\mathrm{O}(d)$ is the subgroup of $\mathrm{GL}(d, \mathbb{R})$ defined by

$O(d) = \{A \in GL(d, \mathbb{R}) \mid \|Ax\| = \|x\|, \text{ for all } x \in \mathbb{R}^d\}$. Both $GL(d, \mathbb{R})$ and $O(d)$ can be viewed as groups of invertible $d \times d$ matrices.

Characteristically, these groups consist of *transformations* of a set and so we are led to the notion of a *G-space* $X$ where we have an *action* of a group $G$ on a set $X$. Formally, this is a group homomorphism from $G$ to the group of bijections of $X$. For example, $S_d$ naturally acts on $[d]$ as permutations and both $GL(d, \mathbb{R})$ and $O(d)$ act on $\mathbb{R}^d$ as linear transformations (or matrix multiplication).

An example, which we use extensively in studying the invariance properties of $\mathcal{L}$, is given by the action of the group $S_k \times S_d \subset S_{k \times d}$, $k, d \in \mathbb{N}$, on $[k] \times [d]$ defined by

$$(\pi, \rho)(i, j) = (\pi^{-1}(i), \rho^{-1}(j)), \ \pi \in S_k, \rho \in S_d, \ (i, j) \in [k] \times [d]. \tag{3.2}$$

This action induces an action on the space $M(k, d)$ of $k \times d$-matrices $A = [A_{ij}]$ by $(\pi, \rho)[A_{ij}] = [A_{\pi^{-1}(i), \rho^{-1}(j)}]$. The action can be defined in terms of permutation matrices but is easier to describe in terms of rows and columns: $(\pi, \rho)A$ permutes rows (resp. columns) of $A$ according to $\pi$ (resp. $\rho$). As mentioned in the introduction, for our choice of $\boldsymbol{V} = \boldsymbol{I}_k$, $\mathcal{L}$ is $S_k \times S_d$-invariant. If $d = k$, define the *diagonal subgroup* $\Delta S_k$ of $S_k \times S_k$ by $\Delta S_k = \{(g, g) \mid g \in S_k\}$. Note that $\Delta S_k \approx S_k$. When we restrict the $S_k \times S_k$-action on $M(k, k)$ to $\Delta S_k$, we refer to the diagonal $S_k$-action, or just the $S_k$-action on $M(k, k)$. This action of $S_k$ on $M(k, k)$ maps diagonal matrices to diagonal matrices and should not be confused with the actions of $S_k$ on $M(k, k)$ defined by either permuting rows or columns.

**Example 2.** Take $p, q \in \mathbb{N}$, $p + q = k$, and consider the diagonal action of $S_p \times S_q \subset S_k$ on $M(k, k)$. Write $A \in M(k, k)$ in block matrix form as $A = \begin{bmatrix} A_{p,p} & A_{p,q} \\ A_{q,p} & A_{q,q} \end{bmatrix}$. If $(g, h) \in S_p \times S_q \subset S_k$, then $(g, h)A = \begin{bmatrix} gA_{p,p} & (g, h)A_{p,q} \\ (h, g)A_{q,p} & hA_{q,q} \end{bmatrix}$ where $gA_{p,p}$ (resp. $hA_{q,q}$) are defined via the diagonal action of $S_p$ (resp. $S_q$) on $A_{p,p}$ (resp. $A_{q,q}$), and $(g, h)A_{p,q}$ and $(h, g)A_{q,p}$ are defined through the natural action of $S_p \times S_q$ on rows and columns. Thus, for $(g, h)A_{p,q}$ (resp. $(h, g)A_{q,p}$) we permute rows (resp. columns) according to $g$ and columns (resp. rows) according to $h$. In the case when $p = k - 1$, $q = 1$, $S_{k-1}$ will act diagonally on $A_{k-1,k-1}$, fix $a_{kk}$, and act by permuting the first $(k - 1)$ entries of the last row and column.

Given $\boldsymbol{W} \in M(k, k)$, the largest subgroup of $S_k \times S_k$ fixing $\boldsymbol{W}$ is called the *isotropy* subgroup of $\boldsymbol{W}$ and is used as means of measuring the symmetry of $\boldsymbol{W}$. The isotropy subgroup of $\boldsymbol{V} \in M(k, k)$ is the diagonal subgroup $\Delta S_k$. Our focus will be on critical points $\boldsymbol{W}$ whose isotropy groups are subgroups of the target matrix $\boldsymbol{V} = \boldsymbol{I}_k$, that is, $\Delta S_k$ and $\Delta S_{k-1}$ (see Figure 2—we use the notation $\Delta S_k$ as the isotropy is a *subgroup* of $S_k \times S_k$). Other choices of target matrices yield different symmetry-breaking of the isotropy of the global minima (see [23] for more details). In the next section, we show how the symmetry of local minima greatly simplifies the analysis of their Hessian.

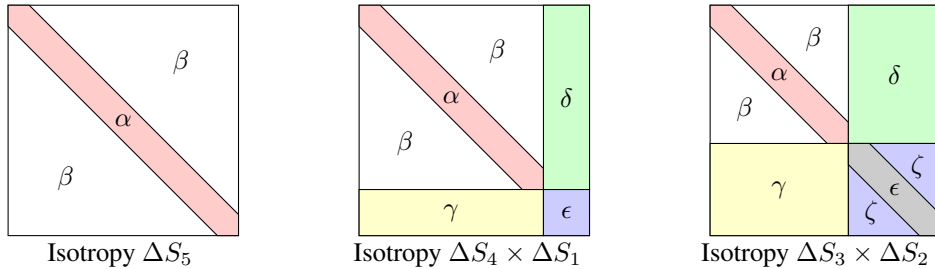

Isotropy $\Delta S_5$     Isotropy $\Delta S_4 \times \Delta S_1$     Isotropy $\Delta S_3 \times \Delta S_2$

Figure 2: A schematic description of $5 \times 5$ matrices with isotropy $\Delta S_5, \Delta S_4 \times \Delta S_1$ and $\Delta S_3 \times S_2$, from left to right (borrowed from [23]). $\alpha, \beta, \gamma, \delta, \epsilon$ and $\zeta$ are assumed to be 'sufficiently' different.

## 3.2 The spectrum of equivariant linear isomorphisms

If $G$ is a subgroup of $O(d)$, the action on $\mathbb{R}^d$ is called an *orthogonal* representation of $G$ (we often drop the qualifier orthogonal). Denote by $(\mathbb{R}^d, G)$ as necessary. The *degree* of a representation $(V, G)$ is the dimension of $V$ ($V$ will always be a linear subspace of some $\mathbb{R}^n$ with the induced Euclidean inner product). The action of $S_k \times S_d \subset S_{k \times d}$ on $M(k, d)$ is orthogonal with respect to the standard

Euclidean inner product on $M(k,d) \approx \mathbb{R}^{k \times d}$ since the action permutes the coordinates of $\mathbb{R}^{k \times d}$ (equivalently, components of $k \times d$ matrices).

Given two representations $(V, G)$ and $(W, G)$, a map $A : V \to W$ is called $G$-equivariant if $A(gv) = gA(v)$, for all $g \in G, v \in V$. If $A$ is linear and equivariant, we say $A$ is a $G$-*map*. Invariant functions naturally provide examples of equivariant maps. Thus the gradient $\nabla \mathcal{F}$ is a $S_k \times S_d$-equivariant self map of $M(k,d)$ and if $\boldsymbol{W}$ is a critical point of $\nabla \mathcal{F}$ with isotropy $G \subset S_k \times S_d$, then $\nabla^2 \mathcal{F}(\boldsymbol{W}) : M(k,d) \to M(k,d)$ is a $G$-map (see [35, 24]). The equivariance of the Hessian is the key ingredient that allows us to study the spectral density at *symmetric* local minima.

A representation $(\mathbb{R}^n, G)$ is *irreducible* if the only linear subspaces of $\mathbb{R}^n$ that are preserved (invariant) by the $G$-action are $\mathbb{R}^n$ and $\{0\}$. Two orthogonal representations $(V, G)$, $(W, G)$ are *isomorphic* (and have the same *isomorphism class*) if there exists a $G$-map $A : V \to W$ which is a linear isomorphism. If $(V, G)$, $(W, G)$ are irreducible but not isomorphic then every $G$-map $A : V \to W$ is zero (as the kernel and the image of a $G$-map are $G$-invariant). If $(V, G)$ is irreducible, then the space $\mathrm{Hom}_G(V, V)$ of $G$-maps (endomorphisms) of $V$ is a real associative division algebra and is isomorphic by a theorem of Frobenius to either $\mathbb{R}, \mathbb{C}$ or $\mathbb{H}$ (the quaternions). The *only* case that will concern us here is when $\mathrm{Hom}_G(V, V) \approx \mathbb{R}$ when we say the representation is *real*.

**Example 3.** Let $n > 1$. Take the natural (orthogonal) action of $S_n$ on $\mathbb{R}^n$ defined by permuting coordinates. The representation is not irreducible since the subspace $T = \{(x, x, \cdots, x) \in \mathbb{R}^n \mid x \in \mathbb{R}\}$ is invariant by the action of $S_n$, as is the hyperplane $H_{n-1} = T^\perp = \{(x_1, \cdots, x_n) \mid \sum_{i \in [n]} x_i = 0\}$. It is easy to check that $(T, S_n)$, also called the *trivial* representation of $S_n$, and $(H_{n-1}, S_n)$, the *standard* representation, are irreducible, real, and not isomorphic.

Every representation $(\mathbb{R}^n, G)$ can be written uniquely, up to order, as an orthogonal direct sum $\oplus_{i \in [m]} V_i$, where each $(V_i, G)$ is an orthogonal direct sum of isomorphic irreducible representations $(V_{ij}, G)$, $j \in [p_i]$, and $(V_{ij}, G)$ is isomorphic to $(V_{i'j'}, G)$ if and only if $i' = i$. The subspaces $V_{ij}$ are *not* uniquely determined if $p_i > 1$. If there are $m$ distinct isomorphism classes $\mathfrak{v}_1, \cdots, \mathfrak{v}_m$ of irreducible representations, then $(\mathbb{R}^n, G)$ may be represented by the sum $p_1 \mathfrak{v}_1 + \cdots + p_m \mathfrak{v}_m$, where $p_i \geq 1$ counts the number of representations with isomorphism class $\mathfrak{v}_i$. Up to order, this sum (that is, the $\mathfrak{v}_i$ and their multiplicities) is uniquely determined by $(\mathbb{R}^n, G)$. This is the *isotypic decomposition* of $(\mathbb{R}^n, G)$ (see [36] and Section B). The isotypic decomposition is a powerful tool for extracting information about the spectrum of $G$-maps.

If $G = S_k$, then every irreducible representation of $S_k$ is real [37, Thm. 4.3]. Suppose, as above, that $(\mathbb{R}^n, S_k) = \oplus_{i \in [m]} V_i$ and $A : \mathbb{R}^n \to \mathbb{R}^n$ is an $S_k$-map. Since the induced maps $A_{ii'} : V_i \to V_{i'}$ must be zero if $i \neq i'$, $A$ is uniquely determined by the $S_k$-maps $A_{ii} : V_i \to V_i$, $i \in [m]$. Fix $i$ and choose an $S_k$-representation $(W, S_k)$ in the isomorphism class $\mathfrak{v}_i$. Choose $S_k$-isomorphisms $W \to V_{ij}$, $j \in [p_i]$. Then $A_{ii}$ induces $\overline{A}_{ii} : W^{p_i} \to W^{p_i}$ and so determines a (real) matrix $M_i \in M(p_i, p_i)$ since $\mathrm{Hom}_{S_k}(W, W) \approx \mathbb{R}$. Different choices of $V_{ij}$, or isomorphism $W \to V_{ij}$, yield a matrix similar to $M_i$. Each eigenvalue of $M_i$ of multiplicity $r$ gives an eigenvalue of $A_{ii}$, and so of $A$, of multiplicity $r \, \mathrm{degree}(\mathfrak{v}_i)$.

**Fact 1.** (Notations and assumptions as above.) If $A$ is the Hessian, all eigenvalues are real and each eigenvalue of $M_i$ of multiplicity $r$ will be an eigenvalue of $A$ with multiplicity $r \, \mathrm{degree}(\mathfrak{v}_i)$. In particular, $A$ has most $\sum_{i \in [m]} p_i$ distinct real eigenvalues—regardless of the dimension of the underlying space.

Our strategy can be now summarized as follows. Given a local minima $\boldsymbol{W}$, we compute the isotropy group $G \subset S_k \times S_d$ of $\boldsymbol{W}$. Since the Hessian of $\mathcal{F}$ at $\boldsymbol{W}$ is a $G$-map, may use the isotypic decomposition of the action of $G$ on $M(k,d)$ to extract the spectral properties of the Hessian. In our setting, local minima have large isotropy groups, typically, as large as $\Delta(S_p \times S_{k-p})$, $0 \leq p < k/2$. Studying the Hessian at these minima requires the isotypic decomposition corresponding to $\Delta(S_p \times S_{k-p})$, $0 \leq p < k/2$, which we detail in Theorem 4 below.

### 3.3 The isotypic decomposition of $(M(k,k), S_k)$ and the spectrum at $\mathbf{W} = \mathbf{V}$

Regard $M(k,k)$ as an $S_k$-space (diagonal action). The trivial representation, denoted by $\mathfrak{t}_k$, and the standard representation, denoted by $\mathfrak{s}_k$, introduced in Example 3 are examples of the many irreducible representations of $S_k$. In the general theory, each irreducible representation of $S_k$ is associated to a

partition of the set $[k]$. The description of the isotypic decomposition of $(M(k,k), S_k)$ is relatively simple and uses just 4 irreducible representations of $S_k$ for $k \geq 4$.

- The trivial representation $\mathfrak{t}_k$ of degree 1.
- The standard representation $\mathfrak{s}_k$ of $S_k$ of degree $k - 1$.
- The exterior square representation $\mathfrak{x}_k = \wedge^2 \mathfrak{s}_k$ of degree $\frac{(k-1)(k-2)}{2}$.
- A representation $\mathfrak{y}_k$ of degree $\frac{k(k-3)}{2}$. We describe $\mathfrak{y}_k$ explicitly later in terms of symmetric matrices (formally, it is the representation associated to the partition $(k-2, 2)$).

We omit the subscript $k$ when clear from the context. Assume that $k \geq 4$. We begin with a well-known result about the representation $\mathfrak{s} \otimes \mathfrak{s}$ (see, e.g., [37]). If $\mathfrak{s} \odot \mathfrak{s}$ denotes the symmetric tensor product of $\mathfrak{s}$, then

$$\mathfrak{s} \otimes \mathfrak{s} = \mathfrak{s} \odot \mathfrak{s} + \mathfrak{x} = \mathfrak{t} + \mathfrak{s} + \mathfrak{y} + \mathfrak{x}. \tag{3.3}$$

Since all the irreducible $S_k$-representations are real, they are isomorphic to their dual representations and so we have the isotypic decomposition

$$M(k,k) \quad \approx \quad \mathbb{R}^k \otimes \mathbb{R}^k \approx (\mathfrak{s} + \mathfrak{t}) \otimes (\mathfrak{s} + \mathfrak{t}) = 2\mathfrak{t} + 3\mathfrak{s} + \mathfrak{x} + \mathfrak{y}, \tag{3.4}$$

since $\mathfrak{t} \otimes \mathfrak{s} = \mathfrak{s}$ and $\mathfrak{t} \otimes \mathfrak{t} = \mathfrak{t}$.

Using Fact 1, information can immediately be deduced from Equation (3.4). For example, if $\boldsymbol{W}$ is a critical point of isotropy $\Delta S_k$ (a fixed point of the $S_k$-action on $M(k,k)$), then the spectrum of the Hessian contains at most $2 + 3 + 1 + 1 = 7$ distinct eigenvalues which distribute as follows: $\mathfrak{t}$ contributes 2 eigenvalues of multiplicity 1, $\mathfrak{s}$ contributes 2 eigenvalues of multiplicity $k - 1$, $\mathfrak{x}$ contributes one eigenvalue of multiplicity $\frac{(k-1)(k-2)}{2}$, and $\mathfrak{y}$ contributes one eigenvalue of multiplicity $\frac{k(k-3)}{2}$. This applies to the global minimum $\boldsymbol{W} = \boldsymbol{V}$ and the spurious minimum of type A.

Next, we would like to compute the actual eigenvalues. We demonstrate the method for the single $\mathfrak{x}$-eigenvalue. Pick a non-zero vector from the $\mathfrak{x}$-representation. For example,

$$\mathfrak{X}^k = \begin{bmatrix} 0 & 1 & \dots & 1 & -(k-2) \\ -1 & 0 & \dots & 0 & 1 \\ \dots & \dots & \dots & \dots & \dots \\ -1 & 0 & \dots & 0 & 1 \\ (k-2) & -1 & \dots & -1 & 0 \end{bmatrix},$$

where rows and columns sum to zero and the only non-zero entries are in rows and columns 1 and $k$. Let $\overline{\mathfrak{X}^k} \in \mathbb{R}^{k \times k}$ be defined by concatenating the rows of $\mathfrak{X}^k$. Since $\mathfrak{x}$ only occurs once in the isotopic decomposition and $\nabla^2 \mathcal{L}(\boldsymbol{V})$ is $S_k$-equivariant, $\overline{\mathfrak{X}^k}$ must be an eigenvector. In particular, $(\nabla^2 \mathcal{L}(\boldsymbol{V}) \overline{\mathfrak{X}^k})_i = \lambda_{\mathfrak{x}} \overline{\mathfrak{X}_i^k}$, all $i \in [k^2]$. Choose $i$ so that $\overline{\mathfrak{X}_i^k} \neq 0$. For example, $\overline{\mathfrak{X}_2^k} = 1$. Matrix multiplication, yields $\lambda_{\mathfrak{x}} = 1/4 - 1/2\pi$ (see Section C for expressions for the Hessian entries).

A similar analysis holds for the eigenvalue associated to $\mathfrak{y}$. The multiple factors $2\mathfrak{t}$ and $3\mathfrak{s}$ are handled by making judicious choices of orthogonal invariant subspaces and representative vectors in $M(k,k)$. A complete derivation of all the eigenvalues, including a detailed list of the representative vectors and expressions for the Hessian of $\mathcal{L}$ at $\boldsymbol{V}$, are provided in the appendix.

## 4  The Hessian spectrum at spurious minima

Having described the general strategy for analyzing the Hessian spectrum for global minima, we now examine the spectrum at various types of spurious minima. We need two additional ingredients: a specification of the entries of a given family of spurious minima and the respective isotypic decomposition; we begin with the latter.

As discussed in the introduction, the symmetry-based analysis of the Hessian relies on the fact that isotropy groups of spurious minima tend to be (and some provably are) maximal subgroups of the target matrix isotropy. For $\boldsymbol{V} = \boldsymbol{I}$, the relevant maximal isotropy groups are of the form $\Delta(S_p \times S_q)$, $p + q = k$. Below, we provide the corresponding isotypic decompositions. Assume $d = k$ and regard $M(k,k)$ as an $S_p \times S_q$-space, where $S_p \times S_q \subset S_k$ and the (diagonal) action of $S_k$ is restricted to the subgroup $S_p \times S_q$.

**Theorem 4.** *The isotypic decomposition of* $(M(k,k), S_p \times S_q)$ *is given by:*

1. *If* $p = k - 1$, $q = 1$, *and* $k \geq 5$,
$$M(k,k) = 5\mathfrak{t} + 5\mathfrak{s}_{k-1} + \mathfrak{x}_{k-1} + \mathfrak{y}_{k-1}.$$

2. *If* $q \geq 2$, $k - 1 > p > p/2$ *and* $k \geq 4 + q$, *then*
$$M(k,k) = 6\mathfrak{t} + 6\mathfrak{s}_p + a\mathfrak{s}_q + \mathfrak{x}_p + \mathfrak{y}_p + b\mathfrak{x}_q + c\mathfrak{y}_q + 2\mathfrak{s}_p \boxtimes \mathfrak{s}_q,$$
*where if* $q = 2$, *then* $a = 4, b = c = 0$; *if* $q = 3$, *then* $a = 5, b = 1, c = 0$; *and if* $q \geq 4$,
*then* $a = 6, b = c = 1$.

[Theorem 4](#) implies that the Hessian spectrum of local minima (or critical points) with isotropy $\Delta(S_p \times S_q)$ has at most 12 distinct eigenvalues if (1) applies, and if (2) holds, at most 19 distinct eigenvalues if $q = 2$, at most 21 distinct eigenvalues if $q = 3$, and at most 22 distinct eigenvalues if $q \geq 4$. Moreover, $k^2 - O(k)$ of the $k^2$ eigenvalues (counting multiplicity) are the $\mathfrak{x}$- and $\mathfrak{y}$-eigenvalues. We omit some less interesting cases when $k$ is small.

Following the same lines of argument described in [Section 3.3](#), our goal is to pick a set of non-zero vectors for each irreducible representation that will allow us to compute the spectrum. While this is simple, estimating the Hessian is not trivial. For this, we need good estimates on the critical points determining the spurious local minima.

In a recent work [24], three infinite families of critical points were described: type A of isotropy $\Delta S_k$, and types I and II of isotropy $\Delta S_{k-1}$. These relatively large isotropy groups made it possible to derive power series in $1/\sqrt{k}$ for the critical points and compute the initial terms. Estimates resulting from these series allow us get sharp estimates on the Hessian which in turn lead to sharp estimates on eigenvalues. The derivation is lengthy and quite technical and is therefore deferred to the appendix. As an illustration of the method, we sketch the derivation of the $\mathfrak{x}$-eigenvalue estimate for the family of type II local minima (case 1 in [Theorem 4](#)).

Briefly, if $(\mathfrak{c}_k)_{k \geq 3}$ denotes the sequence of type II critical points of $\mathcal{F}$, then we may represent $\mathfrak{c}_k$ as a point in $M(k,k)^{S_{k-1}} = \{\boldsymbol{W} \mid g\boldsymbol{W} = \boldsymbol{W}, g \in S_{k-1}\}$—the 5-dimensional fixed point space of the (diagonal) action of $S_{k-1}$ on $M(k,k)$. If $\mathfrak{c}_k = (\xi_1^k, \xi_2^k, \xi_3^k, \xi_4^k, \xi_5^k) \in M(k,k)^{S_{k-1}}$, then $\mathfrak{c}_k$ corresponds to $\boldsymbol{W} = [w_{ij}] \in M(k,k)$ where

$$w_{ii} = \left\{ \begin{array}{ll} \xi_1^k, & i < k \\ \xi_5^k, & i = k \end{array} \right., \quad w_{ij} = \left\{ \begin{array}{ll} \xi_2^k, & i, j < k, \ i \neq j \\ \xi_4^k, & i < k = j \\ \xi_3^k, & j < k = i \end{array} \right. .$$

**Lemma 5** ([24], Section 8]). *(Notation and assumptions as above.) For large enough* $k$, $\mathfrak{c}_k$ *may be written as a convergent power series in* $k^{-\frac{1}{2}}$:

$$\xi_1^k = 1 + \sum_{\ell=4}^{\infty} c_\ell k^{-\ell/2}, \quad \xi_2^k = \sum_{\ell=4}^{\infty} e_\ell k^{-\ell/2}, \quad \xi_3^k = \sum_{\ell=2}^{\infty} f_\ell k^{-\ell/2},$$

$$\xi_4^k = \sum_{\ell=2}^{\infty} g_\ell k^{-\ell/2}, \quad \xi_5^k = -1 + \sum_{\ell=2}^{\infty} d_\ell k^{-\ell/2},$$

*where*

$$c_4 = \frac{8}{\pi}, \qquad d_2 = 2 + \frac{8\pi + 8}{\pi^2}, \ e_4 = -\frac{4}{\pi}, \ f_2 = 2, \ g_2 = -e_4,$$

$$c_5 = -\frac{320\pi}{3\pi^4(\pi-2)}, \ d_3 = \frac{64\pi - 768}{3\pi^4(\pi-2)}, \ e_5 = -\frac{32}{\pi^3}, \ f_3 = 0, \ g_3 = -e_5.$$

Proceeding with the lines of argument described in [Section 3.3](#), we use these power series for $\xi_1, \xi_2, \xi_3, \xi_4, \xi_5$ to derive estimate for the Hessian entries (see Table 1), which in turn give:

$$\left( \nabla^2 \mathcal{L}(\mathfrak{c}_k) \overline{\mathfrak{X}^{k-1}} \right)_2 = (H_{22}^{11} - H_{23}^{11} - H_{33}^{12} + 2H_{13}^{12} - H_{12}^{12}) \overline{\mathfrak{X}^{k-1}}$$

$$= \left( \frac{1}{4} - \frac{1}{2\pi} - \frac{1}{\pi k} + O(k^{-2}) \right) \overline{\mathfrak{X}^{k-1}}_2,$$

| Hessian Entry | Estimate | Hessian Entry | Estimate |
|---|---|---|---|
| $H_{22}^{11}$ | $\frac{1}{2} - \frac{1}{\pi k} + O(k^{-2})$ | $H_{23}^{11}$ | $O(k^{-2})$ |
| $H_{33}^{12}$ | $\frac{1}{4} + O(k^{-2})$ | $H_{13}^{12}$ | $O(k^{-2})$ |
| $H_{12}^{12}$ | $\frac{1}{2\pi} + O(k^{-2})$ | | |

Table 1: Estimates of the Hessian entries for type II critical points based on the formula provided in Section C and Lemma 5 below. $H^{pq}$ denotes the $(p,q)$'th $k \times k$ block of the $k^2 \times k^2$ matrix $\nabla^2 \mathcal{L}(\mathfrak{c}_k)$.

showing that $\frac{1}{4} - \frac{1}{2\pi} - \frac{1}{\pi k} + O(k^{-2})$ is an eigenvalue of $\nabla^2 \mathcal{L}(\mathfrak{c}_k)$ of multiplicity $\frac{(k-2)(k-3)}{2}$ (note that the computation implicitly relies on the symmetry of the entries of $\nabla^2 \mathcal{L}(\mathfrak{c}_k)$). The complete derivation of the eigenvalue estimates stated in Theorem 2 is provided in Sections A-E.

## 5    Conclusion

We exploit the presence of rich symmetry in ST two-layers ReLU models to derive an analytic characterization of the Hessian spectrum in the natural regime where the number of inputs and hidden neurons is finite. This allow us, for the first time, to rigorously confirm (and refute) various hypotheses regarding the mysterious generalization abilities of neural networks. The methods described in the paper apply more broadly [23], and yield different spectral properties for the Hessian that vary by the choice of the underlying distributions, activation functions and architectures. The approach we wish to put forward follow in the tradition of mathematics and physics in that we start with a symmetric model, for which we can prove detailed analytic results, and subsequently break symmetry to get insight into the general theory (since critical points are non-degenerate, the results we obtain are robust under symmetry breaking perturbations of $\mathbf{V}$ [35, 9.2]; see also [22, Cor. 1]).

Some of the results derived in this work seem to challenge several research directions. Although much effort has been invested in establishing conditions under which no spurious minima exist [38, 39, 40], we prove the existence of *infinite* families of spurious minima for a simple shallow ReLU model. The hope for nonconvex optimization landscapes with no spurious minima requires therefore further refinement, at least for certain parameter regimes. Secondly, as demonstrated by type A and type II minima, not all local minima are alike. In particular, the hidden mechanism under which such spurious minima are alleviated may be somewhat different. Lastly, it is the authors' belief that a deep understanding of basic models, such as ST models, is a prerequisite for any general theory aimed at explaining the success of deep learning.

### Acknowledgements

Part of this work was completed while YA was visiting the Simons Institute for the Foundations of Deep Learning program. We thank Amir Ofer, Itai Safran, Ohad Shamir, Michal Shavit and Daniel Soudry for valuable discussions. Thanks also to Bob Howlett, University of Sydney, for help with the representation theory of $S_n$.

### Broader Impact

To the best of our knowledge, there are no ethical aspects or future societal consequences directly involved in our work.

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
