[Supplementary Material]

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

**A hitchhiker's guide to the appendix.** The appendix is organized as follows. In Section A, we provide a description of the Hessian spectrum of type I spurious minima, as well as the additional eigenvalues which correspond to the $d > k$ case. This completes the statement of Theorem 2 given in the main paper. Next, we devote Section B to representation-theoretic preliminaries for the group action under consideration. Concretely, we compute the relevant isotypic decompositions and list our choice of representative vectors. In Section C, we use the symmetry of the Hessian (w.r.t. the group action) to simplify and specialize the generic expressions of the Hessian entries to the families of spurious minima considered in this paper. Once the $\Delta S_k$-case is completed (see Section C.3), we show how to fully analyze the Hessian spectrum of global minima in a relatively simple way using symmetry (see Example 22). In Section D, the long groundwork laid in previous sections is put to use for deriving the Hessian spectrum of types A, I and II minima for $k = d$. The derivation of the additional $d > k$ case eigenvalues is presented in Section E. In Section F, we demonstrate the eigenvalue bulks phenomenon for perturbed minima, as discussed in the follow-up discussion of Theorem 1. We conclude with numerical estimates for the Hessian spectrum which we obtain through LinAlg, a linear algebra package of Python. The numerical results confirm our analytic characterization of the Hessian spectra.

## A   Type I Hessian spectrum and proof of Theorem 2

Below, we provide a description of the Hessian spectrum for type I spurious minima. This completes the statement of Theorem 2 given in the main paper.

*Theorem* 2 (Cont.). Assuming a $k \times k$ orthogonal target matrix $\boldsymbol{V}$, and $k \geq 6$, $\nabla^2 \mathcal{L}$ at type I spurious local minima has 12 distinct strictly positive eigenvalues:

1. $\frac{1}{4} - \frac{1}{2\pi} - \frac{1}{\pi\sqrt{k}} + O(k^{-1})$ of multiplicity $\frac{(k-2)(k-3)}{2}$.

2. $\frac{1}{4} + \frac{1}{2\pi} - \frac{1}{\pi\sqrt{k}} + O(k^{-1})$ of multiplicity $\frac{(k-1)(k-4)}{2}$.

3. 5 Eigenvalues of multiplicity $k - 2$, of which one grows at a rate of $\frac{k+1}{4} + O(k^{-1})$, and the rest converge to small constants.

4. 5 Eigenvalues of multiplicity 1, of which 2 grow at a rate of $c_3 + \frac{k}{4} + o(1)$, one grows at a rate of $c_4 + \frac{k}{2\pi} + o(1)$, $c_3, c_4 > 0$, and the rest converge to small constants.

We extend Theorem 2 to allow for $d > k$. Recall that if $d > k$, we append $d - k$ zeros to the end of each row of $\boldsymbol{V}$ to define $\boldsymbol{V} \in M(k, d)$. We denote the resulting objective function by $\mathcal{F}_n$, where $n = d - k$ and so $\mathcal{F}_0 = \mathcal{F}$, the objective function of Theorem 2.

**Theorem 6.** *Assume the conditions of Theorem 2 and let $d > k \geq 6$. Set $n = d - k$. The sequence of spurious minima described in Lemma 5 uniquely determines a sequence $(\mathfrak{c}_k^n)$ of critical points defining spurious minima for $\mathcal{F}_n$ which have isotropy $\Delta S_{k-1} \times S_n$ ($S_n$ permutes columns). In particular, $\mathcal{F}_n$ is real analytic at $\mathfrak{c}_k^n$, $k \geq 6$, and the spectrum of the Hessian of $\mathcal{F}_n$ will be the union of the spectrum of the Hessian of $\mathcal{F}_0$, together with 3 strictly positive eigenvalues $\lambda_1, \lambda_2, \lambda_3$ satisfying*

1. *$\lambda_1$ has multiplicity $n(k - 2)$ and $\lambda_1 = \frac{1}{4} + O(k^{-1})$.*

2. *$\lambda_2$ has multiplicity $n$ and $\lambda_2 = \frac{1}{4} + O(k^{-\frac{1}{2}})$.*

3. *$\lambda_3$ has multiplicity $n$ and $\lambda_3 = \frac{k+1}{4} + O(k^{-1})$.*

*For the spurious minima of Lemma 5, $\mathcal{F}_n(\mathfrak{c}_k^n) = (\frac{1}{2} - \frac{2}{\pi^2})k^{-1} + O(k^{-\frac{3}{2}})$.*

The proof is given in Section E.

## B   The isotypic decomposition of $(M(k, k), G)$

In this section our aim is give, with minimal prerequisites, the results needed from the representation theory of the symmetric group. A little background in character theory would be helpful for checking a few statements (for example, showing specific representations of $S_k$ are irreducible or real)—for

this the introductory text [41] would suffice. The first three or four lectures in [37] give a good, but terse, introduction to the representation theory of $S_k$. There are many texts covering the general theory, for example [42], but a lot of work is often required to extract the information needed here. Moreover, the representation theory of $S_k$ is special because the ground field can be taken to be $\mathbb{R}$ (or the rationals). Many introductory texts on representation theory work over the complex field: the proofs are often much easier but it is often awkward to translate to results over the real field.

## B.1 The isotypic decomposition

We begin with a precise version of the orthogonal decomposition described in Section 3.2. Suppose $V \subset \mathbb{R}^m$ is a linear subspace, with Euclidean inner product induced from $\mathbb{R}^m$, and $(V, G)$ is an orthogonal $G$-representation.

**Lemma 7.** *The representation $(V, G)$ may be written as an orthogonal direct sum $\bigoplus_{i=1}^m (\oplus_{j=1}^{p_i} V_{ij})$ where $V_{ij} \subset V$, $(V_{ij}, G)$ is irreducible, and $(V_{ij}, G)$ is isomorphic to $(V_{\ell k}, G)$ iff $i = \ell$, and $j, k \in [p_i]$. The subspaces $\oplus_{j=1}^{p_i} V_{ij}$ are unique, $i \in [m]$.*

**Proof** Induction on $n = \dim(V)$. Trivial for $n = 1$. Assume proved for all representations of degree less than $n$. If $(V, G)$ is of degree $n$ either it is irreducible, and there is nothing to prove, or not. If not, there exists a proper $G$-invariant linear subspace $V_1$ of $V$. By the orthogonality of the action, $V_2 = V_1^\perp$ is $G$-invariant and so $(V, G)$ is the orthogonal direct sum of representations $(V_1, G)$ and $(V_2, G)$. Apply the inductive hypothesis to $(V_1, G)$ and $(V_2, G)$. The proof of uniqueness is straightforward and we omit the details. $\square$

If $p_i = 1$, for all $i \in m$, the orthogonal decomposition given by the lemma is unique, up to order; otherwise the decomposition is not unique. For this reason, Theorem 4 was formulated in terms of isomorphism *classes* rather than in terms of specific subspaces.

In spite of the lack of uniqueness of Lemma 7, in some cases there may be *natural* choices of invariant subspace for the irreducible components. This is exactly the situation for the isotypic decomposition of $(M(k, k), G)$, $G = S_p \times S_{k-p}$, given in Theorem 4. This naturality allows us to give natural constructions of the matrices $M_i$, $i \in [m]$, used for determining the spectrum of $G$-maps $A : M(k, k) \to M(k, k)$.

**Example 8.** The isotypic decomposition for $(M(k, k), S_k)$ is $2\mathfrak{t} + 3\mathfrak{s} + \mathfrak{x} + \mathfrak{y}$, $k \geq 4$. The subspace of $M(k, k)$ determined by $2\mathfrak{t}$ is the set of all $k \times k$ matrices $\mathcal{T} = \{T_{a,b} \mid a, b \in \mathbb{R}\}$ where the diagonal entries of $T_{a,b}$ all equal $a$ and the off-diagonal entries all equal $b$. There are many ways to write $\mathcal{T}$ as an orthogonal direct sum. For example, $\mathcal{T} = T_{1,1}\mathbb{R} \oplus T_{\frac{2}{k}, -\frac{1}{k(k-1)}}\mathbb{R}$. However, there is only one natural way: $\mathcal{T} = T_{1,0}\mathbb{R} \oplus T_{0,1}\mathbb{R}$. Define $\mathfrak{D}_1^k = T_{1,0}$, $\mathfrak{D}_2^k = T_{0,1}$. If we take the *standard* realization of $(\mathfrak{t}, S_k)$ to be $(\mathbb{R}, S_k)$, where $S_k$ acts trivially on $\mathbb{R}$, then we have natural $S_k$-maps $\alpha_1, \alpha_2 : \mathbb{R} \to M(k, k)$ defined by $\alpha_i(t) = t\mathfrak{D}_i^k$, $i = 1, 2$. If $A : M(k, k) \to M(k, k)$ is an $S_k$-map, then $A$ restricts to the $S_k$-map $A_\mathfrak{t} : \mathcal{T} \to \mathcal{T}$ and $A_\mathfrak{t}$ uniquely determines a $2 \times 2$-matrix $[a_{ij}]$ by $A_\mathfrak{t}(\mathfrak{D}_i^k) = a_{i1}\mathfrak{D}_1^k + a_{i2}\mathfrak{D}_2^k$, $i = 1, 2$. The eigenvalues (and multiplicities in this case) of $A_\mathfrak{t} : \mathcal{T} \to \mathcal{T}$ are the same as the eigenvalues of $[a_{ij}]$. If we choose a different orthogonal decomposition of $\mathcal{T}$, we get a different $2 \times 2$-matrix that is similar to $[a_{ij}]$ and so has the same eigenvalues.

In the isotypic decompositions of $M(k, k)$ we consider in detail here, only $\mathfrak{t}$ and $\mathfrak{s}$ occur with multiplicity greater than 1 (later we address the exterior tensor product representation $2\mathfrak{s}_p \boxtimes \mathfrak{s}_q$—but methods are the same). Before describing how we handle the factors $\mathfrak{s}$, we need a more explicit description of the representation $(M(k, k), S_k)$.

## B.2 Decomposition of $(M(k, k), S_k)$ into spaces of matrices.

Assume $k \geq 4$ in what follows (results are easily obtained if $k \leq 3$ but are not interesting for our applications).

Let $\mathbb{D}_k$ denote the space of diagonal $k \times k$-matrices, $\mathbb{A}_k$ the space of skew-symmetric $k \times k$-matrices, and $\mathbb{S}_k$ the space of symmetric $k \times k$-matrices with diagonal entries zero. We have the orthogonal direct sum decomposition

$$M(k, k) = \mathbb{D}_k \oplus \mathbb{A}_k \oplus \mathbb{S}_k$$

Since $S_k$ acts diagonally on $M(k, k)$, this direct sum is $S_k$-invariant.

Recall that $H_{k-1} \subset \mathbb{R}^k$ is the hyperplane $\sum_{i \in [k]} x_i = 0$. In Example 3, we defined $(H_{k-1}, S_k)$ and $(T, S_k)$ to be the standard and trivial representations of $S_k$. We write here $(\mathbb{R}, S_k)$, rather than $(T, S_k)$, but caution that there is always at least one non-trivial representation of $S_k$ on $\mathbb{R}$. However, these representations do not not occur here. View $(H_{k-1}, S_k)$ and $(\mathbb{R}, S_k)$ as standard *models* or *realizations* of the isomorphism classes $\mathfrak{s}_k$ and $\mathfrak{t}$.

**Lemma 9.** $\mathbb{D}_k$ *is the orthogonal $S_k$-invariant direct sum $\mathbb{D}_{k,1} \oplus \mathbb{D}_{k,2}$, where*

1. $\mathbb{D}_{k,1}$ *is the space of diagonal matrices with all entries equal and is naturally isomorphic to $(T, S_k)$.*

2. $\mathbb{D}_{k,2}$ *is the $(k-1)$-dimensional space of diagonal matrices with diagonal entries summing to zero and is naturally isomorphic to $(H_{k-1}, S_k)$.*

*In particular, the isotypic decomposition of $(\mathbb{D}, S_k)$ is $\mathfrak{t} + \mathfrak{s}_k$.*

**Proof** For (1), define the $S_k$ map $\mathbb{R} \to \mathbb{D}_{k,1}$ by $t \mapsto t\mathfrak{D}_1^k$ and for (2), map $(x_1, \cdots, x_k) \in H_{k-1}$ to the diagonal matrix $D$ with entries $d_{ii} = x_i$, $i \in [k]$. $\qquad \square$

The lemma gives a simple instance of natural choices of subspace in the isotopic decomposition as well as a natural choice of matrix $\mathfrak{D}_1^k \in \mathbb{D}_{k,1}$ corresponding to $1 \in T$ (we give a choice of matrix for $\mathbb{D}_{k,2}$ shortly).

Next we extend the previous lemma to $\mathbb{A}_k$ and $\mathbb{S}_k$ and give and define *explicit* matrices in the isotypic components.

**Lemma 10.** $\mathbb{A}_k$ *is the orthogonal $S_k$-invariant direct sum $\mathbb{A}_{k,1} \oplus \mathbb{A}_{k,2}$, where*

1. $\mathbb{A}_{k,1}$ *is the $(k-1)$-dimensional space of matrices $[a_{ij}]$ for which there exists $(x_1, \cdots, x_k) \in H_{k-1}$ such that for all $i, j \in [k]$, $a_{ij} = x_i - x_j$,*

2. $\mathbb{A}_{k,2}$ *consists of all skew-symmetric matrices with row sums zero.*

*As representations, $(\mathbb{A}_{k,1}, S_k)$ is isomorphic to $(H_{k-1}, S_k)$ and $(\mathbb{A}_{k,2}, S_k)$ is isomorphic to $(\wedge^2 H_{k-1}, S_k)$. In particular, the isotypic decomposition of $(\mathbb{A}_k, S_k)$ is $\mathfrak{s}_k + \mathfrak{x}_k$.*

**Proof** The isotypic decomposition of $(\mathbb{A}_k, S_k)$ and irreducibility of the exterior square representation may be found in [42, 37]. Alternatively, use the explicit description and character theory to verify irreducibility. $\qquad \square$

**Lemma 11.** $\mathbb{S}_k$ *is the orthogonal $S_k$-invariant direct sum $\mathbb{S}_{1,k} \oplus \mathbb{S}_{2,k} \oplus \mathbb{S}_{3,k}$, where*

1. $\mathbb{S}_{1,k}$ *is the 1-dimensional space of symmetric matrices with diagonal entries zero and all off diagonal entries equal.*

2. $\mathbb{S}_{2,k}$ *is the $(k-1)$-dimensional space of matrices $[a_{ij}] \in \mathbb{S}_k$ for which there exists $(x_1, \cdots, x_k) \in H_{k-1}$ such that for all $i, j \in [k]$, $i \neq j$, $a_{ij} = x_i + x_j$.*

3. $\mathbb{S}_{3,k}$ *consists of all symmetric matrices in $\mathbb{S}_k$ with all row (equivalently, column) sums zero.*

4. $dim(\mathbb{S}_{3,k}) = \frac{k(k-3)}{2}$.

*The representations $(\mathbb{S}_{k,i}, S_k)$ are irreducible, $i \in [3]$: $(\mathbb{S}_{k,1}, S_k)$ is isomorphic to the trivial representation, $(\mathbb{S}_{k,2})$ is isomorphic to the standard representation and $(\mathbb{S}_{k,3}, S_k)$ is isomorphic to the $S_k$-representation associated to the partition $(k-2, 2)$ (isomorphism type $\mathfrak{y}_k$).*

**Proof** It is straightforward to check the orthogonality, (1–4) and the $S_k$-invariance of the decomposition. The isotypic decomposition of $\mathbb{S}_{1,k} \oplus \mathbb{S}_{2,k}$ is $\mathfrak{t} + \mathfrak{s}_k$. It is known that the isotypic decomposition of $(\mathbb{S}_k, S_k)$ is $\mathfrak{t} + \mathfrak{s}_k + \mathfrak{y}_k$ [42, 37]. Since we have already identified the factors $\mathfrak{t}, \mathfrak{s}_k$, $(\mathbb{S}_{k,3}, S_k)$ has isomorphism type $\mathfrak{y}_k$. Alternatively, use the explicit description of $(\mathbb{S}_{k,3}, S_k)$ and character theory to verify irreducibility—which is all we need. $\qquad \square$

## B.3 The general method

We have now identified three sub-representations in $(M(k, k), S_k)$ that are isomorphic to the standard representation $(H_{k-1}, S_k)$. Moreover lemmas 9, 10, and 11 give explicit parametrizations of the

representations in terms of the standard representation $(H_{k-1}, S_k)$. Choose a non-zero vector in $H_{k-1}$, for example $(1, -1, 0, \cdots, 0)$. Denote the corresponding elements in $\mathbb{D}_k$, $\mathbb{A}_k$ and $\mathbb{S}_k$ by $\mathfrak{S}_1^k$, $\mathfrak{S}_2^k$ and $\mathfrak{S}_3^k$ respectively. Then

$$
\mathfrak{S}_1^k = \begin{bmatrix}
1 & 0 & \dots & 0 & 0 \\
0 & -1 & \dots & 0 & 0 \\
\dots & \dots & \dots & \dots & \dots \\
0 & 0 & \dots & 0 & 0 \\
0 & 0 & \dots & 0 & 0
\end{bmatrix},
$$

$$
\mathfrak{S}_2^k = \begin{bmatrix}
0 & 2 & 1 & \dots & 1 & 1 \\
-2 & 0 & -1 & \dots & -1 & -1 \\
-1 & 1 & 0 & \dots & 0 & 0 \\
\dots & \dots & \dots & \dots & \dots & \dots \\
-1 & 1 & 0 & \dots & 0 & 0 \\
-1 & 1 & 0 & \dots & 0 & 0
\end{bmatrix}, \quad
\mathfrak{S}_3^k = \begin{bmatrix}
0 & 0 & 1 & \dots & 1 & 1 \\
0 & 0 & -1 & \dots & -1 & -1 \\
1 & -1 & 0 & \dots & 0 & 0 \\
\dots & \dots & \dots & \dots & \dots & \dots \\
1 & -1 & 0 & \dots & 0 & 0 \\
1 & -1 & 0 & \dots & 0 & 0
\end{bmatrix}.
$$

Suppose $A : M(k,k) \to M(k,k)$ is an $S_k$-map. Set $\mathbb{V} = \mathbb{D}_{k,2} \oplus \mathbb{A}_{k,1} \oplus \mathbb{S}_{k,2}$ so that $\mathbb{V}$ has isotypic decomposition $3\mathfrak{s}_k$. Setting $A_\mathfrak{s} = A|\mathbb{V}$, we have $A_\mathfrak{s} : \mathbb{V} \to \mathbb{V}$. Since $\mathfrak{s}_k$ is a real representation,

$$
A(\mathfrak{S}_i^k) = \sum_{j \in [3]} a_{ij} \mathfrak{S}_j^k, \ i \in [3],
$$

where $[a_{ij}]$ is a real $3 \times 3$-matrix. The eigenvalues of the matrix $[a_{ij}]$ give the eigenvalues of $A_\mathfrak{s} : \mathbb{V} \to \mathbb{V}$ (with multiplicities multiplied by $(k-1)$).

We have shown how to deal with multiple factors of $\mathfrak{t}$ and $\mathfrak{s}$. For the representations $\mathfrak{x}_k$ and $\mathfrak{y}_k$, we have $A|\mathfrak{x}_k = \lambda_\mathfrak{x} I$, $A|\mathfrak{y}_k = \lambda_\mathfrak{y} I$. It is enough to compute $A(M)_i$ where $M$ is a non-zero matrix in $\mathbb{A}_{k,2}$ (resp. $\mathbb{S}_{k,3}$) with $M_i \neq 0$ ($M$ in vectorized form so $i \in [k^2]$). To simplify computations, we choose matrices with many zeros and take

$$
\mathfrak{X}^k = \begin{bmatrix}
0 & 1 & 1 & \dots & 1 & -(k-2) \\
-1 & 0 & 0 & \dots & 0 & 1 \\
-1 & 0 & 0 & \dots & 0 & 1 \\
\dots & \dots & \dots & \dots & \dots & \dots \\
-1 & 0 & 0 & \dots & 0 & 1 \\
(k-2) & -1 & -1 & \dots & -1 & 0
\end{bmatrix} \in \mathbb{A}_{k,2}
$$

and

$$
\mathfrak{Y}^k = \begin{bmatrix}
0 & k-3 & 3-k & \dots & 0 & 0 & 0 \\
k-3 & 0 & 0 & \dots & -1 & -1 & -1 \\
3-k & 0 & 0 & \dots & 1 & 1 & 1 \\
0 & -1 & 1 & \dots & 0 & 0 & 0 \\
\dots & \dots & \dots & \dots & \dots & \dots \\
0 & -1 & 1 & \dots & 0 & 0 & 0 \\
0 & -1 & 1 & \dots & 0 & 0 & 0
\end{bmatrix} \in \mathbb{S}_{k,3}
$$

## B.4    Isotypic decomposition of $(M(k,k), S_p \times S_q)$

Assume $p + q = k$, regard $S_p \times S_q$ as a subgroup of $S_k$ and restrict the diagonal action of $S_k$ on $M(k,k)$ to $S_p \times S_q$ to define $M(k,k)$ as an $S_p \times S_q$-space. We assume $k > p > k/2$ so that $S_p \times S_q$ will be a maximal intransitive subgroup of $S_k$ [23, 24]. Clearly, $M(k,k)$ decomposes as an orthogonal $S_p \times S_q$-invariant direct sum

$$
M(k,k) = M(p,p) \oplus M(p,q) \oplus M(q,p) \oplus M(q,q),
$$

where $M(p,p)$ is an $S_p$-space and $M(q,q)$ is an $S_q$ space (diagonal actions). We regard $M(p,q)$ and $M(q,p)$ as $S_p \times S_q$-spaces. Thus, $S_p$ acts on $M(p,q)$ (resp. $M(q,p)$) by permuting rows (resp. columns) and $S_q$ acts on $M(p,q)$ (resp. $M(q,p)$) by permuting columns (resp. rows). At first sight this convention may seem confusing but observe that the map $M(p,q) \to M(q,p); A \mapsto A^T$, is a linear isomorphism and an $S_p \times S_q$-map. Hence the representations $(M(p,q), S_p \times S_q)$ and $(M(q,p), S_p \times S_q)$ are isomorphic.

If $A \in M(k,k)$, write $A$ in block form as $A = \begin{bmatrix} A_{p,p} & A_{p,q} \\ A_{q,p} & A_{q,q} \end{bmatrix}$, where $A_{r,s} \in M(r,s)$, $(r,s) \in \{p,q\}$. Certain special block matrices will be needed for the analysis of the eigenvalue structure. We make use of the matrices defined in the previous section.

### Block matrix decompositions related to $\mathfrak{x}$

Define
$$\mathfrak{X}^{p,p} = \begin{bmatrix} \mathfrak{X}^p & \mathbf{0}_{p,q} \\ \mathbf{0}_{q,p} & \mathbf{0}_{q,q} \end{bmatrix}, \quad \mathfrak{X}^{q,q} = \begin{bmatrix} \mathbf{0}_{q,q} & \mathbf{0}_{p,q} \\ \mathbf{0}_{q,p} & \mathfrak{X}^q \end{bmatrix},$$
where for the definition of $\mathfrak{X}^{q,q}$ it is assumed that $q \geq 3$.

### Block matrix decompositions related to $\mathfrak{y}$

Define
$$\mathfrak{y}^{p,p} = \begin{bmatrix} \mathfrak{y}^p & \mathbf{0}_{p,q} \\ \mathbf{0}_{q,p} & \mathbf{0}_{q,q} \end{bmatrix}, \quad \mathfrak{y}^{q,q} = \begin{bmatrix} \mathbf{0}_{q,q} & \mathbf{0}_{p,q} \\ \mathbf{0}_{q,p} & \mathfrak{y}^q \end{bmatrix},$$
where for the definition of $\mathfrak{Y}^{q,q}$ it is assumed that $q > 3$.

### Block matrix decompositions related to $\mathfrak{t}$

Let $\mathcal{I}_{r,s}$ denote the $r \times s$-matrix with all entries equal to 1. Define
$$\mathfrak{D}_1^{p,p} = \begin{bmatrix} \mathfrak{D}_1^p & \mathbf{0}_{p,q} \\ \mathbf{0}_{q,p} & \mathbf{0}_{q,q} \end{bmatrix}, \qquad\qquad \mathfrak{D}_2^{p,p} = \begin{bmatrix} \mathfrak{D}_2^p & \mathbf{0}_{p,q} \\ \mathbf{0}_{q,p} & \mathbf{0}_{q,q} \end{bmatrix},$$
$$\mathfrak{D}_1^{q,q} = \begin{bmatrix} \mathbf{0}_{p,p} & \mathbf{0}_{p,q} \\ \mathbf{0}_{q,p} & \mathfrak{D}_1^q \end{bmatrix}, \qquad\qquad \mathfrak{D}_2^{q,q} = \begin{bmatrix} \mathbf{0}_{p,p} & \mathbf{0}_{p,q} \\ \mathbf{0}_{q,p} & \mathfrak{D}_2^q \end{bmatrix},$$
$$\mathfrak{D}_3^{p,q} = \begin{bmatrix} \mathbf{0}_{p,p} & \mathcal{I}_{p,q} \\ \mathbf{0}_{q,p} & \mathbf{0}_{q,q} \end{bmatrix}, \qquad\qquad \mathfrak{D}_3^{q,p} = \begin{bmatrix} \mathbf{0}_{p,p} & \mathbf{0}_{p,q} \\ \mathcal{I}_{q,p} & \mathbf{0}_{q,q} \end{bmatrix},$$
where for the definition of $\mathfrak{D}_2^{q,q}$, it is assumed that $q \geq 2$.

### Block matrix decompositions related to $\mathfrak{s}$

Define $\mathfrak{S}_r^{p,q}, \mathfrak{S}_c^{p,q} \in M(p,q)$ by
$$\mathfrak{S}_r^{p,q} = \begin{bmatrix} 1 & -1 & 0 & \ldots & 0 \\ 1 & -1 & 0 & \ldots & 0 \\ \ldots & \ldots & \ldots & \ldots & \ldots \\ 1 & -1 & 0 & \ldots & 0 \end{bmatrix}, \mathfrak{S}_c^{p,q} = \begin{bmatrix} 1 & 1 & 1 & \ldots & 1 \\ -1 & -1 & -1 & \ldots & -1 \\ \ldots & \ldots & \ldots & \ldots & \ldots \\ 0 & 0 & 0 & \ldots & 0 \end{bmatrix},$$
and $\mathfrak{S}_r^{q,p}, \mathfrak{S}_c^{q,p} \in M(q,p)$ by $\mathfrak{S}_r^{q,p} = (\mathfrak{S}_c^{p,q})^T$, $\mathfrak{S}_c^{q,p} = (\mathfrak{S}_r^{p,q})^T$. Note that $\mathfrak{S}_r^{p,q}$ and $\mathfrak{S}_c^{q,p}$ are only defined if $q \geq 2$. Set

$$
\begin{aligned}
\mathfrak{S}_1^{p,p} &= \begin{bmatrix} \mathfrak{S}_1^p & \mathbf{0}_{p,q} \\ \mathbf{0}_{q,p} & \mathbf{0}_{q,q} \end{bmatrix}, & \mathfrak{S}_1^{q,q} &= \begin{bmatrix} \mathbf{0}_{p,p} & \mathbf{0}_{p,q} \\ \mathbf{0}_{q,p} & \mathfrak{S}_1^q \end{bmatrix}, \\
\mathfrak{S}_2^{p,p} &= \begin{bmatrix} \mathfrak{S}_2^p & \mathbf{0}_{p,q} \\ \mathbf{0}_{q,p} & \mathbf{0}_{q,q} \end{bmatrix}, & \mathfrak{S}_2^{q,q} &= \begin{bmatrix} \mathbf{0}_{p,p} & \mathbf{0}_{p,q} \\ \mathbf{0}_{q,p} & \mathfrak{S}_2^q \end{bmatrix}, \\
\mathfrak{S}_3^{p,p} &= \begin{bmatrix} \mathfrak{S}_3^p & \mathbf{0}_{p,q} \\ \mathbf{0}_{q,p} & \mathbf{0}_{q,q} \end{bmatrix}, & \mathfrak{S}_3^{q,q} &= \begin{bmatrix} \mathbf{0}_{p,p} & \mathbf{0}_{p,q} \\ \mathbf{0}_{q,p} & \mathfrak{S}_3^q \end{bmatrix}, \\
\mathfrak{S}_4^{p,q} &= \begin{bmatrix} \mathbf{0}_{p,p} & \mathfrak{S}_c^{p,q} \\ \mathbf{0}_{q,p} & \mathbf{0}_{q,q} \end{bmatrix}, & \mathfrak{S}_4^{q,p} &= \begin{bmatrix} \mathbf{0}_{p,p} & \mathfrak{S}_r^{p,q} \\ \mathbf{0}_{q,p} & \mathbf{0}_{q,q} \end{bmatrix}, \\
\mathfrak{S}_5^{p,q} &= \begin{bmatrix} \mathbf{0}_{p,p} & \mathbf{0}_{p,q} \\ \mathfrak{S}_r^{q,p} & \mathbf{0}_{q,q} \end{bmatrix}, & \mathfrak{S}_5^{q,p} &= \begin{bmatrix} \mathbf{0}_{p,p} & \mathbf{0}_{p,q} \\ \mathfrak{S}_c^{q,p} & \mathbf{0}_{q,q} \end{bmatrix}.
\end{aligned}
$$

The first column defines representative elements in each of the factors comprising $5\mathfrak{s}_p$ that lie in the image of of $(1, -1, 0, \cdots, 0) \in H_{p-1}$ by the natural map of $(H_{p-1}, S_p)$ onto that factor; similarly for the second column (with $p$ replaced by $q$).

**Block matrix decompositions related to $\mathfrak{s}_p \boxtimes \mathfrak{s}_q$**

Recall that $\mathfrak{s}_p \boxtimes \mathfrak{s}_q$ is the exterior tensor product of the $S_p$-representation $\mathfrak{s}_p$ and the $S_q$-representation $\mathfrak{s}_q$. The degree of $\mathfrak{s}_p \boxtimes \mathfrak{s}_q$ is $(p-1)(q-1)$. Since $p > k/2$, $p \neq q$ and so $\mathfrak{s}_p \boxtimes \mathfrak{s}_q$ is irreducible. Just as we view $(M(p,q), S_p \times S_q)$ and $(M(q,p), S_p \times S_q)$ as isomorphic representations, we regard $\mathfrak{s}_q \boxtimes \mathfrak{s}_p$ as the isomorphism class of an $S_p \times S_q$ representation and then $\mathfrak{s}_p \boxtimes \mathfrak{s}_q = \mathfrak{s}_q \boxtimes \mathfrak{s}_p$.

Assume $q \geq 2$. Define

$$\mathfrak{H}_u^{p,q} = \begin{bmatrix} 1 & -1 & 0 & \cdots & 0 \\ -1 & 1 & 0 & \cdots & 0 \\ 0 & 0 & 0 & \cdots & 0 \\ \cdots & \cdots & \cdots & \cdots & \cdots \\ 0 & 0 & 0 & \cdots & 0 \end{bmatrix} \in M(p,q), \quad \mathfrak{H}_l^{q,p} = (\mathfrak{H}_u^{p,q})^T \in M(q,p)$$

Define

$$\mathfrak{H}^{p,q} = \begin{bmatrix} \mathbf{0}_{p,p} & \mathfrak{H}_u^{p,q} \\ \mathbf{0}_{q,p} & \mathbf{0}_{q,q} \end{bmatrix}, \quad \mathfrak{H}^{q,p} = \begin{bmatrix} \mathbf{0}_{p,p} & \mathbf{0}_{p,q} \\ \mathfrak{H}_l^{q,p} & \mathbf{0}_{q,q} \end{bmatrix},$$

Note that $\mathfrak{H}_1^{p,q} \in M(p,q)$, $\mathfrak{H}_l^{q,p} \in M(q,p)$. The isotopic decomposition of $(M(p,q), S_p \times S_q)$ (equivalently, $(M(q,p), S_p \times S_q)$) is

$$(\mathfrak{s}_p + \mathfrak{t}) \boxtimes (\mathfrak{s}_q + \mathfrak{t})) = \mathfrak{s}_p \boxtimes \mathfrak{s}_q + \mathfrak{s}_p + \mathfrak{s}_q + \mathfrak{t}$$

Hence $M(p,q) \oplus M(q,p)$ contributes $2\mathfrak{s}_p \boxtimes \mathfrak{s}_q + 2\mathfrak{s}_p + 2\mathfrak{s}_q + 2\mathfrak{t}$ to the isotypic decomposition of $(M(k,k), S_p \times S_q)$.

If $A : M(p,q) \to M(q,p)$ is an $S_p \times S_q$-map, then $A(\mathfrak{H}_u^{p,q}) = c\mathfrak{H}_l^{q,p}$, for some $c \in \mathbb{R}$. as an $S_p \times S_q$ representation (we switch the order of the action on the target). In particular, the linear isomorphism $H : M(p,q) \to M(q,p)$, $A \mapsto A^T$, is an $S_p \times S_q$-map and $H(\mathfrak{H}_1^{p,q}) = \mathfrak{H}_1^{q,p}$. In order to compute spectrum associated to $2\mathfrak{s}_p \boxtimes \mathfrak{s}_q$, we use the representative matrices $\mathfrak{H}^{p,q}, \mathfrak{H}^{q,p}$. Trivial factors add the representative matrices $\mathfrak{D}_3^{p,q}, \mathfrak{D}_3 q, p$ and $\mathfrak{s}_p, \mathfrak{s}_q$ add the 4 representative matrices $\mathfrak{S}_i^{r,s}$, where $i = 4, 5$ and $r, s \in \{p, q\}$, $r \neq s$.

All the algebra is now in place for computing the spectra of $S_p \times S_q$-maps of $M(k,k)$, where $p, q, k$ satisfy the conditions of Theorem 4.

## C   Computation of the Hessian of $\mathcal{F}$.

We assume that $d = k$ (the case $d > k$ is done is Section E). We make use of the computations given in [22, 4.3.1] where the parameters are viewed as column vectors rather than as row vectors, the natural choice for the matrix formalism. The result, however, is independent of whatever viewpoint is adopted. Here we represent as columns (labelled by superscripts) to keep compatibility with notation in [22]. The result we give applies to the case $\mathbf{W} = \mathbf{V}$ although the Hessian formula [22, 4.1.1] is not well-defined when $\mathbf{W} = \mathbf{V}$ (division by zero). We remark that $\mathcal{F}(\mathbf{W})$ is $C^2$ at $\mathbf{W} = \mathbf{V}$, but not real analytic, or even smooth [24].

It follows from our analysis that if none of the parameter vectors $\boldsymbol{w}^j$ in $\mathbf{W}$ has isotropy $\Delta S_k$, then the Hessian depends only on (a) the angles between between parameter vectors and (b) angles between parameter vectors and the the target parameters $\mathbf{v}^j$, $j \in [k]$ determining $\mathbf{V}$. In particular, *there is no dependence on the norms $\|\boldsymbol{w}^j\|$*. If $\mathbf{W}$ has isotropy $\Delta S_{k-1}$, and $\boldsymbol{w}^k \neq \mathbf{0}$, then a similar result holds but now with mild dependence on norms of parameters. Isotropy groups which are not diagonal often lead to parallel parameter vectors and loss of differentiability of $\mathcal{F}$ (see [24, Ex. 4.9]).

Henceforth, we always assume that no rows of $\mathbf{W} = [\boldsymbol{w}^1, \cdots, \boldsymbol{w}^k]$ are parallel. In particular, that $\boldsymbol{w}^j \neq \mathbf{0}$ and $|\langle \boldsymbol{w}^i, \boldsymbol{w}^j \rangle| \neq \|\boldsymbol{w}^i\|\|\boldsymbol{w}^j\|$, $i, j \in [k]$, $i \neq j$.

### C.1   Formula for the Hessian $\mathcal{F}$ at $\mathbf{W} = [\mathbf{w}^1, \cdots, \mathbf{w}^k]$

We recall some results and notation from [22, 4.4.1]. Specifically, for non-parallel $\boldsymbol{w}, \mathbf{v} \in \mathbb{R}^k$, let $\theta_{\boldsymbol{w},\mathbf{v}} \in (0, \pi)$ denote the angle between $\boldsymbol{w}, \mathbf{v}$ and define

1. $\mathbf{n}_{\boldsymbol{w},\mathbf{v}} = \frac{\boldsymbol{w}}{\|\boldsymbol{w}\|} - \cos(\theta_{\boldsymbol{w},\mathbf{v}}) \frac{\mathbf{v}}{\|\mathbf{v}\|}$.

2. $\bar{\mathbf{n}}_{\boldsymbol{w},\mathbf{v}} = \frac{\mathbf{n}_{\boldsymbol{w},\mathbf{v}}}{\|\mathbf{n}_{\boldsymbol{w},\mathbf{v}}\|}$.

Note that

$$\|\mathbf{n}_{\boldsymbol{w},\mathbf{v}}\| = \sin(\theta_{\boldsymbol{w},\mathbf{v}}). \tag{C.5}$$

If $\mathbf{v}, \boldsymbol{w}$ are parallel but not zero, then $\mathbf{n}_{\boldsymbol{w},\mathbf{v}} = \mathbf{0}$ and we *define* $\bar{\mathbf{n}}_{\boldsymbol{w},\mathbf{v}} = \mathbf{0}$—this choice gives the correct value for the Hessian $H$ of $\mathcal{F}$ if $\boldsymbol{W} = \mathbf{V}$.

We write the Hessian $H$ of $\mathcal{F}$ as a $k \times k$-matrix of $k \times k$-blocks: $H = [H^{pq}]$. Since $H$ is symmetric, $H^{pq} = (H^{qp})^T$, $p, q \in [k]$, and $H^{pp}$ is symmetric. Each block $H^{pq}$ corresponds to derivatives with respect to $\boldsymbol{w}^p, \boldsymbol{w}^q$.

Let $\mathbf{I} = \mathbf{I}_k \in M(k,k)$ denote the identity matrix. Given non-parallel $\boldsymbol{w}, \mathbf{v} \in \mathbb{R}^k$, define $h_1, h_2 \in M(k,k)$ by

$$h_1(\boldsymbol{w}, \mathbf{v}) = \frac{\sin(\theta_{\boldsymbol{w},\mathbf{v}})\|\mathbf{v}\|}{2\pi\|\boldsymbol{w}\|} \left( \mathbf{I} - \frac{\boldsymbol{w}\boldsymbol{w}^T}{\|\boldsymbol{w}\|^2} + \bar{\mathbf{n}}_{\mathbf{v},\boldsymbol{w}}\bar{\mathbf{n}}_{\mathbf{v},\boldsymbol{w}}^T \right), \tag{C.6}$$

$$h_2(\boldsymbol{w}, \mathbf{v}) = \frac{1}{2\pi} \left( (\pi - \theta_{\boldsymbol{w},\mathbf{v}})\mathbf{I} + \frac{\bar{\mathbf{n}}_{\boldsymbol{w},\mathbf{v}}\mathbf{v}^T}{\|\mathbf{v}\|} + \frac{\bar{\mathbf{n}}_{\mathbf{v},\boldsymbol{w}}\boldsymbol{w}^T}{\|\boldsymbol{w}\|} \right). \tag{C.7}$$

**Lemma 12** ([22, Theorem 5]). *The Hessian* $H = [H^{pq}]$ *of* $\mathcal{F}$ *at the critical point* $\boldsymbol{W} = [\boldsymbol{w}^1, \cdots, \boldsymbol{w}^k]$ *is given by*

$$H^{pp} = \frac{1}{2}\mathbf{I} + \sum_{q \in \mathbf{k}} (h_1(\boldsymbol{w}^p, \boldsymbol{w}^q) - h_1(\boldsymbol{w}^p, \mathbf{v}^q)), \ p \in [k],$$

$$H^{pq} = h_2(\boldsymbol{w}^p, \boldsymbol{w}^q), \ p, q \in [k], \ p \neq q.$$

## C.2 Expressions for $h_1, h_2$.

We work towards obtaining more geometric expressions for the blocks $H^{pq}$. This will involve a careful analysis of the terms $h_1, h_2$ in the preceeding lemma. The term $h_1(\boldsymbol{w}^p, \mathbf{v}^p)$, used only in the description of the diagonal blocks, is particularly tricky as $\boldsymbol{w}^p$ is often close to being parallel to $\mathbf{v}^p$ in our applications.

Let $\langle \ , \ \rangle$ denote the standard Euclidean inner product on $\mathbb{R}^k$ and $\veebar$ denote the *exclusive* or.

**Lemma 13.** *If* $q \in [k]$, $\boldsymbol{w} = [w_1, \cdots, w_k]^T \in \mathbb{R}^k$ *and* $\boldsymbol{w}, \mathbf{v}^q$ *are not parallel, then*

$$h_1(\boldsymbol{w}, \mathbf{v}^q)_{ij} = \frac{\sin(\alpha_{\boldsymbol{w}q})}{2\pi\|\boldsymbol{w}\|} \left( \delta_{ij} - \frac{w_i w_j}{\|\boldsymbol{w}\|^2} + K_{ij}^{\boldsymbol{w}q} \frac{w_i w_j}{\|\boldsymbol{w}\|^2} \right), \ (i,j) \neq (q,q)$$

$$h_1(\boldsymbol{w}, \mathbf{v}^q)_{qq} = \frac{\sin^3(\alpha_{\boldsymbol{w}q})}{\pi\|\boldsymbol{w}\|},$$

*where* $\alpha_{\boldsymbol{w}q} = \cos^{-1}\left(\frac{\langle \boldsymbol{w}, \mathbf{v}^q \rangle}{\|\boldsymbol{w}\|}\right)$ *and*

$$K_{ij}^{\boldsymbol{w}q} = \begin{cases} -1, \ i \veebar j = q, \\ \frac{w_q^2}{\sum_{\ell \neq q} w_\ell^2} = \cot^2(\alpha_{\boldsymbol{w}q}), \ i, j \neq q. \end{cases}$$

**Proof** The proof is a straightforward computation using (C.6) and we only give details for $h_1(\boldsymbol{w}, \mathbf{v}^q)_{qq}$. By (C.6) and (C.5), we have

$$h_1(\boldsymbol{w}, \mathbf{v}^q)_{qq} = \frac{\sin(\alpha_{\boldsymbol{w}q})}{2\pi\|\boldsymbol{w}\|} \left[ 1 - \frac{w_q^2}{\|\boldsymbol{w}\|^2} + \left(1 - \cos(\alpha_{\boldsymbol{w}q})\frac{w_q}{\|\boldsymbol{w}\|}\right)^2 / \sin^2(\alpha_{\boldsymbol{w}q}) \right].$$

Since $\cos(\alpha_{\boldsymbol{w}q}) = \frac{w_q}{\|\boldsymbol{w}\|}$, $1 - \cos(\alpha_{\boldsymbol{w}q})\frac{w_q}{\|\boldsymbol{w}\|} = 1 - \cos^2(\alpha_{\boldsymbol{w}q}) = \sin^2(\alpha_{\boldsymbol{w}q})$. Hence $h_1(\boldsymbol{w}, \mathbf{v}^q)_{qq} = \frac{\sin(\alpha_{\boldsymbol{w}q})}{2\pi\|\boldsymbol{w}\|} \left(1 - \cos^2(\alpha_{\boldsymbol{w}q}) + \sin^2(\alpha_{\boldsymbol{w}q})\right)$ giving the result since $1 - \cos^2(\alpha_{\boldsymbol{w}q}) + \sin^2(\alpha_{\boldsymbol{w}q}) = 2\sin^2(\alpha_{\boldsymbol{w}q})$. $\square$

*Remark* 14. $K_{ij}^{wq}$ is well-defined if $i, j \neq q$—since $w$ and $\mathbf{v}^q$ are not parallel. Moreover, even though $K_{ij}^{wq}$ may be large, because of the division by $\sum_{p \neq q} w_p^2$, $|K_{ij}^{wq} w_i w_j| \leq \|w\|/2$ by the Cauchy-Schwartz inequality. This allows us to show the formula we derive below for the Hessian applies when $W = V$, even though $w^i$ is parallel to $\mathbf{v}^i$, $i \in [k]$, and that $\mathcal{F}$ is $C^2$ at $W = V$.

**Lemma 15.** *(Notation and assumptions as above.) For $i, j, p, q \in [k]$, $p \neq q$,*

$$
\begin{aligned}
h_1(\boldsymbol{w}^p, \boldsymbol{w}^q)_{ij} &= \frac{\sin(\Theta_{pq})\|\boldsymbol{w}^q\|}{2\pi\|\boldsymbol{w}_p\|} \left( \delta_{ij} - \frac{w_i^p w_j^p}{\|\boldsymbol{w}^p\|^2} \right) + \\
&\quad \frac{\|\boldsymbol{w}^q\|}{2\pi\|\boldsymbol{w}_p\|\sin(\Theta_{pq})} \left( \frac{w_i^q w_j^q}{\|\boldsymbol{w}^q\|^2} - \cos(\Theta_{pq}) \frac{w_i^p w_j^q + w_i^q w_j^p}{\|\boldsymbol{w}^p\|\|\boldsymbol{w}^q\|} + \cos^2(\Theta_{pq}) \frac{w_i^p w_j^p}{\|\boldsymbol{w}^p\|^2} \right),
\end{aligned}
$$

*where $\Theta_{pq} = \cos^{-1}\left( \frac{\langle \boldsymbol{w}^p, \boldsymbol{w}^q \rangle}{\|\boldsymbol{w}^p\|\|\boldsymbol{w}^q\|} \right)$.*

**Proof** A straightforward computation using (C.6). □

**Lemma 16.** *(Notation and assumptions as above.) For $i, j, p, q \in [k]$, $p \neq q$,*

$$
h_2(\boldsymbol{w}^p, \boldsymbol{w}^q)_{ij} = \frac{(\pi - \Theta_{pq})\delta_{ij}}{2\pi} + \frac{1}{2\pi\sin(\Theta_{pq})} L_{ij}^{pq},
$$

*where*

$$
L_{ij}^{pq} = \frac{w_i^p w_j^q + w_i^q w_j^p}{\|\boldsymbol{w}^p\|\|\boldsymbol{w}^q\|} - \cos(\Theta_{pq}) \left( \frac{w_i^p w_j^p}{\|\boldsymbol{w}^p\|^2} + \frac{w_i^q w_j^q}{\|\boldsymbol{w}^q\|^2} \right).
$$

**Proof** A straightforward computation using (C.7). □

*Remarks* 17. (1) Since no columns of $W$ are parallel, division by $\sin(\Theta_{pq})$ is safe in both lemmas. In our applications, $\sin(\Theta_{pq})$ will typically be close to 1 for large $k$.
(2) If we let $\alpha_{pq}$ denote the angle between $\boldsymbol{w}^p$ and $\mathbf{v}^q$, $p, q \in [k]$, then all of the terms in second lemma can be written in terms of the angles $\alpha_{iq}$ and $\Theta_{pq}$ with no norm terms appearing ($w_q^p/\|\boldsymbol{w}^p\| = \cos(\alpha_{pq})$). This is true in the first lemma if all $\boldsymbol{w}^j$ have the same norm.

### C.3 The Hessian of critical points with isotropy $\Delta S_k$.

We give a formula for the Hessian at critical points $W = [\boldsymbol{w}^1, \cdots, \boldsymbol{w}^k]$ with isotropy $\Delta S_k$. We continue to assume $k = d$. Since isotropy is $S_k$, columns can never be parallel: if two columns are parallel, then since $W$ is fixed by $\Delta S_k$, all columns must be equal and so the isotropy of $W$ is strictly bigger than $\Delta S_k$.

Since $W$ has isotropy $\Delta S_k$, $\|\boldsymbol{w}^i\|$ is independent of $i \in [k]$ and we set $\|\boldsymbol{w}^i\| = \tau$, $i \in [k]$. Set $w_i^i = R$, $w_i^j = S$, $i, j \in [k]$, $j \neq i$, so that the diagonal entries of $W$ are all equal to $R$, the off-diagonal entries all equal to $S$. Since the isotropy of $W$ is $\Delta S_k$, $S \neq R$. Define the angles

1. $\Theta = \cos^{-1}\left( \frac{\langle \boldsymbol{w}^i, \boldsymbol{w}^j \rangle}{\tau^2} \right)$, $i, j \in [k], i \neq j$.

2. $\alpha = \cos^{-1}\left( \frac{\langle \boldsymbol{w}^i, \mathbf{v}^j \rangle}{\tau} \right)$, $i, j \in [k], i \neq j$.

3. $\beta = \cos^{-1}\left( \frac{\langle \boldsymbol{w}^i, \mathbf{v}^i \rangle}{\tau} \right)$, $i \in [k]$.

and note that $\Theta$, $\alpha$ and $\beta$ are independent of $i, j \in [k]$.

For $i, j, p \in [k]$, we tabulate the possible values of $A_{ij}^p \doteq w_i^p w_j^p / \tau^2$.

$$
\begin{aligned}
A_{ij}^p &= R^2/\tau^2 = \cos^2(\beta), \quad i = j = p \\
A_{ij}^p &= RS/\tau^2 = \cos(\alpha)\cos(\beta), \quad i \curlyvee j = p \\
A_{ij}^p &= S^2/\tau^2 = \cos^2(\alpha), \quad i, j \neq p,
\end{aligned}
$$

and define

$$A_{ij}^{pq} = \cos^2(\alpha) + \cos^2(\beta), \qquad\qquad B_{ij}^{pq} = 2\cos(\alpha)\cos(\beta), \quad i,j \in \{p,q\}, i = j$$
$$= 2\cos(\alpha)\cos(\beta), \qquad\qquad = \cos^2(\alpha) + \cos^2(\beta), \quad i,j \in \{p,q\}, i \neq j$$
$$= \cos(\alpha)\cos(\beta) + \cos^2(\alpha), \qquad\qquad = \cos(\alpha)\cos(\beta) + \cos^2(\alpha), \quad i \veebar j \in \{p,q\}$$
$$= 2\cos^2(\alpha) \qquad\qquad = 2\cos^2(\alpha), \quad i,j \notin \{p,q\}.$$

Note that $A_{ij}^{pq}, B_{ij}^{pq}$ are symmetric in $p,q$ and $i,j$.

**Lemma 18** (Off diagonal blocks)**.** *(Notation and assumptions as above.) If $p,q,i,j \in [k]$, with $p \neq q$, then*

$$H_{ij}^{pq} = \frac{(\pi - \Theta)\delta_{ij}}{2\pi} + \frac{B_{ij}^{pq} - \cos(\Theta)A_{ij}^{pq}}{2\pi\sin(\Theta)}.$$

*In particular $H^{pq} = H^{qp}$ and are symmetric matrices.*

**Proof** Immediate from Lemma 16, the definitions of $A_{ij}^p$, $A_{ij}^{pq}$ and $B_{ij}^{pq}$ and the symmetry of $A_{ij}^{pq}, B_{ij}^{pq}$ n $p,q$ and $i,j$. □

We need a preliminary result before we give a precise description of $[H^{ij}]$.

**Lemma 19.** *(Notation and assumptions as above.) If $p,q,i,j \in [k]$, $p \neq q$, then*

$$h_1(\boldsymbol{w}^p, \boldsymbol{w}^q)_{ij} = \frac{\sin(\Theta)}{2\pi}\left(\delta_{ij} - \frac{w_i^p w_j^p}{\tau^2}\right) +$$
$$\frac{1}{2\pi\tau^2\sin(\Theta)}\left(w_i^q w_j^q - \cos(\Theta)(w_i^p w_j^q + w_i^q w_j^p) + \cos^2(\Theta)w_i^p w_j^p\right)$$
$$= \frac{\sin(\Theta)}{2\pi}\left(\delta_{ij} - A_{ij}^p\right) + \frac{1}{2\pi\sin(\Theta)}\left(A_{ij}^q - \cos(\Theta)B_{ij}^{pq} + \cos^2(\Theta)A_{ij}^p\right)$$

*Let $p,q,i,j \in [k]$. If $K_{ij}^{pq} \doteq K_{ij}^{\boldsymbol{w}^p q}$, then*

1. $p \neq q$

$$h_1(\boldsymbol{w}^p, \mathbf{v}^q)_{ij} = \frac{\sin(\alpha)}{2\pi\tau}\left(\delta_{ij} - A_{ij}^p + K_{ij}^{pq}A_{ij}^p\right), \; (i,j) \neq (q,q)$$
$$= \frac{\sin^3(\alpha)}{\pi\tau}, \; i = j = q,$$

*where if $i,j \neq q$,*

$$K_{ij}^{pq}A_{ij}^p = \begin{cases} \cos^2(\beta)\cot^2(\alpha), \; i = j = p \\ -\cos(\alpha)\cos(\beta), \; i,j \in \{p,q\}, \; i \neq j \\ \cos(\alpha)\cos(\beta)\cot^2(\alpha), \; i \veebar j = p, \; i,j \neq q \\ -\cos^2(\alpha), \; i \veebar j = q, \; i,j \neq p \\ \cos^2(\alpha)\cot^2(\alpha), \; i,j \notin \{p,q\}. \end{cases}$$

2. $p = q$

$$h_1(\boldsymbol{w}^q, \mathbf{v}^q)_{ij} = \frac{\sin(\beta)}{2\pi\tau}\left(\delta_{ij} - A_{ij}^q + K_{ij}^{qq}A_{ij}^q\right), \; (i,j) \neq (q,q)$$
$$= \frac{\sin^3(\beta)}{\pi\tau}, \; i = j = q,$$

*where if $i,j \neq q$,*

$$K_{ij}^{qq}A_{ij}^q = \begin{cases} \frac{\cos^2(\beta)}{k-1}, \; i,j \neq q. \\ -\cos(\alpha)\cos(\beta), \; i = q \neq j, \; j = q \neq i \end{cases}$$

**Proof** We verify the statements concerning the terms $K_{ij}^{qq} A_{ij}^q$. First note that both $K_{ij}^{qq}$ and $A_{ij}^q$ are symmetric in $i, j$. If $i, j \neq q$, then $K_{ij}^{qq} = \frac{(w_q^q)^2}{\sum_{\ell \neq q}(w_\ell^q)^2} = \frac{R^2}{(k-1)S^2}$, and $A_{ij}^q = \cos^2(\alpha) = \frac{S^2}{\tau^2}$. Hence $K_{ij}^{qq} A_{ij}^q = \frac{R^2}{\tau^2(k-1)} = \frac{\cos^2(\beta)}{k-1}$. On the other hand if $i = q, j \neq q$, $K_{qj}^{qq} = -1$ and $A_{qj}^q = \cos(\alpha)\cos(\beta)$. Hence $K_{qj}^{qq} A_{qj} = -\cos(\alpha)\cos(\beta)$. $\qquad\square$

**Proposition 20** (Diagonal blocks). *(Notation and assumptions as above.) Let $i, j, p \in [k]$.*

1.

$$
\begin{aligned}
H_{pp}^{pp} &= \frac{1}{2} + \frac{(k-1)\sin^2(\beta)}{2\pi}\left(\sin(\Theta) - \frac{\sin(\alpha)}{\tau}\right) - \frac{\sin^3(\beta)}{\pi\tau} + \\
&\quad \frac{(k-1)}{2\pi}\left(\frac{1}{\sin(\Theta)}\left(\cos(\alpha) - \cos(\Theta)\cos(\beta)\right)^2 - \frac{\cot(\alpha)\cos(\alpha)\cos^2(\beta)}{\tau}\right).
\end{aligned}
$$

2. *If $i \neq p$, then*

$$
\begin{aligned}
H_{ii}^{pp} &= \frac{1}{2} + \frac{(k-2)\sin^2(\alpha)}{2\pi}\left(\sin(\Theta) - \frac{\sin(\alpha)}{\tau}\right) + \\
&\quad \frac{\sin^2(\alpha)}{\pi}\left(\frac{\sin(\Theta)}{2} - \frac{\sin(\alpha)}{\tau}\right) + \frac{(k-2)}{2\pi}\left(\frac{\cos^2(\alpha)}{\sin(\Theta)}(1-\cos(\Theta))^2 - \frac{\cot(\alpha)\cos^3(\alpha)}{\tau}\right) + \\
&\quad \frac{1}{2\pi\sin(\Theta)}(\cos(\beta) - \cos(\Theta)\cos(\alpha))^2 - \frac{\sin(\beta)}{2\pi\tau}\left(\sin^2(\alpha) + \frac{\cos^2(\beta)}{k-1}\right)
\end{aligned}
$$

1. *If $i \veebar j = p$, then*

$$
\begin{aligned}
H_{1j}^{pp} &= -\frac{(k-1)\cos(\alpha)\cos(\beta)}{2\pi}\left(\sin(\Theta) - \frac{\sin(\alpha)}{\tau}\right) + \\
&\quad \frac{(k-2)\cos(\alpha)}{2\pi}\left(\frac{\cos(\alpha) - \cos(\Theta)(\cos(\alpha) + \cos(\beta)) + \cos^2(\Theta)\cos(\beta)}{\sin(\Theta)}\right) - \\
&\quad \frac{(k-2)}{2\pi\tau}\sin(\alpha)\cos(\alpha)\cos(\beta)\cot^2(\alpha) + \frac{\sin(\alpha)\cos(\alpha)\cos(\beta)}{2\pi\tau} + \\
&\quad \frac{1}{2\pi\sin(\Theta)}\left(\cos(\alpha)\cos(\beta) - \cos(\Theta)(\cos^2(\alpha) + \cos^2(\beta)) + \cos^2(\Theta)\cos(\alpha)\cos(\beta)\right) + \\
&\quad \frac{\cos(\alpha)\sin(\beta)\cos(\beta)}{\pi\tau}
\end{aligned}
$$

2. *If $i, j \neq p$, then*

$$
\begin{aligned}
H_{ij}^{pp} &= -\frac{(k-1)\cos^2(\alpha)}{2\pi}\left(\sin(\Theta) - \frac{\sin(\alpha)}{\tau}\right) + \\
&\quad \frac{(k-3)\cos^2(\alpha)}{2\pi}\left(\frac{(1-\cos(\Theta))^2}{\sin(\Theta)} - \frac{\sin(\alpha)\cot^2(\alpha)}{\tau}\right) + \\
&\quad \frac{\cos(\alpha)}{\pi}\left(\frac{\cos(\beta) - \cos(\Theta)(\cos(\beta) + \cos(\alpha)) + \cos^2(\Theta)\cos(\alpha)}{\sin(\Theta)} + \frac{\sin(\alpha)\cos(\alpha)}{\tau}\right) + \\
&\quad \frac{\sin(\beta)}{2\pi\tau}\left(\cos^2(\alpha) - \frac{\cos^2(\beta)}{k-1}\right)
\end{aligned}
$$

**Proposition 21** (Off diagonal blocks). *(Notation and assumptions as above.) Given $i, j, p, q \in [k]$, $p \neq q$.*

1. $i \in \{p, q\}$,

$$
H_{ii}^{pq} = \frac{\pi - \Theta}{2\pi} + \frac{1}{2\pi\sin(\Theta)}\left(2\cos(\alpha)\cos(\beta) - \cos(\Theta)(\cos^2(\alpha) + \cos^2(\beta))\right)
$$

2. $i \notin \{p, q\}$,

$$H_{ii}^{pq} = \frac{\pi - \Theta}{2\pi} + \frac{\cos^2(\alpha)}{\pi \sin(\Theta)} \big(1 - \cos(\Theta)\big)$$

3. $i, j \in \{p, q\}$, $i \neq j$,

$$H_{ij}^{pq} = \frac{1}{2\pi \sin(\Theta)} \big( \cos^2(\alpha) + \cos^2(\beta) - 2\cos(\Theta)\cos(\alpha)\cos(\beta) \big)$$

4. $i \veebar j \in \{p, q\}$,

$$H_{ij}^{pq} = \frac{\cos(\alpha)(\cos(\alpha) + \cos(\beta))}{2\pi \sin(\Theta)} \big(1 - \cos(\Theta)\big)$$

5. $i, j \notin \{p, q\}$,

$$H_{ij}^{pq} = \frac{\cos^2(\alpha)}{\pi \sin(\Theta)} \big(1 - \cos(\Theta)\big)$$

**Proof** Both results follow straightforwardly from the definitions of $A_{ij}^p$, $B_{ij}^{pq}$ and Lemmas 18, 19. $\quad\square$

**Example 22** (Spectrum of the Hessian at $W = V$). For $i, j \in [k]$, let $\delta_{ij}$ be equal to 1 if $i = j$, and 0 otherwise, and $\boldsymbol{\delta}_{ij} \in M(k, k)$ be the matrix with $i, j$ entry equal to 1 and all other entries zero. If $W = V$, take $\alpha = \Theta = \pi/2$ and $\beta = 0$ in Propositions 20, 21. The Hessian $H = [H^{pq}]$ of $\mathcal{F}$ at $V$ is then given by

$$H^{pp} = \frac{1}{2}\mathbf{I}, \; p \in [k],$$

$$H^{pq} = \frac{1}{4}\left(\mathbf{I} + \frac{2}{\pi}(\boldsymbol{\delta}_{pq} + \boldsymbol{\delta}_{qp})\right), \; p \neq q.$$

In particular

$$H_{ij}^{pp} = \left\{ \begin{array}{l} \frac{1}{2}, \; i = j \\ 0, \; i \neq j \end{array} \right., \qquad p \in [k],$$

$$H_{ij}^{pq} = \left\{ \begin{array}{l} \frac{1}{4}, \; i = j \\ \frac{1}{2\pi}, \; i, j \in \{p, q\}, \; i \neq j, \\ 0, \; \text{otherwise} \end{array} \right. \qquad p, q \in [k], p \neq q$$

Let $\mathbf{r}_i$ denote row $i$ of $H$, $i \in [3]$.

**The eigenvalue associated to $\mathfrak{x}_k$.**

Assume $k > 3$ and take $\mathfrak{X}^k \in \mathbb{A}_{k,2} \subset M(k, k)$ with vectorization $\overline{\mathfrak{X}^k}$ (by rows) as defined previously. Computing $\langle \mathbf{r}_2, \overline{\mathfrak{X}^k} \rangle$, we find that

$$H(\mathfrak{X}^k)_{12} = \langle \mathbf{r}_2, \overline{\mathfrak{X}^k} \rangle = \frac{1}{4} - \frac{1}{2\pi}.$$

Since $\mathfrak{X}_{12} = 1$, the eigenvalue $\lambda_{\mathfrak{x}}$ associated to $\mathfrak{x}_k$ is $\frac{1}{4} - \frac{1}{2\pi}$, and has multiplicity $(k-1)(k-2)/2$.

**The eigenvalue associated to $\mathfrak{y}_k$.**

Using the same method as above, $\lambda_{\mathfrak{y}} = \frac{1}{4} + \frac{1}{2\pi}$, for all $k \geq 5$.

**The eigenvalues associated to $\mathfrak{t}$.**

In this case, we compute $\langle \mathbf{r}_i, \mathfrak{D}_j^k \rangle$, for $i, j \in \mathbf{2}$ (see Example 8 for $\mathfrak{D}_j^k$), to find the matrix $\begin{bmatrix} \frac{1}{2} + \frac{k-1}{2\pi} & \frac{k-1}{4} \\ \frac{1}{4} & \frac{1}{2\pi} + \frac{k}{4} \end{bmatrix}$ giving the eigenvalues $\lambda_{\mathfrak{t}}^1, \lambda_{\mathfrak{t}}^2$ associated to the factor $2\mathfrak{t}$. For $k = 6$, we find $\lambda_{\mathfrak{t}}^1 = 0.8896627389$, $\lambda_{\mathfrak{t}}^2 = 2.0652669197$. As functions of $k$, $\lambda_{\mathfrak{t}}^i$ monotonically increase like $c_i k$, where $c_1 \approx 0.16$ and $c_2 = 0.25$.

**The eigenvalues associated to $\mathfrak{s}_k$.**

Denote the matrix associated to the factor $3\mathfrak{s}_k$ by $B = [\beta_{ij}] \in M(3,3)$. That is,

$$H(\mathfrak{S}_i) = \beta_{i1}\mathfrak{S}_1 + \beta_{i2}\mathfrak{S}_2 + \beta_{i3}\mathfrak{S}_3, \ i \in \mathbf{3}.$$

Since $H(\mathfrak{S}_i)_j = \langle \mathbf{r}_j, \mathfrak{S}_i \rangle$, it follows that for $i \in \mathbf{3}$,

$$\langle \mathbf{r}_1, \mathfrak{S}_i \rangle = \beta_{i1}, \quad \langle \mathbf{r}_2, \mathfrak{S}_i \rangle = 2\beta_{i2}, \quad \langle \mathbf{r}_3, \mathfrak{S}_i \rangle = \beta_{i2} + \beta_{i3}$$

where the factor 2 in the second equation occurs since the 12-component of $\mathfrak{S}_2$ is 2. Setting $h_{ij} = \langle \mathbf{r}_j, \mathfrak{S}_i \rangle$,

$$[h_{ij}] = \begin{bmatrix} \frac{1}{2} - \frac{1}{2\pi} & -\frac{1}{4} & 0 \\ -\frac{k}{4} & \frac{k+2}{4} - \frac{1}{\pi} & \frac{1}{4} - \frac{1}{2\pi} \\ \frac{k-2}{4} & -\frac{k-2}{4} & \frac{1}{4} + \frac{1}{2\pi} \end{bmatrix},$$

and so

$$B = \begin{bmatrix} \frac{1}{2} - \frac{1}{2\pi} & -\frac{1}{8} & \frac{1}{8} \\ -\frac{k}{4} & \frac{k+2}{8} - \frac{1}{2\pi} & -\frac{k}{8} \\ \frac{k-2}{4} & -\frac{k-2}{8} & \frac{k}{8} + \frac{1}{2\pi} \end{bmatrix}.$$

Since this equation has real roots for all $k \geq 5$, we can solve in terms of trigonometric functions using the formula of Fran*c*cois Viéte. From this we find that for $k = 6$, the eigenvalues are

$$\lambda_{\mathfrak{s}}^1 = 1.712918525755, \ \lambda_{\mathfrak{s}}^2 = 0.287081474245, \ \lambda_{\mathfrak{s}}^3 = 0.090845056908.$$

Numerical examination of eigenvalues for different values of $k$ reveals that the last eigenvalue is constant and equal to $\frac{1}{4} - \frac{1}{2\pi}$—the same as $\lambda_{\mathbf{r}}$. It may be shown that the characteristic equation of $B$ has the factorization

$$\left( \lambda - \frac{1}{4} + \frac{1}{2\pi} \right) \left( \lambda^2 - (\frac{k}{4} + \frac{1}{2})\lambda + \frac{1}{16}(k - 4\pi^2 + 4\pi + 1) \right).$$

Analysis of the roots of the quadratic term reveal that $\lambda_{\mathfrak{s}}^1 = \frac{k+1}{4} + O(k^{-1})$, and $\lambda_{\mathfrak{s}}^2 = \frac{1}{4} + O(k^{-1})$ is monotone decreasing with limit $0.25$. In particular, the eigenvalues of the Hessian are uniformly bounded above zero.

## C.4 The Hessian at critical points with isotropy $S_{k-1}$.

We assume $\boldsymbol{W} = [\boldsymbol{w}^1, \cdots, \boldsymbol{w}^k]$ is a critical point of $\mathcal{F}$ with isotropy $S_{k-1}$ and that $\boldsymbol{w}_k \neq \mathbf{0}$. The isotropy $S_{k-1}$ then guarantees that no two columns of $\boldsymbol{W}$ are parallel. We give an angle representation of the Hessian at $\boldsymbol{W}$. In this case, we need 7 angles which we describe below. Since $\boldsymbol{W}$ has isotropy $S_{k-1}$, $\|\boldsymbol{w}^i\|$ is independent of $i \in [k-1]$. Set $\|\boldsymbol{w}^i\| = \tau$, $i < k$, and $\|\boldsymbol{w}^k\| = \tau_k$.

Define the angles

1. $\Theta = \cos^{-1}\left( \frac{\langle \boldsymbol{w}^i, \boldsymbol{w}^j \rangle}{\tau^2} \right), i, j < k, i \neq j$.

2. $\Lambda = \cos^{-1}\left( \frac{\langle \boldsymbol{w}^i, \boldsymbol{w}^k \rangle}{\tau\tau_k} \right), i < k$.

3. $\alpha_{ii} = \cos^{-1}\left( \frac{\langle \boldsymbol{w}^i, \mathbf{v}^i \rangle}{\tau} \right) = \cos^{-1}\left( \frac{\boldsymbol{w}_i^i}{\tau} \right), i < k$.

4. $\alpha_{ij} = \cos^{-1}\left( \frac{\langle \boldsymbol{w}^i, \mathbf{v}^j \rangle}{\tau} \right) = \cos^{-1}\left( \frac{\boldsymbol{w}_j^i}{\tau} \right), i, j < k, i \neq j$.

5. $\alpha_{ik} = \cos^{-1}\left( \frac{\langle \boldsymbol{w}^i, \mathbf{v}^k \rangle}{\tau} \right) = \cos^{-1}\left( \frac{\boldsymbol{w}_k^i}{\tau} \right), i < k$.

6. $\alpha_{kk} = \cos^{-1}\left( \frac{\langle \boldsymbol{w}^k, \mathbf{v}^k \rangle}{\tau_k} \right) = \cos^{-1}\left( \frac{\boldsymbol{w}_k^k}{\tau_k} \right)$.

7. $\alpha_{kj} = \cos^{-1}\left( \frac{\langle \boldsymbol{w}^k, \mathbf{v}^j \rangle}{\tau_k} \right) = \cos^{-1}\left( \frac{\boldsymbol{w}_j^k}{\tau_k} \right), j < k$.

So as to simplify and shorten some of the expressions involved in the description of the Hessian, we set

1. $\cos(\alpha_{ij}) = \mathfrak{cos}_{ij}$, $\cos(\alpha_{ik}) = \mathfrak{cos}_{ik}$, and $\cos(\alpha_{ii}) = \mathfrak{cos}_{ii}$, $i, j < k$.
2. $\cos(\alpha_{kj}) = \mathfrak{cos}_{kj}$ and $\cos(\alpha_{kk}) = \mathfrak{cos}_{kk}$.
3. $\cot(\alpha_{ij}) = \mathfrak{cot}_{ij}$, $\cot(\alpha_{ij}) = \mathfrak{cot}_{ik}$, and $\cot(\alpha_{kj}) = \mathfrak{cot}_{kj}$
4. $\sin(\alpha_{ij}) = \mathfrak{sin}_{ij}$, $\sin(\alpha_{ik}) = \mathfrak{sin}_{ik}$ and $\sin(\alpha_{ii}) = \mathfrak{sin}_{ii}$
5. $\sin(\alpha_{kj}) = \mathfrak{sin}_{kj}$ and $\sin(\alpha_{kk}) = \mathfrak{sin}_{kk}$.

Along similar lines to the previous section, we define and tabulate the values of $A_{ij}^p$, $A_{ij}^{pq}$ and $B_{ij}^{pq}$, $i, j, p \in [k]$. For $i, j < k$, define $\rho = w_{ii}^i$, $\varepsilon = w_{ij}^i$, $\zeta = w_{ik}^i$, $\eta = w_{kj}^i$ and $\nu = w_{kk}^k$. Note that by $S_{k-1}$ symmetry, $\rho, \varepsilon, \zeta, \eta$ do not depend on the choice of $i, j \in [k-1]$.

1. $p < k$.
$$A_{ij}^p = \rho^2/\tau^2 = \mathfrak{cos}_{ii}^2, \quad i = j = p$$
$$A_{ij}^p = \rho\varepsilon/\tau^2 = \mathfrak{cos}_{ii}\mathfrak{cos}_{ij}, \quad i, j < k, \ i \veebar j = p$$
$$A_{ij}^p = \rho\zeta/\tau^2 = \mathfrak{cos}_{ii}\mathfrak{cos}_{ik}, \quad i \veebar j = k, \ i \veebar j = p$$
$$A_{ij}^p = \varepsilon^2/\tau^2 = \mathfrak{cos}_{ij}^2, \quad i, j < k, \ i, j \neq p$$
$$A_{ij}^p = \varepsilon\zeta/\tau^2 = \mathfrak{cos}_{ij}\mathfrak{cos}_{ik}, \quad i \veebar j = k, \ i, j \neq p$$
$$A_{ij}^p = \zeta^2/\tau^2 = \mathfrak{cos}_{ik}^2, \quad i = j = k$$

2. $p = k$.
$$A_{ij}^k = \nu^2/\tau_k^2 = \mathfrak{cos}_{kk}^2, \quad i = j = k$$
$$A_{ij}^k = \nu\eta/\tau_k^2 = \mathfrak{cos}_{kk}\mathfrak{cos}_{kj}, \quad i \veebar j = k$$
$$A_{ij}^k = \eta^2/\tau_k^2 = \mathfrak{cos}_{kj}^2, \quad i, j < k$$

1. If $p, q, i, j < k$, $p \neq q$, define

$$A_{ij}^{pq} = \mathfrak{cos}_{ii}^2 + \mathfrak{cos}_{ij}^2, \qquad\qquad B_{ij}^{pq} = 2\mathfrak{cos}_{ii}\mathfrak{cos}_{ij}, \quad i = j \in \{p, q\}$$
$$= 2\mathfrak{cos}_{ii}\mathfrak{cos}_{ij}, \qquad\qquad\quad = \mathfrak{cos}_{ii}^2 + \mathfrak{cos}_{ij}^2, \quad i, j \in \{p, q\}, i \neq j$$
$$= \mathfrak{cos}_{ij}\mathfrak{cos}_{ii} + \mathfrak{cos}_{ij}^2, \qquad\quad = \mathfrak{cos}_{ij}\mathfrak{cos}_{ii} + \mathfrak{cos}_{ij}^2, \quad i \veebar j \in \{p, q\}$$
$$= 2\mathfrak{cos}_{ij}^2 \qquad\qquad\qquad\quad = 2\mathfrak{cos}_{ij}^2, \quad i, j \notin \{p, q\}$$

2. If $p, q < k$, $p \neq q$, $i \veebar j = k$, define

$$A_{ij}^{pq} = \mathfrak{cos}_{ik}(\mathfrak{cos}_{ii} + \mathfrak{cos}_{ij}) \qquad B_{ij}^{pq} = \mathfrak{cos}_{ik}(\mathfrak{cos}_{ii} + \mathfrak{cos}_{ij}), \quad i \veebar j \in \{p, q\}$$
$$= 2\mathfrak{cos}_{ik}\mathfrak{cos}_{ij} \qquad\qquad\qquad = 2\mathfrak{cos}_{ik}\mathfrak{cos}_{ij}, \quad i, j \notin \{p, q\}$$

3. $p, q < k$, $p \neq q$, $i = j = k$, define

$$A_{kk}^{pq} = 2\mathfrak{cos}_{ik}^2 \qquad\qquad\qquad\qquad B_{kk}^{pq} = 2\mathfrak{cos}_{ik}^2$$

4. If $p \veebar q = k$, and $i, j < k$ define

$$A_{ij}^{pq} = \mathfrak{cos}_{kj}^2 + \mathfrak{cos}_{ii}^2, \qquad\qquad B_{ij}^{pq} = 2\mathfrak{cos}_{kj}\mathfrak{cos}_{ii}, \quad i = j \in \{p, q\}$$
$$= \mathfrak{cos}_{kj}^2 + \mathfrak{cos}_{ij}\mathfrak{cos}_{ii} \qquad\qquad = \mathfrak{cos}_{kj}(\mathfrak{cos}_{ij} + \mathfrak{cos}_{ii}), \quad i \veebar j \in \{p, q\}$$
$$= \mathfrak{cos}_{kj}^2 + \mathfrak{cos}_{ij}^2, \qquad\qquad\quad = 2\mathfrak{cos}_{kj}\mathfrak{cos}_{ij}, \quad i, j \notin \{p, q\}$$

5. If $p \veebar q = k$, and $i \veebar j = k$ define

$$A_{ij}^{pq} = \mathfrak{cos}_{kk}\mathfrak{cos}_{kj} + \mathfrak{cos}_{ik}\mathfrak{cos}_{ii}, \qquad B_{ij}^{pq} = \mathfrak{cos}_{kk}\mathfrak{cos}_{ii} + \mathfrak{cos}_{kj}\mathfrak{cos}_{ik}, \ i, j \in \{p, q\}$$
$$= \mathfrak{cos}_{kk}\mathfrak{cos}_{kj} + \mathfrak{cos}_{ik}\mathfrak{cos}_{ij} \qquad\qquad = \mathfrak{cos}_{kk}\mathfrak{cos}_{ij} + \mathfrak{cos}_{kj}\mathfrak{cos}_{ik}, \ i \veebar j \in \{p, q\}$$

6. If $p \veebar q = k$, and $i = j = k$ define

$$A_{kk}^{pq} = \mathfrak{cos}_{kk}^2 + \mathfrak{cos}_{ik}^2, \qquad\qquad\qquad B_{kk}^{pq} = 2\mathfrak{cos}_{kk}\mathfrak{cos}_{ik}$$

Note that $A_{ij}^{pq}, B_{ij}^{pq}$ are symmetric in $p, q$ and $i, j$.

**Proposition 23** (Off diagonal blocks). *(Notation and assumptions as above.) If $p, q, i, j \in [k]$, with $p \neq q$. then*

1. *If $p, q < k$,*

$$H_{ij}^{pq} = \frac{(\pi - \Theta)\delta_{ij}}{2\pi} + \frac{B_{ij}^{pq} - \cos(\Theta)A_{ij}^{pq}}{2\pi \sin(\Theta)}.$$

2. *If $p \vee q = k$,*

$$H_{ij}^{pq} = \frac{(\pi - \Lambda)\delta_{ij}}{2\pi} + \frac{B_{ij}^{pq} - \cos(\Lambda)A_{ij}^{pq}}{2\pi \sin(\Lambda)}.$$

*In particular, $H^{pq} = H^{qp}$ and are symmetric matrices.*

**Proof** Along exactly the same lines as that of Lemma 18. $\qquad\qquad\square$

Before giving the main lemma for computation of the terms $h_1(\boldsymbol{w}^p, \boldsymbol{w}^q)$, and $h_1(\boldsymbol{w}^p, \boldsymbol{v}^q)$, we need to extend the definition of $K_{ij}^{pq}$ to allow for $S_{k-1}$ symmetry.

Given $i, j, p, q \in [k]$, $p \neq q$, define for $(i, j) \neq (q, q)$,

$$K_{ij}^{pq} = \left\{ \begin{array}{l} -1, i \vee j = q \\ \cot_{ij}^2, p \neq q, \ p, q < k, \ i, j \neq q \\ \cot_{ik}^2, q = k, \ p < k, \ i, j \neq q \\ \cot_{kj}^2, p = k, \ q < k, \ i, j \neq q \end{array} \right.$$

In case $p = q$, it is more convenient to give the values of $K_{ij}^{qq} A_{ij}^q$ rather than $K_{ij}^{qq}$, for $(i, j) \neq (q, q)$ ($K_{qq}^{qq} A_{qq}^q$ is not defined). Given $i, j, p, q \in [k]$, with $p = q$, and $(i, j) \neq (q, q)$,

1. If $q \neq k$, then

$$K_{ij}^{qq} A_{ij}^q = \left\{ \begin{array}{ll} \frac{\cos_{ii}^2 \cos_{ij}^2}{(k-2)\cos_{ij}^2 + \cos_{ik}^2}, & i, j \notin \{q, k\} \\[2mm] \frac{\cos_{ii}^2 \cos_{ik}^2}{(k-2)\cos_{ij}^2 + \cos_{ik}^2}, & i, j = k \\[2mm] \frac{\cos_{ii}^2 \cos_{ij} \cos_{ik}}{(k-2)\cos_{ij}^2 + \cos_{ik}^2}, & i \vee j = k, \ i, j \neq q \\[2mm] -\cos_{ij}\cos_{ii}, & i \vee j = q, \ i, j \neq k \\[2mm] -\cos_{ik}\cos_{ii}, & i \vee j = q, \ i \vee j = k \end{array} \right.$$

2. $q = k$, then

$$K_{ij}^{kk} A_{ij}^k = \left\{ \begin{array}{ll} \frac{\cos_{kk}^2}{k-1}, & i, j \neq k, \\ -\cos_{kj}\cos_{kk}, & i \vee j = k \end{array} \right.$$

*Remark* 24. The expressions $K_{ij}^{qq} A_{ij}^q$ are all bounded by 1. For the type II critical points of $\mathcal{F}$, $K_{ij}^{qq} A_{ij}^q$ may be very small. For example, if $q \neq k$ and $i, j \notin \{q, k\}$, then $K_{ij}^{qq} A_{ij}^q = O(k^{-2})$. On the other hand if $i, j = k \neq q$, $K_{ij}^{qq} A_{ij}^q \approx 1$ for large $k$. $\qquad\maltese$

**Lemma 25.** *(Notation and assumptions as above.) If $p, q, i, j \in [k]$, $p \neq q$, then*

1. *If $p, q < k$*

$$h_1(\boldsymbol{w}^p, \boldsymbol{w}^q)_{ij} = \frac{\sin(\Theta)}{2\pi} \left(\delta_{ij} - A_{ij}^p\right) + \frac{1}{2\pi \sin(\Theta)} \left(A_{ij}^q - \cos(\Theta)B_{ij}^{pq} + \cos^2(\Theta)A_{ij}^p\right)$$

2. *if $p < k, q = k$*

$$h_1(\boldsymbol{w}^p, \boldsymbol{w}^k)_{ij} = \frac{\tau_k \sin(\Lambda)}{2\pi\tau} \left(\delta_{ij} - A_{ij}^p\right) + \frac{\tau_k}{2\pi\tau \sin(\Lambda)} \left(A_{ij}^k - \cos(\Lambda)B_{ij}^{pk} + \cos^2(\Lambda)A_{ij}^p\right)$$

3. *if $p = k, q < k$.*

$$h_1(\boldsymbol{w}^k, \boldsymbol{w}^q)_{ij} \quad = \quad \frac{\tau \sin(\Lambda)}{2\pi\tau_k} \left(\delta_{ij} - A_{ij}^k\right) + \frac{\tau}{2\pi\tau_k \sin(\Lambda)} \left(A_{ij}^q - \cos(\Lambda)B_{ij}^{kq} + \cos^2(\Lambda)A_{ij}^k\right)$$

*If $p, q, i, j \in [k]$, then*

1. *$p \neq q$, $p, q < k$.*

$$h_1(\boldsymbol{w}^p, \mathbf{v}^q)_{ij} \quad = \quad \frac{\sin(\alpha_{ij})}{2\pi\tau} \left(\delta_{ij} - A_{ij}^p + K_{ij}^{pq} A_{ij}^p\right), \ (i,j) \neq (q,q)$$

$$= \quad \frac{\sin^3(\alpha_{ij})}{\pi\tau}, \ i = j = q,$$

2. *$p \neq q$, $p < k = q$,*

$$h_1(\boldsymbol{w}^p, \mathbf{v}^k)_{ij} \quad = \quad \frac{\sin(\alpha_{ik})}{2\pi\tau} \left(\delta_{ij} - A_{ij}^p + K_{ij}^{pk} A_{ij}^p\right), \ (i,j) \neq (k,k)$$

$$= \quad \frac{\sin^3(\alpha_{ik})}{\pi\tau}, \ i = j = k,$$

3. *$q < k = p$,*

$$h_1(\boldsymbol{w}^k, \mathbf{v}^q)_{ij} \quad = \quad \frac{\sin(\alpha_{kj})}{2\pi\tau_k} \left(\delta_{ij} - A_{ij}^k + K_{ij}^{kq} A_{ij}^k\right), \ (i,j) \neq (q,q)$$

$$= \quad \frac{\sin^3(\alpha_{kj})}{\pi\tau_k}, \ i = j = q,$$

4. *$p = q < k$,*

$$h_1(\boldsymbol{w}^p, \mathbf{v}^p)_{ij} \quad = \quad \frac{\sin(\alpha_{ii})}{2\pi\tau} \left(\delta_{ij} - A_{ij}^p + K_{ij}^{pp} A_{ij}^p\right), \ (i,j) \neq (p,p)$$

$$= \quad \frac{\sin^3(\alpha_{ii})}{\pi\tau}, \ i = j = p,$$

5. *$p = q = k$,*

$$h_1(\boldsymbol{w}^k, \mathbf{v}^k)_{ij} \quad = \quad \frac{\sin(\alpha_{kk})}{2\pi\tau_k} \left(\delta_{ij} - A_{ij}^k + K_{ij}^{kk} A_{ij}^k\right), \ (i,j) \neq (k,k)$$

$$= \quad \frac{\sin^3(\alpha_{kk})}{\pi\tau_k}, \ i = j = k,$$

**Proposition 26** (Off diagonal blocks). *Assume $p, q, i, j \in [k]$ and $p \neq q$.*

*(A) If $p, q < k$, $i = j$, then if*

*(a) $i \notin \{p,q\}$, $i < k$,*

$$H_{ii}^{pq} \quad = \quad \frac{\pi - \Theta}{2\pi} + \frac{\cos_{ij}^2(1 - \cos(\Theta))}{\pi \sin(\Theta)}$$

*(b) $i \in \{p,q\}$,*

$$H_{ii}^{pq} \quad = \quad \frac{\pi - \Theta}{2\pi} + \frac{2\cos_{ii}\cos_{ij} - (\cos_{ii}^2 + \cos_{ij}^2)\cos(\Theta)}{2\pi \sin(\Theta)}$$

*(c) $i = k$,*

$$H_{kk}^{pq} \quad = \quad \frac{\pi - \Theta}{2\pi} + \frac{\cos_{ik}^2(1 - \cos(\Theta))}{\pi \sin(\Theta)}$$

*(B) If $p \veebar q = k$, $i = j$, then if*

   *(a) $i < k$, $i = p \veebar q$,*

$$H_{ii}^{pq} = \frac{\pi - \Lambda}{2\pi} + \frac{2\cos_{ii}\cos_{kj} - (\cos_{ii}^2 + \cos_{kj}^2)\cos(\Lambda)}{2\pi\sin(\Lambda)}$$

   *(b) $i < k$, $i \notin \{p, q\}$,*

$$H_{ii}^{pq} = \frac{\pi - \Lambda}{2\pi} + \frac{2\cos_{ij}\cos_{kj} - (\cos_{ij}^2 + \cos_{kj}^2)\cos(\Lambda)}{2\pi\sin(\Lambda)}$$

   *(c) $i = k$,*

$$H_{kk}^{pq} = \frac{\pi - \Lambda}{2\pi} + \frac{2\cos_{ik}\cos_{kk} - (\cos_{ik}^2 + \cos_{kk}^2)\cos(\Lambda)}{2\pi\sin(\Lambda)}$$

*(C) If $p, q < k$, $i \neq j$, then if*

   *(a) $i, j \notin \{p, q\}$, $i, j < k$,*

$$H_{ij}^{pq} = \frac{\cos_{ij}^2(1 - \cos(\Theta))}{\pi\sin(\Theta)}$$

   *(b) $i \veebar j \in \{p, q\}$, $i, j < k$,*

$$H_{ij}^{pq} = \frac{\cos_{ij}\cos_{ii} + \cos_{ij}^2}{2\pi\sin(\Theta)}\big(1 - \cos(\Theta)\big)$$

   *(c) $i, j \in \{p, q\}$, $i, j < k$,*

$$H_{ij}^{pq} = \frac{1}{2\pi\sin(\Theta)}\big(\cos_{ii}^2 + \cos_{ij}^2 - 2\cos(\Theta)\cos_{ii}\cos_{ij}\big)$$

   *(d) $i \veebar j \in \{p, q\}$, $i \veebar j = k$,*

$$H_{ij}^{pq} = \frac{\cos_{ik}(\cos_{ii} + \cos_{ij})}{2\pi\sin(\Theta)}\big(1 - \cos(\Theta)\big)$$

   *(e) $i, j \notin \{p, q\}$, $i \veebar j = k$,*

$$H_{ij}^{pq} = \frac{\cos_{ik}\cos_{ij}}{\pi\sin(\Theta)}\big(1 - \cos(\Theta)\big)$$

*(D) If $p \veebar q = k$, $i \neq j$, then if*

   *(a) $i, j \notin \{p, q\}$, $i, j < k$,*

$$H_{ij}^{pq} = \frac{2\cos_{ij}\cos_{kj} - (\cos_{ij}^2 + \cos_{kj}^2)\cos(\Lambda)}{2\pi\sin(\Lambda)}$$

   *(b) $i \veebar j \in \{p, q\}$, $i, j < k$,*

$$H_{ij}^{pq} = \frac{\cos_{kj}(\cos_{ij} + \cos_{ii}) - \cos(\Lambda)(\cos_{kj}^2 + \cos_{ij}\cos_{ii})}{2\pi\sin(\Lambda)}$$

   *(c) $i \veebar j = k$, $i, j \in \{p, q\}$.*

$$H_{ij}^{pq} = \frac{\cos_{kk}\cos_{ii} + \cos_{kj}\cos_{ik} - \cos(\Lambda)(\cos_{kk}\cos_{kj} + \cos_{ik}\cos_{ii})}{2\pi\sin(\Lambda)}$$

   *(d) $i \veebar j = k$, and $i \veebar j \notin \{p, q\}$.*

$$H_{ij}^{pq} = \frac{\cos_{kk}\cos_{ij} + \cos_{ik}\cos_{kj} - \cos(\Lambda)(\cos_{kk}\cos_{kj} + \cos_{ik}\cos_{ij})}{2\pi\sin(\Lambda)}$$

In particular, $H^{pq} = H^{qp}$ and the matrices $H^{pq}$ are all symmetric.

**Proposition 27** (Diagonal blocks). *Assume $p, i, j \in [k]$.*

*(A) If $i = j$, then if*

*(a) $i \notin \{p, k\}$, $p < k$,*

$$
\begin{aligned}
H_{ii}^{pp} &= \frac{1}{2} + \frac{(k-2)\sin_{ij}^2}{2\pi}\left(\sin(\Theta) - \frac{\sin_{ij}}{\tau}\right) + \\
&\quad \frac{(k-3)}{2\pi}\left(\frac{\cos_{ij}^2(1-\cos(\Theta))^2}{\sin(\Theta)} - \frac{\sin_{ij}\cot_{ij}^2\cos_{ij}^2}{\tau}\right) + \\
&\quad \frac{(\cos_{ii} - \cos(\Theta)\cos_{ij})^2}{2\pi\sin(\Theta)} - \frac{\sin_{ij}^3}{2\pi\tau} + \frac{\sin_{ij}^2(\tau_k\sin(\Lambda) - \sin_{ik})}{2\pi\tau} - \frac{\sin_{ik}\cot_{ik}^2\cos_{ij}^2}{2\pi\tau} + \\
&\quad \frac{\tau_k}{2\pi\tau\sin(\Lambda)}\left(\cos_{kj} - \cos(\Lambda)\cos_{ij}\right)^2 - \frac{\sin_{ii}}{2\pi\tau}\left(\sin_{ij}^2 + \frac{\cos_{ii}^2\cos_{ij}^2}{(k-2)\cos_{ij}^2 + \cos_{ik}^2}\right)
\end{aligned}
$$

*(b) $i = p$, $p < k$,*

$$
\begin{aligned}
H_{pp}^{pp} &= \frac{1}{2} + \frac{(k-2)\sin_{ii}^2}{2\pi}\left(\sin(\Theta) - \frac{\sin_{ij}}{\tau}\right) + \\
&\quad \frac{(k-2)}{2\pi}\left(\frac{(\cos_{ij} - \cos(\Theta)\cos_{ii})^2}{\sin(\Theta)} - \frac{\sin_{ij}\cos_{ii}^2\cot_{ij}^2}{\tau}\right) + \\
&\quad \frac{\sin_{ii}^2}{2\pi\tau}\left(\tau_k\sin(\Lambda) - \sin_{ik}\right) - \frac{\sin_{ii}^3}{\pi\tau} + \\
&\quad \frac{\tau_k}{2\pi\tau\sin(\Lambda)}\left(\cos_{kj} - \cos(\Lambda)\cos_{ii}\right)^2 - \frac{\sin_{ik}\cot_{ik}^2\cos_{ii}^2}{2\pi\tau}
\end{aligned}
$$

*(c) if $i = k$, $p < k$,*

$$
\begin{aligned}
H_{kk}^{pp} &= \frac{1}{2} + \frac{(k-2)\sin_{ik}^2}{2\pi}\left(\sin(\Theta) - \frac{\sin_{ij}}{\tau}\right) + \\
&\quad \frac{(k-2)\cos_{ik}^2}{2\pi}\left(\frac{(1-\cos(\Theta))^2}{\sin(\Theta)} - \frac{\sin_{ij}\cot_{ij}^2}{\tau}\right) + \\
&\quad \frac{\sin_{ik}^2}{2\pi\tau}\left(\tau_k\sin(\Lambda) - \sin_{ii}\right) - \frac{\sin_{ik}^3}{\pi\tau} + \\
&\quad \frac{\tau_k}{2\pi\tau\sin(\Lambda)}\left(\cos_{kk} - \cos(\Lambda)\cos_{ik}\right)^2 - \\
&\quad \frac{\sin_{ii}}{2\pi\tau}\left(\frac{\cos_{ii}^2\cos_{ik}^2}{(k-2)\cos_{ij}^2 + \cos_{ik}^2}\right)
\end{aligned}
$$

*(d) $i \neq k$, $p = k$,*

$$
\begin{aligned}
H_{ii}^{kk} &= \frac{1}{2} + \frac{(k-1)\sin_{kj}^2}{2\pi\tau_k}\left(\tau\sin(\Lambda) - \sin_{kj}\right) + \\
&\quad \frac{(k-2)}{2\pi\tau_k}\left(\frac{\tau(\cos_{ij} - \cos(\Lambda)\cos_{kj})^2}{\sin(\Lambda)} - \sin_{kj}\cos_{kj}^2\cot_{kj}^2\right) - \\
&\quad \frac{\sin_{kj}^3}{2\pi\tau_k} - \frac{\sin_{kk}}{2\pi\tau_k}\left(\sin_{kj}^2 + \frac{\cos_{kk}^2}{k-1}\right) + \\
&\quad \frac{\tau}{2\pi\tau_k\sin(\Lambda)}\left(\cos_{ii} - \cos(\Lambda)\cos_{kj}\right)^2
\end{aligned}
$$

*(e)* $i = k,\ p = k,$

$$
\begin{aligned}
H_{kk}^{kk} &= \frac{1}{2} + \frac{(k-1)\sin_{kk}^2}{2\pi\tau_k}\big(\tau\sin(\Lambda) - \sin_{kj}\big) + \\
&\quad \frac{(k-1)}{2\pi\tau_k}\left(\frac{\tau(\cos_{ik} - \cos(\Lambda)\cos_{kk})^2}{\sin(\Lambda)} - \cot_{kj}^2\cos_{kk}^2\sin_{kj}\right) - \frac{\sin_{kk}^3}{\pi\tau_k}
\end{aligned}
$$

*(B) If $i \neq j$, then if*

*(a)* $i, j \notin \{p, k\},\ p < k,$

$$
\begin{aligned}
H_{ij}^{pp} &= \frac{\cos_{ij}^2(k-2)}{2\pi}\left(\frac{\sin_{ij}}{\tau} - \sin(\Theta)\right) + \frac{(k-4)\cos_{ij}^2}{2\pi\sin(\Theta)}\big(1 - \cos(\Theta)\big)^2 - \\
&\quad \frac{(k-4)\sin_{ij}\cos_{ij}^2\cot_{ij}^2}{2\pi\tau} + \frac{\sin_{ij}\cos_{ij}^2}{\pi\tau} + \\
&\quad \frac{\cos_{ij}}{\pi\sin(\Theta)}\big(\cos_{ii} - \cos(\Theta)(\cos_{ii} + \cos_{ij}) + \cos^2(\Theta)\cos_{ij}\big) + \\
&\quad \frac{\cos_{ij}^2}{2\pi\tau}\big(\sin_{ik} - \tau_k\sin(\Lambda)\big) + \frac{\tau_k}{2\pi\tau\sin(\Lambda)}\big(\cos_{kj} - \cos(\Lambda)\cos_{ij}\big)^2 - \\
&\quad \frac{\sin_{ik}}{2\pi\tau}\cot_{ik}^2\cos_{ij}^2 + \frac{\sin_{ii}}{2\pi\tau}\left(\cos_{ij}^2 - \frac{\cos_{ii}^2\cos_{ij}^2}{(k-2)\cos_{ij}^2 + \cos_{ik}^2}\right)
\end{aligned}
$$

*(b)* $i \veebar j = p,\ i, j \neq k,\ p < k.$

$$
\begin{aligned}
H_{ip}^{pp} &= \frac{(k-2)\cos_{ij}\cos_{ii}}{2\pi}\left(\frac{\sin_{ij}}{\tau} - \sin(\Theta)\right) + \\
&\quad \frac{(k-3)\cos_{ij}}{2\pi\sin(\Theta)}\big(\cos_{ij} - \cos(\Theta)(\cos_{ii} + \cos_{ij}) + \cos^2(\Theta)\cos_{ii}\big) - \\
&\quad \frac{(k-3)\sin_{ij}\cos_{ij}\cos_{ii}\cot_{ij}^2}{2\pi\tau} + \\
&\quad \frac{\big(\cos_{ii}\cos_{ij} - \cos(\Theta)(\cos_{ii}^2 + \cos_{ij}^2) + \cos^2(\Theta)\cos_{ii}\cos_{ij}\big)}{2\pi\sin(\Theta)} + \frac{\sin_{ij}\cos_{ij}\cos_{ii}}{2\pi\tau} + \\
&\quad \frac{\cos_{ij}\cos_{ii}}{2\pi\tau}\big(\sin_{ik} - \tau_k\sin(\Lambda)\big) - \frac{\sin_{ik}}{2\pi\tau}\cos_{ii}\cos_{ij}\cot_{ik}^2 + \\
&\quad \frac{\tau_k}{2\pi\tau\sin(\Lambda)}\big(\cos_{kj}^2 - \cos(\Lambda)(\cos_{kj}[\cos_{ij} + \cos_{ii}]) + \cos^2(\Lambda)\cos_{ii}\cos_{ij}\big) + \frac{\sin_{ii}}{\pi\tau}\cos_{ii}\cos_{ij}
\end{aligned}
$$

*(c)* $i \veebar j = k,\ i, j \neq p,\ p < k,$

$$
\begin{aligned}
H_{ik}^{pp} &= \frac{(k-2)\cos_{ij}\cos_{ik}}{2\pi}\left(\frac{\sin_{ij}}{\tau} - \sin(\Theta)\right) + \\
&\quad \frac{(k-3)\cos_{ij}\cos_{ik}}{2\pi\sin(\Theta)}\big(1 - \cos(\Theta)\big)^2 - \frac{(k-3)\sin_{ij}\cos_{ij}\cos_{ik}\cot_{ij}^2}{2\pi\tau} + \\
&\quad \frac{\cos_{ik}\big(\cos_{ii} - \cos(\Theta)(\cos_{ii} + \cos_{ij}) + \cos^2(\Theta)\cos_{ij}\big)}{2\pi\sin(\Theta)} + \frac{\sin_{ij}\cos_{ij}\cos_{ik}}{2\pi\tau} + \\
&\quad \frac{\cos_{ij}\cos_{ik}}{2\pi\tau}\big(\sin_{ik} - \tau_k\sin(\Lambda)\big) + \\
&\quad \frac{\tau_k\big(\cos_{kk}\cos_{kj} - \cos(\Lambda)(\cos_{kk}\cos_{ij} + \cos_{ik}\cos_{kj}) + \cos^2(\Lambda)\cos_{ij}\cos_{ik}\big)}{2\pi\tau\sin(\Lambda)} + \\
&\quad \frac{\sin_{ik}\cos_{ik}\cos_{ij}}{2\pi\tau} + \frac{\sin_{ii}}{2\pi\tau}\left(\cos_{ij}\cos_{ik} - \frac{\cos_{ii}^2\cos_{ij}\cos_{ik}}{(k-2)\cos_{ij}^2 + \cos_{ik}^2}\right)
\end{aligned}
$$

(d) $i, j \in \{p, k\}$, $p < k$,

$$
\begin{aligned}
H_{pk}^{pp} &= \frac{(k-2)\cos_{ii}\cos_{ik}}{2\pi}\left(\frac{\sin_{ij}}{\tau} - \sin(\Theta)\right) + \\
&\quad \frac{(k-2)\cos_{ik}}{2\pi\sin(\Theta)}\big(\cos_{ij} - \cos(\Theta)(\cos_{ii} + \cos_{ij}) + \cos^2(\Theta)\cos_{ii}\big) - \\
&\quad \frac{(k-2)\sin_{ij}\cos_{ii}\cos_{ik}\cot_{ij}^2}{2\pi\tau} + \frac{\cos_{ii}\cos_{ik}}{2\pi\tau}\big(\sin_{ik} - \tau_k\sin(\Lambda)\big) + \\
&\quad \frac{\tau_k\big(\cos_{kk}\cos_{kj} - \cos(\Lambda)(\cos_{ii}\cos_{kk} + \cos_{ik}\cos_{kj}) + \cos^2(\Lambda)\cos_{ii}\cos_{ik}\big)}{2\pi\tau\sin(\Lambda)} + \\
&\quad \frac{\sin_{ik}\cos_{ii}\cos_{ik}}{2\pi\tau} + \frac{\sin_{ii}\cos_{ii}\cos_{ik}}{\pi\tau}
\end{aligned}
$$

(e) $i, j \neq p$, $p = k$,

$$
\begin{aligned}
H_{ij}^{kk} &= \frac{(k-1)\cos_{kj}^2}{2\pi\tau_k}\big(\sin_{kj} - \tau\sin(\Lambda)\big) + \frac{(k-3)\tau}{2\pi\tau_k\sin(\Lambda)}\big(\cos_{ij} - \cos(\Lambda)\cos_{kj}\big)^2 - \\
&\quad \frac{(k-3)\sin_{kj}\cot_{kj}^2\cos_{kj}^2}{2\pi\tau_k} + \frac{\sin_{kj}\cos_{kj}^2}{\pi\tau_k} + \\
&\quad \frac{\tau}{\pi\tau_k\sin(\Lambda)}\big(\cos_{ii}\cos_{ij} - \cos(\Lambda)\cos_{kj}(\cos_{ii} + \cos_{ij}) + \cos^2(\Lambda)\cos_{kj}^2\big) + \\
&\quad \frac{\sin_{kk}}{2\pi\tau_k}\left(\cos_{kj}^2 - \frac{\cos_{kk}^2}{k-1}\right)
\end{aligned}
$$

(f) $i \vee j = p$, $p = k$,

$$
\begin{aligned}
H_{ik}^{kk} &= \frac{(k-1)\cos_{kk}\cos_{kj}}{2\pi\tau_k}\big(\sin_{kj} - \tau\sin(\Lambda)\big) - \frac{(k-2)\sin_{kj}}{2\pi\tau_k}\cot_{kj}^2\cos_{kk}\cos_{kj} + \\
&\quad \frac{(k-2)\tau\big(\cos_{ij}\cos_{ik} - \cos(\Lambda)(\cos_{kk}\cos_{ij} + \cos_{ik}\cos_{kj}) + \cos^2(\Lambda)\cos_{kk}\cos_{kj}\big)}{2\pi\tau_k\sin(\Lambda)} + \\
&\quad \frac{\sin_{kj}}{2\pi\tau_k}\cos_{kk}\cos_{kj} + \frac{\sin_{kk}}{\pi\tau_k}\cos_{kk}\cos_{kj} + \\
&\quad \frac{\tau}{2\pi\tau_k\sin(\Lambda)}\big(\cos_{ii}\cos_{ik} - \cos(\Lambda)(\cos_{kk}\cos_{ii} + \cos_{ik}\cos_{kj}) + \cos^2(\Lambda)\cos_{kk}\cos_{kj}\big)
\end{aligned}
$$

# D  Estimating the Hessian spectrum

The stage is now set for deriving the estimates for the Hessian spectrum. We first present a detailed derivation of the spectrum of type II minima which follows along the same lines of Example 22, and then briefly state the adjustments needed for the analysis of types A and I.

## D.1  The spectrum at type II minima

First, we use the infinite series representation for type II minima given in Lemma 5 to obtain the following estimates (notations as in Proposition 26 and Proposition 27):

1. $\cos(\Theta) = (2e_4 + 4)k^{-2} + 2e_5 k^{-\frac{5}{2}}$.
2. $\sin(\Theta) = 1 + O(k^{-4})$
3. $\cos(\Lambda) = -(e_4 + 2)k^{-1} - e_5 k^{-\frac{3}{2}}$.
4. $\sin(\Lambda) = 1 - \frac{(e_4+2)^2}{2k^2} - (e_4 + 2)e_5 k^{-\frac{5}{2}}$.
5. $\cos_{ii} = 1 - \frac{2}{k^2}$.
6. $\sin_{ii} = 2k^{-1} + (\frac{e_4^2}{4} + 2 - d_2)k^{-2}$.

7. $\cos_{ij} = \frac{e_4}{k^2} + e_5 k^{-\frac{5}{2}}$.

8. $\sin_{ij} = 1 + O(k^{-4})$.

9. $\cos_{ik} = 2k^{-1} + (2 - d_2)k^{-2}$.

10. $\sin_{ik} = 1 - 2k^{-2}$.

11. $\cos_{kk} = -1 + \frac{e_4^2}{2k} + e_4 e_5 k^{-\frac{3}{2}}$.

12. $\sin_{kk} = -\frac{e_4}{\sqrt{k}} - e_5 k^{-1}$.

13. $\cos_{kj} = -e_4 k^{-1} - e_5 k^{-\frac{3}{2}}$.

14. $\sin_{kj} = 1 - \frac{e_4^2}{2k^2}$.

15. $\tau = 1 + (c_4 + 2)k^{-2} + c_5 k^{-\frac{5}{2}}$

16. $\tau^{-1} = 1 - (c_4 + 2)k^{-2} - c_5 k^{-\frac{5}{2}}$.

17. $\tau_k = 1 + \frac{e_4^2 - 2d_2}{2}k^{-1} + (e_4 e_5 - d_3)k^{-\frac{3}{2}}$

18. $\tau_k^{-1} = 1 - \frac{e_4^2 - 2d_2}{2}k^{-1} - (e_4 e_5 - d_3)k^{-\frac{3}{2}}$

Next, we use these 'primitive' estimates to compute the Hessian entries. The estimates for the entries of off-diagonal blocks are obtained through the respective expressions in Proposition 26, see Table 2.

| Case | Sub-case | Hessian entry | Estimate |
|------|----------|---------------|----------|
| A | a | $H_{ii}^{pq}$ | $\frac{1}{4} + O\left(k^{-2}\right)$ |
| A | b | $H_{ii}^{pq}$ | $\frac{1}{4} + O\left(k^{-2}\right)$ |
| A | c | $H_{kk}^{pq}$ | $\frac{1}{4} + O\left(k^{-2}\right)$ |
| B | a | $H_{ii}^{pq}$ | $\frac{1}{4} - \frac{e_4}{\pi}k^{-1} - \frac{e_5 k^{-1.5}}{\pi} + O\left(k^{-2}\right)$ |
| B | b | $H_{ii}^{pq}$ | $\frac{1}{4} + \left(-\frac{e_4}{2\pi} - \frac{1}{\pi}\right)k^{-1} - \frac{e_5 k^{-1.5}}{2\pi} + O\left(k^{-2}\right)$ |
| B | c | $H_{ii}^{pq}$ | $\frac{1}{4} - \frac{2}{\pi}k^{-1} + O\left(k^{-2}\right)$ |
| C | a | $H_{ij}^{pq}$ | $O\left(k^{-2}\right)$ |
| C | b | $H_{ij}^{pq}$ | $O\left(k^{-2}\right)$ |
| C | c | $H_{ij}^{pq}$ | $\frac{1}{2\pi} + O\left(k^{-2}\right)$ |
| C | d | $H_{ij}^{pq}$ | $\frac{k^{-1}}{\pi} + O\left(k^{-2}\right)$ |
| C | e | $H_{ij}^{pq}$ | $O\left(k^{-2}\right)$ |
| D | a | $H_{ij}^{pq}$ | $O\left(k^{-2}\right)$ |
| D | b | $H_{ij}^{pq}$ | $-\frac{e_5 k^{-1.5}}{2\pi} - \frac{e_4 k^{-1}}{2\pi} + O\left(k^{-2}\right)$ |
| D | c | $H_{ij}^{pq}$ | $-\frac{1}{2\pi} + \frac{e_4 e_5 k^{-1.5}}{2\pi} + \frac{e_4^2 k^{-1}}{4\pi} + O\left(k^{-2}\right)$ |
| D | d | $H_{ij}^{pq}$ | $O\left(k^{-2}\right)$ |

Table 2: Estimates for the entries of off-diagonal blocks of the Hessian using the expression derived in Proposition 26.

Similarly, the estimates for the entries of the diagonal blocks are obtained through the relevant expressions in Proposition 27.

| Hessian entry | Estimate |
|---|---|
| $H_{ii}^{pp}$ | $0.5 + k^{-1.5}\left(\frac{0.5c_5}{\pi} - \frac{0.5d_3}{\pi} + \frac{0.5e_4}{\pi}e_5\right) + k^{-1.0}\left(\frac{0.5c_4}{\pi} - \frac{0.5d_2}{\pi} + \frac{0.25e_4^2}{\pi}\right) + \mathcal{O}\left(k^{-2}\right)$ |
| $H_{pp}^{pp}$ | $0.5 + \mathcal{O}\left(k^{-2}\right)$ |
| $H_{kk}^{pp}$ | $0.5 + k^{-1.5}\left(\frac{0.5c_5}{\pi} - \frac{1.0d_3}{\pi}\right) + k^{1.0}\left(\frac{0.5c_4}{\pi} - \frac{1.0d_2}{\pi} + \frac{1.0}{\pi}\right) + \mathcal{O}\left(k^{-2}\right)$ |
| $H_{ii}^{kk}$ | $0.5 + k^{-1.5}\left(\frac{0.5c_5}{\pi} + \frac{0.5d_2}{\pi}e_4 - \frac{0.25e_4^3}{\pi} - \frac{0.5e_4}{\pi}e_5 + \frac{e_4}{2\pi} - \frac{1.0e_5}{\pi}\right)$ |
| | $+k^{-1.0}\left(\frac{0.5c_4}{\pi} - \frac{1.0e_4}{\pi} + \frac{e_5}{2\pi}\right) + \frac{e_4 k^{-0.5}}{2\pi} + \mathcal{O}\left(k^{-2}\right)$ |
| $H_{kk}^{kk}$ | $0.5 + \frac{e_4^3 k^{-1.5}}{\pi} + \mathcal{O}\left(k^{-2}\right)$ |
| $H_{ij}^{pp}$ | $\mathcal{O}\left(k^{-2}\right)$ |
| $H_{ip}^{pp}$ | $\mathcal{O}\left(k^{-2}\right)$ |
| $H_{ik}^{pp}$ | $k^{-1.0}\left(\frac{0.5e_4}{\pi} + \frac{1}{\pi}\right) + \frac{0.5e_5}{\pi}k^{-1.5} + \mathcal{O}\left(k^{-2}\right)$ |
| $H_{pk}^{pp}$ | $\mathcal{O}\left(k^{-2}\right)$ |
| $H_{ij}^{kk}$ | $\frac{e_4 k^{-1.5}}{2\pi} + \mathcal{O}\left(k^{-2}\right)$ |
| $H_{ik}^{kk}$ | $-\frac{e_4^2 k^{-1.5}}{\pi} + \mathcal{O}\left(k^{-2}\right)$ |

Table 3: Estimates for the entries of diagonal blocks of the Hessian using Proposition 27.

Our next goal is to compute the product of the Hessian of type II minima by the representative vectors described in Section B.4.

**The eigenvalues $\lambda_{\mathfrak{r}}$ and $\lambda_{\mathfrak{y}}$.** Computing $H(\mathfrak{X}^{k-1,1})_{12} = \langle \mathbf{r}_2, \mathfrak{X}^{k-1,1}\rangle$, we find that

$$H(\mathfrak{X}^{k-1,1})_{12} = H_{22}^{11} - H_{ij}^{11} - H_{12}^{12} + 2H_{1j}^{12} - H_{33}^{12} = \frac{1}{4} - \frac{1}{2\pi} - \frac{1}{\pi k} + O(k^{-2}).$$

Since $\mathfrak{X}_{12}^{k-1,1} = 1$, $\lambda_{\mathfrak{r}} = \frac{1}{4} - \frac{1}{2\pi} - \frac{1}{\pi k} + O(k^{-2})$. Along similar lines. we find that

$$\begin{aligned}
H(\mathfrak{Y}^{k-1,1})_{12} =& (k-4)H_{22}^{11} - (k-4)H_{ij}^{11} + (k-4)H_{12}^{12} \\
& - 2(k-4))H_{1j}^{12} - (k-4)H_{33}^{12} + 2(k-4)H_{ij}^{12} \\
=& \frac{1}{4} + \frac{1}{2\pi} - \frac{1}{\pi k} + O(k^{-2}),
\end{aligned}$$

by which we conclude $\lambda_{\mathfrak{y}} = \frac{1}{4} + \frac{1}{2\pi} - \frac{1}{\pi k} + O(k^{-2})$.

*Remarks* 28. (1) The $1/k$ term that occurs for both eigenvalues appears to be a correction in going from $k$ to $k-1$—to the $(k-1)^2$ block in $M(k,k)$.

(2) Using the results in [24] it is not difficult to compute the coefficient of $k^{-2}$ in power series (in $1/\sqrt{k}$) for both eigenvalues. Although the coefficient of $k^{-\frac{3}{2}}$ is zero, the coefficients of higher order fractional powers of $1/k$ are typically non-zero. ✠

**The eigenvalues associated to $\mathfrak{s}_k$.** As in the case of $\boldsymbol{W} = \boldsymbol{V}$ (Example Example 22), we denote the matrix associated to the factor $3\mathfrak{s}_k$ by $B_{\mathfrak{s}} \in M(3,3)$, and use Table 4 to show that modulo $o(1)$ terms,

$$B_{\mathfrak{s}} = \begin{pmatrix}
0.5 & -0.25k & 0.25k & 0 & 0.25 \\
-0.125 & 0.125k + 0.5 & -0.125k & 0 & -0.125 \\
0.125 & -0.125k - 0.25 & 0.125k + 0.25 & 0 & 0.125 \\
0 & 0 & 0 & 0.25 & -0.5/\pi \\
0.25 & -0.25k & 0.25k & -0.5/\pi & 0.5
\end{pmatrix}$$

This allows us to compute the coefficients of the linear term for eigenvalues of the form $a + bk + o(1)$. Indeed, taking the limit of $B_{\mathfrak{s}}/k$ for $k \to \infty$, we have that the $b$-coefficients are the eigenvalues of

$$\begin{pmatrix}
0 & -0.25 & 0.25 & 0 & 0 \\
0 & 0.125 & -0.125 & 0 & 0 \\
0 & -0.125 & 0.125 & 0 & 0 \\
0 & 0 & 0 & 0 & 0 \\
0 & -0.25 & 0.25 & 0 & 0
\end{pmatrix}$$

which are zero, except for a single eigenvalue which equals 0.25.

|  | $\mathfrak{S}_1^{k-1,k-1}$ |
|---|---|
| $H(\cdot)_{11}$ | $H_{11}^{11} - H_{12}^{12}$ |
| $H(\cdot)_{12}$ | $H_{1j}^{11} - H_{11}^{12}$ |
| $H(\cdot)_{13}$ | $H_{1j}^{11} - H_{1j}^{12}$ |
| $H(\cdot)_{1k}$ | $H_{1k}^{11} - H_{1k}^{12}$ |
| $H(\cdot)_{k1}$ | $H_{11}^{1k} - H_{1j}^{1k}$ |
|  | $\mathfrak{S}_2^{k-1,k-1}$ |
| $H(\cdot)_{11}$ | $(k-1)H_{1j}^{11} - (k-1)H_{11}^{12}$ |
| $H(\cdot)_{12}$ | $2H_{22}^{11} + (k-3)H_{ij}^{11} - 2H_{12}^{12} - 2(k-3)H_{1j}^{12} + (k-3)H_{33}^{12}$ |
| $H(\cdot)_{13}$ | $H_{22}^{11} + (k-2)H_{ij}^{11} - H_{12}^{12} - (k-3)H_{1j}^{12} - H_{33}^{12}$ |
| $H(\cdot)_{1k}$ | $(k-1)H_{ik}^{11} - (k-1)H_{1k}^{12}$ |
| $H(\cdot)_{k1}$ | $(k-1)H_{1j}^{1k} - (k-1)H_{22}^{1k}$ |
|  | $\mathfrak{S}_3^{k-1,k-1}$ |
| $H(\cdot)_{11}$ | $(k-3)H_{1j}^{11} + (k-3)H_{11}^{12} - 2(k-3)H_{1j}^{12}$ |
| $H(\cdot)_{12}$ | $(k-3)H_{ij}^{11} - (k-3)H_{33}^{12}$ |
| $H(\cdot)_{13}$ | $H_{22}^{11} + (k-4)H_{ij}^{11} + H_{12}^{12} + (k-5)H_{1j}^{12} - H_{33}^{12} - 2(k-4)H_{ij}^{12}$ |
| $H(\cdot)_{1k}$ | $(k-3)H_{ik}^{11} + (k-3)H_{1k}^{12} - 2(k-3)H_{ik}^{12}$ |
| $H(\cdot)_{k1}$ | $(k-3)H_{1j}^{1k} + (k-3)H_{22}^{1k} - 2(k-3)H_{ij}^{1k}$ |
|  | $\mathfrak{S}_4^{k-1,1}$ |
| $H(\cdot)_{11}$ | $H_{1k}^{11} - H_{1k}^{12}$ |
| $H(\cdot)_{12}$ | $H_{ik}^{11} - H_{1k}^{12}$ |
| $H(\cdot)_{13}$ | $H_{ik}^{11} - H_{ik}^{12}$ |
| $H(\cdot)_{1k}$ | $H_{kk}^{11} - H_{kk}^{12}$ |
| $H(\cdot)_{k1}$ | $H_{1k}^{1k} - H_{ik}^{1k}$ |
|  | $\mathfrak{S}_5^{1,k-1}$ |
| $H(\cdot)_{11}$ | $H_{11}^{1k} - H_{1j}^{1k}$ |
| $H(\cdot)_{12}$ | $H_{1j}^{1k} - H_{22}^{1k}$ |
| $H(\cdot)_{13}$ | $H_{1j}^{1k} - H_{ij}^{1k}$ |
| $H(\cdot)_{1k}$ | $H_{1k}^{1k} - H_{ik}^{1k}$ |
| $H(\cdot)_{k1}$ | $H_{11}^{kk} - H_{ij}^{kk}$ |

Table 4: Product of the Hessian matrix by the representative vectors of the standard representation, as described in Section B.4

**The eigenvalues associated to $\mathfrak{t}_k$.** The matrix associated to the factor $3\mathfrak{t}_k$, $B_{\mathfrak{t}} = [\beta_{ij}] \in M(3,3)$ is computed through Table Table 5. Modulo $o(1)$ terms, we have

$$
B_{\mathfrak{t}} = \begin{pmatrix}
0.5k/\pi + 0.5 & 0.25k & 0.25 & 0 & -0.5/\pi \\
0.25 & 0.25k + 0.5 & 0.25 & 0 & 0 \\
0.25 & 0.25k & 0.5 & -0.5/\pi & 0 \\
0 & 0 & -0.5/\pi & 0.25k + 0.5 & 0.25 \\
-0.5k/\pi + 1.0 & 1.0 & 1.0 & 0.25k + 1.0 & 0.5
\end{pmatrix}
$$

This allows us to easily compute the coefficients of the linear term for eigenvalues of the form $a + bk + o(1)$. Indeed, this follows by computing the spectrum of $B_{\mathfrak{t}}/k$ where $k \to \infty$,

$$
\begin{pmatrix}
0.5/\pi & 0.25 & 0 & 0 & 0 \\
0 & 0.25 & 0 & 0 & 0 \\
0 & 0.25 & 0 & 0 & 0 \\
0 & 0 & 0 & 0.25 & 0 \\
-0.5/\pi & 0 & 0 & 0.25 & 0
\end{pmatrix}
$$

which is $0$, $\frac{1}{4}$ and $\frac{1}{2\pi}$ of multiplicity $2, 1, 2$, respectively.

| | $\mathfrak{D}_1^{k-1,k-1}$ |
|---|---|
| $H(\cdot)_{11}$ | $H_{11}^{11} + (k-2)H_{12}^{12}$ |
| $H(\cdot)_{12}$ | $H_{1i}^{11} + H_{11}^{12} + (k-3)H_{1j}^{12}$ |
| $H(\cdot)_{1k}$ | $H_{1k}^{11} + (k-2)H_{1k}^{12}$ |
| $H(\cdot)_{k1}$ | $H_{11}^{1k} + (k-2)H_{1j}^{1k}$ |
| $H(\cdot)_{kk}$ | $(k-1)H_{1k}^{1k}$ |
| | $\mathfrak{D}_2^{k-1,k-1}$ |
| $H(\cdot)_{11}$ | $(k-2)H_{1j}^{11} + (k-2)H_{11}^{12} + (k-2)(k-3)H_{1j}^{12}$ |
| $H(\cdot)_{12}$ | $H_{22}^{11} + (k-3)H_{ij}^{11} + H_{12}^{12} + 2(k-3)H_{1j}^{12} + (k-3)H_{33}^{12} + (k^2 - 7k + 8)H_{ij}^{12}$ |
| $H(\cdot)_{1k}$ | $(k-2)H_{ik}^{11} + (k-2)H_{1k}^{12} + (k-2)(k-3)H_{ik}^{12}$ |
| $H(\cdot)_{k1}$ | $(k-2)H_{1j}^{1k} + (k-2)H_{22}^{1k} + (k-2)(k-3)H_{ij}^{1k}$ |
| $H(\cdot)_{kk}$ | $(k-1)(k-2)H_{ik}^{1k}$ |
| | $\mathfrak{D}_3^{k-1,1}$ |
| $H(\cdot)_{11}$ | $H_{11}^{1k} + (k-2)H_{1j}^{1k}$ |
| $H(\cdot)_{12}$ | $H_{1j}^{1k} + H_{22}^{1k} + (k-3)H_{ij}^{1k}$ |
| $H(\cdot)_{1k}$ | $H_{1k}^{1k} + (k-2)H_{ik}^{1k}$ |
| $H(\cdot)_{k1}$ | $H_{11}^{kk} + (k-2)H_{ij}^{kk}$ |
| $H(\cdot)_{kk}$ | $(k-1)H_{ik}^{kk}$ |
| | $\mathfrak{D}_3^{1,k-1}$ |
| $H(\cdot)_{11}$ | $H_{1k}^{11} + (k-2)H_{1k}^{12}$ |
| $H(\cdot)_{12}$ | $H_{ik}^{11} + H_{1k}^{12} + (k-3)H_{ik}^{12}$ |
| $H(\cdot)_{1k}$ | $H_{kk}^{11} + (k-2)H_{kk}^{12}$ |
| $H(\cdot)_{k1}$ | $H_{1k}^{1k} + (k-2)H_{ik}^{1k}$ |
| $H(\cdot)_{kk}$ | $(k-1)H_{kk}^{1k}$ |
| | $\mathfrak{D}_1^{1,1}$ |
| $H(\cdot)_{11}$ | $H_{1k}^{1k}$ |
| $H(\cdot)_{12}$ | $H_{ik}^{1k}$ |
| $H(\cdot)_{1k}$ | $H_{kk}^{1k}$ |
| $H(\cdot)_{k1}$ | $H_{ik}^{kk}$ |
| $H(\cdot)_{kk}$ | $H_{kk}^{kk}$ |

Table 5: Product of the Hessian matrix by the representative vectors of the trivial representation, as described in Section B.4

### D.2 The spectrum of type A and type I minima

The computation of the Hessian spectrum at types A and I uses the estimates derived in [24], which we provide here for convenience. Modulo high-order terms, we have (notations as in Lemma 5)

$$\text{Type A:} \quad \xi_1, \xi_5 \sim -1 + 2k^{-1} + \left(\frac{8}{\pi} - 4\right)k^{-2}, \quad \xi_2, \xi_3, \xi_4 \sim 2k^{-1} + \left(\frac{4}{\pi} - 2\right)k^{-2}.$$

$$\text{Type I:} \quad \xi_1 = -1 + \sum_{n=2}^{\infty} c_n k^{-\frac{n}{2}}, \quad \xi_2 = \sum_{n=2}^{\infty} e_n k^{-\frac{n}{2}}, \quad \xi_5 = 1 + \sum_{n=2}^{\infty} d_n k^{-\frac{n}{2}},$$

$$\xi_3 = \sum_{n=2}^{\infty} f_n k^{-\frac{n}{2}}, \quad \xi_4 = \sum_{n=4}^{\infty} g_n k^{-\frac{n}{2}},$$

where

| | | | | | | | | |
|---|---|---|---|---|---|---|---|---|
| $c_2 =$ | $2$ | $d_2 =$ | $\frac{8(\pi-1)}{\pi^2}$ | $e_2 =$ | $2$ | $f_2 =$ | $0$ | $g_2 =$ $2 - \frac{4}{\pi}$ |
| $c_3 =$ | $0$ | $d_3 =$ | $-4.798751$ | $e_3 =$ | $0$ | $f_3 =$ | $0$ | $g_3 =$ $\frac{32}{\pi^2}\left(\frac{1}{\pi} - 1\right)$ |
| $c_4 =$ | $\frac{16}{\pi} - 4$ | | | $e_4 =$ | $\frac{8}{\pi} - 2$ | $f_4 =$ | $\frac{16}{\pi^2} - \frac{12}{\pi}$ | |
| $c_5 =$ | $4.441691$ | | | $e_5 =$ | $\frac{8(\pi^2 + 4(\pi-1))}{\pi^3}$ | $f_5 =$ | $6.205827$ | |

The rest of the derivation follows along the same lines of type II minima.

Let us show how to compute the 3 distinct eigenvalues of type A which are related to the standard representation. Here, the matrix associated with the $3\varsigma$ factor is

$$M = \begin{pmatrix} -0.5/\pi + 0.5 & 2.0/\pi - 0.25k & -0.5 + 0.25k \\ -0.125 & 0.5/\pi + 0.25 + 0.125k & 0.25 - 0.125k \\ 0.125 & -0.125k & -0.5/\pi + 0.125k \end{pmatrix}.$$

We now express the 3 eigenvalues by $a_i k + b_i + o(1)$. The coefficients of the linear terms can be computed by taking $k \to \infty$ in $M/k$, which gives

$$\begin{pmatrix} 0 & -0.25 & 0.25 \\ 0 & 0.125 & -0.125 \\ 0 & -0.125 & 0.125 \end{pmatrix},$$

whose eigenvalues are easily shown to be $0, 0, 0.25$. It remains to compute the constant terms $b_i$. To this end, note that

$$b_1 + b_2 + b_3 = \text{the constant term of trace}(M) = -0.5/\pi + 0.75,$$
$$2a_1 b_1 = \text{the coefficient of } k \text{ in trace}(M^2) = 0.125,$$
$$a_1 b_1 b_2 = \text{the coefficient of } k \text{ in } \det(M) = 0.03125/\pi + 0.015625.$$

The system of equations yields $b_1 = \frac{1}{4}, b_2 = \frac{1}{4}, b_3 = \frac{1}{4} - \frac{1}{2\pi}$.

## E    Completion of the proof of Theorem 2

### E.1    Extension to the case $d > k$

Given $d > k$, append $d - k$ zeros to the end of each row of $\boldsymbol{W} \in M(k, k)$ to define $\widetilde{\boldsymbol{W}} \in M(k, d)$. Similarly, extend the target $\mathbf{V}$ to $\widetilde{\mathbf{V}} \in M(k, d)$. Denote the associated objective function by $\widetilde{\mathcal{F}}$ and note that if $\boldsymbol{W} \in M(k, k)$ is a critical point of $\mathcal{F}$, then $\widetilde{\boldsymbol{W}} \in M(k, d)$ is a critical point of $\widetilde{\mathcal{F}}$.

We make use of the following result, adapted from Lemma 8 in [22]

**Lemma 29.** *(Notation and assumptions as above.) Assume $d > k$ and set $m = d - k$. Let $\boldsymbol{W}$ be a critical point of $\mathcal{F}$ which has no parallel rows. Then the Hessian $\widetilde{H}$ of $\widetilde{\mathcal{F}}$ at $\widetilde{\boldsymbol{W}}$ may, after a permutation of rows and columns, be written in block diagonal form $[\widetilde{H_{ii}}]_{i \in [m+1]}$ where $\widetilde{H_{11}} = H \in M(k^2, k^2)$ is the Hessian of $\mathcal{F}$, and for $i > 1$, the matrices $\widetilde{H_{ii}}$ are all equal to the $k \times k$-matrix $M = [m_{ij}]$ defined by*

$$m_{ij} = \begin{cases} \frac{1}{2} + \frac{1}{2\pi} \sum_{\ell \in [k]} \left( \frac{\sin(\theta_{\boldsymbol{w}_i, \boldsymbol{w}_\ell}) \|\boldsymbol{w}_\ell\|}{\|\boldsymbol{w}_i\|} - \frac{\sin(\theta_{\boldsymbol{w}_i, \mathbf{v}_\ell})}{\|\boldsymbol{w}_i\|} \right), & i = j \\ \frac{1}{2\pi}(\pi - \theta_{\boldsymbol{w}_i, \boldsymbol{w}_j}), & i \neq j \end{cases}$$

**Theorem 30.** *(Assumptions and notation of Theorem 2) Let $d > k$.*

1. *Suppose $\widetilde{\boldsymbol{W}} = \widetilde{\mathbf{V}}$. In addition to the eigenvalues described in Theorem 2, there will be an 2 additional eigenvalues: one equal to $\frac{1}{4}$, multiplicity $m(k-1)$, the other to $\frac{k+2}{4}$, multiplicity $m$.*

2. *Suppose $\boldsymbol{W}$ is of type A. Then $\widetilde{\boldsymbol{W}}$ will have an additional 2 eigenvalues. One equal to $\frac{1}{4} - \frac{1}{\pi\sqrt{k}} + O(k^{-1})$, multiplicity $m(k-1)$, the other to $\frac{k+1}{4} - \frac{1}{\pi\sqrt{k}} + O(k^{-1})$, multiplicity $m$.*

3. *Suppose $\boldsymbol{W}$ is of type II. Then $\widetilde{\boldsymbol{W}}$ will have an additional 3 eigenvalues. One equal to $\frac{1}{4} + \frac{4}{\pi^2 k} + O(k^{-2})$ of multiplicity $m(k-2)$, and two eigenvalues of multiplicity $m$, one equal to $\frac{k+1}{4} + O(k^{-\frac{1}{2}})$, the other to $\frac{1}{2} + O(k^{-\frac{1}{2}})$.*

*In particular, type A and type II spurious minima exist for all $d \geq k \geq 6$.*

**Proof** Suppose $\widetilde{\boldsymbol{W}} = \widetilde{\mathbf{V}}$. The matrix $M \in M(k, k)$ defines an $S_k$-map of $\mathbb{R}^k$. Computing $M$, we find that $m_{ii} = \frac{1}{2}$ and $m_{ij} = \frac{1}{4}$, $i, j \in [k]$, $i \neq j$. Write $(\mathbb{R}^k, S_k)$ uniquely as the orthogonal direct

sum $(H_{k-1}, S_k) \oplus (T, S_k)$ Since $M$ is an $S_k$-map, $M : H_{k-1} \to H_{k-1}$ and $M : T \to T$. Taking $X = [1, -1, 0, \cdots, 0] \in H_{k-1}$, $M(X) = (m_{11} - m_{12})X$, giving the eigenvalue $\frac{1}{4}$. Similarly, for the eigenvalue associated to $(T_k, S_k)$ is $\frac{k+2}{4}$. The argument for Type A critical points is similar: both the diagonal and off-diagonal entries are easily computed given the estimates on the critical points used in the proof of Theorem 2. Finally, for type II critical points, we use the $S_{k-1}$-representation $(\mathbb{R}^k, S_{k-1})$ which has isotypic decomposition $\mathfrak{s}_{k-1} + 2\mathfrak{t}$. The eigenvalue associated to the $\mathfrak{s}_{k-1}$ factor is found exactly as for type A critical points and only uses the $m_{11}$ and $m_{12}$ entries of $M$. For the eigenvalues associated to the factor $2\mathfrak{t}$, we use the realizations spanned by the basis vector $\mathbf{v}_k$ and the vector $\sum_{i \in [k-1]} \mathbf{v}_i$. However, $\sin(\theta_{\mathbf{w}_k, \mathbf{v}_k})$ appears in the expression for $m_{kk}$ and this leads to the presence of terms in $k^{-\frac{1}{2}}$ since $\sin(\theta_{\mathbf{w}_k, \mathbf{v}_k}) = \frac{4}{\pi\sqrt{k}}$ [24]. $\qquad\square$

## F  Empirical results

### F.1  Perturbing the trained model

Our analysis shows that local minima exhibit a small number of distinct eigenvalues, independent of the number inputs $d$ and hidden neurons $k$. However, during the training processes we expect to see a small number clusters of eigenvalues forming upon convergence. Below, we perturb the type II local minima of $k = 20$ by adding an independent zero-mean Gaussian noise per entry for different choices of variance.

Figure 3: The spectrum of the Hessian at type II spurious minima where the entries are perturb by adding independent zero-mean Gaussian random entries with different variance values. As expected, the eigenvalues accumulate in clusters around the eigenvalues of the type II minima.

### F.2  Eigenvalue data for type A, I, II spurious minima

In the sequel, we provide numerical estimates for the Hessian spectrum at types A, I and II minima. The Hessian is computed using the expressions given in Section C, and evaluated using the estimates of the spurious minima. The spectrum is then approximated numerically using LinAlg, a linear algebra package of Python.

| k | 4 | 5 | 6 | 7 | 8 | 9 | 10 | 11 | 12 | 13 | 14 | 15 | 20 | 50 | 100 |
|---|---|---|---|---|---|---|---|---|---|---|---|---|---|---|---|
| xi_1 | -0.4898600208 | -0.5940757868 | -0.6633973683 | -0.7126281816 | -0.7493249249 | -0.7777074711 | -0.8003040345 | -0.8187167157 | -0.8340080421 | -0.8469096664 | -0.857941499 | -0.8674829742 | -0.9007460136 | -0.9602572702 | -0.9800854958 |
| xi_2 | 0.4954785342 | 0.3967289644 | 0.3307097833 | 0.2834991026 | 0.2480699062 | 0.220516412 | 0.1984765225 | 0.1804470696 | 0.1654251182 | 0.1527161193 | 0.1418239656 | 0.132384904 | 0.09935065305 | 0.03983992056 | 0.01995135856 |
| | | | | | | | | | | | | | | | |
| Trivial Rep (times 1) | 1.4560394 | 0.70990586 | 0.8480067 | 0.990511 | 1.135952 | 1.2834839 | 1.4325787 | 1.5828876 | 1.7341673 | 1.8862413 | 2.0389771 | 2.1922746 | 2.9648666 | 7.6802244 | 15.608578 |
| | 0.57904404 | 1.7242408 | 1.9897203 | 2.2535357 | 2.5160947 | 2.7776048 | 3.0382068 | 3.298006 | 3.5570922 | 3.8155413 | 4.0734167 | 4.3307796 | 5.611412 | 13.199169 | 25.748272 |
| Standard Rep (times k-1) | 1.109484036 | 1.375470284 | 1.635817171 | 1.893362863 | 2.149283757 | 2.404151115 | 2.658279921 | 2.911859526 | 3.165013547 | 3.417828734 | 3.670365601 | 3.922670617 | 5.181727777 | 12.70512831 | 25.2178866 |
| | 0.2203240728 | 0.2049980342 | 0.1946480554 | 0.1875084205 | 0.1824611761 | 0.1788291887 | 0.1761892425 | 0.1742064477 | 0.1728748546 | 0.1718852454 | 0.1712038256 | 0.1707629932 | 0.1707309861 | 0.08774232198 | 0.08977066797 |
| | 0.04751754505 | 0.0514193864 | 0.05510416677 | 0.05842876435 | 0.0613882308 | 0.06401088927 | 0.06633332964 | 0.06839218345 | 0.07021950498 | 0.07184625776 | 0.07329648491 | 0.07459396964 | 0.07936788816 | 0.1834150619 | 0.1974143104 |
| X Rep (times (k-1)(k-2)/2) | -0.01753896847 | -0.005831507035 | 0.001503654756 | 0.006847392768 | 0.01105740014 | 0.01453184709 | 0.01748837531 | 0.02005898207 | 0.02232948318 | 0.02436031401 | 0.02619439736 | 0.02786448598 | 0.0344661884 | 0.05203547329 | 0.06214639544 |
| Y Rep (times k(k-3)/2) | 0.298961103 | 0.3122170269 | 0.3202357292 | 0.3258672953 | 0.3301824629 | 0.3336762488 | 0.3366119266 | 0.3391433954 | 0.3413672745 | 0.3433507383 | 0.3451385498 | 0.3467643261 | 0.3531934023 | 0.3704492748 | 0.3804876208 |

Figure 4: The spectrum of type A spurious minima.

| k | 5 | 6 | 7 | 8 | 9 | 10 | 11 | 12 | 13 | 14 | 15 | 20 | 50 | 100 |
|---|---|---|---|---|---|---|---|---|---|---|---|---|---|---|
| xi_1 | -0.4830355462 | -0.5877302017 | -0.6579648893 | -0.70808019 | -0.7455234023 | -0.7745103399 | -0.7975913049 | -0.8163930175 | -0.8319990065 | -0.8451574585 | -0.8564009402 | -0.8946139295 | -0.959294303 | -0.9798436861 |
| xi_2 | 0.4884077711 | 0.3911538654 | 0.3262295256 | 0.2798049673 | 0.2449666066 | 0.2178625868 | 0.1961761796 | 0.178430816 | 0.1636412895 | 0.1511256282 | 0.1403954144 | 0.1036641823 | 0.04046198792 | 0.02009875975 |
| xi_3 | 0.01440735973 | 0.01677031362 | 0.01818315076 | 0.01896456522 | 0.01932369293 | 0.01940222927 | 0.0192957152 | 0.01906863983 | 0.01876427588 | 0.0184120735 | 0.0180320798 | 0.01606495388 | 0.009090335323 | 0.005277459527 |
| xi_4 | -0.01527369217 | -0.01379889472 | -0.01202812067 | -0.01040931892 | -0.009029099865 | -0.007876361661 | -0.006917063442 | -0.006116172395 | -0.00544350044 | -0.004874521939 | -0.00438973409 | -0.002787631404 | -0.0005807633564 | -0.0001645277529 |
| xi_5 | 1.063150521 | 1.068395642 | 1.069424581 | 1.06852328 | 1.066734455 | 1.06456357 | 1.062264857 | 1.059968247 | 1.057739533 | 1.055610406 | 1.053594085 | 1.045172924 | 1.023263207 | 1.013111353 |
| | | | | | | | | | | | | | | |
| Trivial Rep (times 1) | 1.74485985 | 2.00748071 | 2.268785083 | 2.52929655 | 2.789167588 | 3.048453427 | 3.307187958 | 3.565399005 | 3.823122832 | 4.080388805 | 4.33723197 | 5.616135615 | 13.20110106 | 25.74931705 |
| | 1.396092281 | 1.648778854 | 1.901337223 | 2.153887011 | 2.406412581 | 2.658877863 | 2.911255781 | 3.163529273 | 3.415692853 | 3.667744066 | 3.919685917 | 5.177942127 | 12.70217173 | 25.21607128 |
| | 0.7221266194 | 0.8566065873 | 0.9968903313 | 1.140882465 | 1.287421413 | 1.43580951 | 1.585596641 | 1.736480223 | 1.888246349 | 2.040738104 | 2.193838276 | 2.965833076 | 7.680449743 | 15.60864623 |
| | 0.2547984971 | 0.2436021821 | 0.2360487686 | 0.230795479 | 0.2270566597 | 0.2243502517 | 0.2223668139 | 0.2209034859 | 0.2198216631 | 0.2190235799 | 0.218442759 | 0.2174068768 | 0.2223895252 | 0.2285558823 |
| | 0.07739493357 | 0.07768427594 | 0.07813568317 | 0.07863686523 | 0.07914477647 | 0.07964019484 | 0.08011357874 | 0.080561551 | 0.08098252367 | 0.08137570871 | 0.08174390423 | 0.08325409544 | 0.08709450714 | 0.08880556756 |
| Standard Rep (times k-2) | 1.385696445 | 1.643764231 | 1.899867182 | 2.154735278 | 2.408795079 | 2.66228793 | 2.915358842 | 3.168098934 | 3.420573561 | 3.672825958 | 3.924890952 | 5.183170997 | 12.70550562 | 25.21800575 |
| | 0.345675456 | 0.3513956535 | 0.3544633906 | 0.3565880066 | 0.358286142 | 0.3597465232 | 0.3610517106 | 0.3622431587 | 0.3633437286 | 0.3643675924 | 0.3653260152 | 0.3693468508 | 0.3814014159 | 0.3887547905 |
| | 0.2140349063 | 0.201678982 | 0.1931210066 | 0.187036907 | 0.1826232661 | 0.17937935 | 0.1769781035 | 0.1752005927 | 0.1738939428 | 0.1729486752 | 0.17228538 | 0.1715226146 | 0.1832385386 | 0.1971764903 |
| | 0.0629435939 | 0.06429450196 | 0.0659687698 | 0.06766043518 | 0.06927782766 | 0.07079034436 | 0.0721901269 | 0.07347733273 | 0.07465752584 | 0.07573748682 | 0.07672530439 | 0.08053494072 | 0.08788777421 | 0.08981091997 |
| | 0.0257839133 | 0.0278933091 | 0.03042886615 | 0.03293615983 | 0.03527622583 | 0.03741438549 | 0.03935304206 | 0.04110846721 | 0.04270044768 | 0.04414838909 | 0.04547086142 | 0.05067136473 | 0.06384494842 | 0.07103627267 |
| X Rep (times (k-2)(k-3)/2) | -0.01777676493 | -0.005825300235 | 0.001661930699 | 0.007101179101 | 0.01137061138 | 0.01488125976 | 0.01785925776 | 0.02043984085 | 0.02271434665 | 0.02474402636 | 0.02657379396 | 0.03366916254 | 0.05186842754 | 0.06213029474 |
| Y Rep (times (k-1)(k-4)/2) | 0.2987003326 | 0.3126812577 | 0.3211107254 | 0.3269506097 | 0.3313557804 | 0.3348731101 | 0.3377957046 | 0.3402929604 | 0.3424740732 | 0.3444090188 | 0.3461468816 | 0.3528721631 | 0.3703959286 | 0.3805052638 |

Figure 5: The spectrum type I spurious minima.

| k | 5 | 6 | 7 | 8 | 9 | 10 | 11 | 12 | 13 | 14 | 15 | 20 | 50 | 100 |
|---|---|---|---|---|---|---|---|---|---|---|---|---|---|---|
| xi_1 | 0.9616068174 | 0.9867036362 | 0.9968227946 | 1.001434253 | 1.003474556 | 1.004354823 | 1.004662096 | 1.004678965 | 1.004549164 | 1.004348139 | 1.0041668 | 1.003003746 | 1.00072305 | 1.000207713 |
| xi_2 | -0.07908255776 | -0.06041345074 | -0.03508016694 | -0.02588362808 | -0.01991470917 | -0.01581192581 | -0.01286627299 | -0.01067773591 | -0.009006055193 | -0.0076997 0374 | -0.006659067921 | -0.003655140678 | -0.0005580411545 | -0.0001363604554 |
| xi_3 | 0.287548952 | 0.2245161452 | 0.1859910548 | 0.1598160433 | 0.1407087506 | 0.1260422609 | 0.1143656351 | 0.1048096181 | 0.09681901944 | 0.09002247393 | 0.08416022593 | 0.06371471024 | 0.02618552295 | 0.01318159833 |
| xi_4 | 0.3760650557 | 0.3080111148 | 0.2616910791 | 0.2280090166 | 0.202525064 | 0.1823677558 | 0.1660288578 | 0.1524928777 | 0.1410788904 | 0.1313131566 | 0.1228549211 | 0.0931874083 | 0.0385657623 | 0.01960293703 |
| xi_5 | -0.5597784395 | -0.601512325 | -0.6327039183 | -0.6582562628 | -0.6800502369 | -0.6990368175 | -0.7157963022 | -0.7307373008 | -0.7441491733 | -0.7562658311 | -0.7672705594 | -0.8100205756 | -0.9092014274 | -0.9512631475 |
| | | | | | | | | | | | | | | |
| Trivial Rep (times 1) | 1.74888179 | 2.02373391 | 2.295489804 | 2.564593456 | 2.831433737 | 3.096356053 | 3.359655841 | 3.621578783 | 3.882325475 | 4.142962393 | 4.400926516 | 5.685563785 | 13.26772252 | 25.80391663 |
| | 1.352933708 | 1.617871289 | 1.880596512 | 2.141277694 | 2.400283634 | 2.657945764 | 2.914530035 | 3.170242179 | 3.425240691 | 3.6796493 | 3.933565354 | 5.198019132 | 12.7273582 | 25.23830254 |
| | 0.7281653401 | 0.8694840157 | 1.0131957 | 1.15874289 | 1.305788955 | 1.454087619 | 1.603448578 | 1.753719625 | 1.904775009 | 2.056513357 | 2.208850154 | 2.977355079 | 7.683041551 | 15.6084993 |
| | 0.2234922735 | 0.2211724347 | 0.2194946734 | 0.2181558785 | 0.2170622739 | 0.2161654018 | 0.2154323696 | 0.2148365746 | 0.2143550761 | 0.2139703923 | 0.2136667681 | 0.2129914195 | 0.2162712472 | 0.2217633258 |
| | 0.05484339971 | 0.06244204294 | 0.06834925311 | 0.07286984944 | 0.07636068021 | 0.07909314639 | 0.08126332124 | 0.08300942882 | 0.08442931397 | 0.08559776211 | 0.08656692283 | 0.08955913818 | 0.0924073933 | 0.09215372556 |
| Standard Rep (times k-2) | 1.408298036 | 1.671970277 | 1.931566257 | 2.188882771 | 2.444731209 | 2.6995429 | 2.953583796 | 3.207030307 | 3.460005327 | 3.712601009 | 3.964884557 | 5.223144208 | 12.73899878 | 25.24451896 |
| | 0.3273211384 | 0.3388550294 | 0.34772121 | 0.3544345581 | 0.3596246645 | 0.3637405037 | 0.3670837346 | 0.369855904 | 0.372193668 | 0.3741961337 | 0.375932867 | 0.3820679692 | 0.3940261326 | 0.3989494024 |
| | 0.2391507758 | 0.2383455741 | 0.2381737662 | 0.238373771 | 0.2387562001 | 0.2392149557 | 0.2396955197 | 0.2401700884 | 0.2406244382 | 0.2410522281 | 0.2414520331 | 0.2430664059 | 0.2468108805 | 0.2483286417 |
| | 0.0723950212 | 0.07729074833 | 0.08040751665 | 0.08253332498 | 0.08405573712 | 0.0851860203 | 0.08605060341 | 0.08672803056 | 0.08726831449 | 0.08770730756 | 0.08806937875 | 0.08918826705 | 0.09054433281 | 0.0907623865 |
| | -0.007701481192 | 0.004700453437 | 0.01405908725 | 0.02134645822 | 0.02718223121 | 0.03196399919 | 0.03595817112 | 0.03934816015 | 0.04226265346 | 0.04479863565 | 0.0470269957 | 0.05509339701 | 0.07140544709 | 0.07816495097 |
| X Rep (times (k-2)(k-3)/2) | 0.03828606755 | 0.04637753963 | 0.05217475817 | 0.05657319725 | 0.06004140154 | 0.06285340339 | 0.06518359482 | 0.06714876741 | 0.06882896274 | 0.07028251886 | 0.0715531969 | 0.07609294355 | 0.08469195664 | 0.08772839606 |
| Y Rep (times (k-1)(k-4)/2) | 0.3560564518 | 0.3638006747 | 0.3694555461 | 0.3738227487 | 0.377314508 | 0.3801749647 | 0.3825637996 | 0.3845902085 | 0.3863299191 | 0.387840271 | 0.3891634941 | 0.3939078748 | 0.4028793573 | 0.406003207 |

Figure 6: The spectrum type II spurious minima.