[Reviews · NeurIPS 2020]

Review 1

Summary and Contributions: The authors develop an analysis of the eigenvalues distribution of the hessian of the Loss based of the symmetry properties of the Loss function.

Strengths: The paper is relevant for the community since it introduces a novel mathematical framework for the analysis of the hessian spectrum which allow a close form solution for the eigenvalues and their degeneracy.

Weaknesses: -The paper is quite technical in some parts and difficult to follow. -For a classification task the gradient during optimization will be mostly driven by few directions whose number roughly correspond to the number of classes C (see e.g. Sagun, Bottou, LeCun 2018). The authors prove that those directions, for a shallow relu net, have a fixed cardinality. Can the author explain how to reconcile these two different findings? *35 are these the all set of symmetries of the Loss? If not so would it be possible to extend to other type of groups/symmetries? *69 Th 2 : it is quite difficult to have an intuitive view of the results. Would it be possible to show a simple but clear example with a toy model? *192 are equivariant properties of the hessian related one to one with invariant properties of the loss? Suppose the dataset is composed by orbits of a group. Then the Loss is invariant w.r.t. to the group. Is the hessian equivariant w.r.t. the group?

Correctness: To my knowledge yes, but as I said the paper is quite technical and implies extensive knowledge of group theory.

Clarity: Yes.

Relation to Prior Work: Yes.

Reproducibility: Yes

Additional Feedback: After reading the authors feedback I still think this is a good paper for the new approach they took in their analysis and therefore worth to be presented to Neurips community. However authors would need to do a better job in simplifying the whole narrative and structure since the mathematical framework is almost certainly new to most of the readers. This can be done in a following long paper with no length restrictions.


Review 2

Summary and Contributions: This paper characterised the eigenspectrum of the hessian at minima of student teacher one hidden layer networks with fixed output layers and gaussian inputs

Strengths: The characterisation is precise and seems to leverage powerful tools from group theory. Furthermore, the paper shows a good overview of related work and is well written.

Weaknesses: The paper assumes gaussian inputs, a fixed output layer and planted targets. Furthermore, only depth one is considered. This setting might be standard in related work but I still wonder about the generality of these assumptions. For example, the network in Eq. (1.1) cannot predict negative numbers, due to the fixed output layer (w/o biases) which at first glance seems like a reduction of expressivity to me. The spectrum is only characterised at minima and not for example at random initialisation. Moreover, potential extensions for deeper networks are not mentioned. Empirical simulations to test this would have been nice. Finally, implications of the results for training neural networks with gradient based methods or for generalisation are rarely discussed.

Correctness: The result seems plausible but *I did not check the proofs* !!

Clarity: yes, the paper is well written and nicely structured. The title is adequate and the abstract is precise.

Relation to Prior Work: The authors show a solid overview over the related work. A comparison to the marchenko pastor type distribution of eigenvalues of deep nets at random initialisation would have been interesting.

Reproducibility: Yes

Additional Feedback: I am generally unsure about the relevance of this work due to the restrictions mentioned above as well as the unclear implications of this result. Particularly, I do agree that this is a counter-example to the flat minima hypothesis as claimed in l133 but it is a very specific one and does by no means out rule this hypothesis for general deep learning settings, where It is usually debated in. On the other hand, I do see that a lot of heavy mathematical lifting has been done for the derivation of the results. As I am no expert in group theory, I cannot judge the contributions made here (novelty and difficulty of the proof technique, etc.). As a result, I am not giving a strong signal. @AC: Please consider my review an educated guess (see confidence score). @Authors: If you can add some convincing lines on the importance of your results for training and generalisation of networks with gradient descent, I am willing to raise my score. ------ Update after rebuttal: I thank the authors for their answers to my comments. Unfortunately, my concerns remain in place so I am not raising my score but as most reviewers are on the accept side (and as I am as mentioned before no expert in symmetry breaking) I am not going to fight for my stance.


Review 3

Summary and Contributions: This paper aims to analytically characterize the spectrum of the hessian (at local minima) of the expected risk of a two-layer ReLU network for a student-teacher setting, and assuming that the data distribution is the standard Gaussian distribution, and all the output weights are fixed to one. When the input dimension is greater than the number of neurons (k), they show that at least an order of k^2 - O(k) eigenvalues concentrate around zero, and the remaining eigenvalues are away from zero. The fact that the paper considers a simple setting with Gaussian inputs and a shallow network allows them to obtain a closed formula for the hessian (independent of inputs), which is a key step in obtaining the entire spectrum.

Strengths: Understanding the spectrum of the hessian is a quite desirable yet challenging task in deep learning as it is related to understanding the generalization performance of bad vs good local minima, and also the local geometry of the loss function around these minima. The present paper addresses this question by taking a simple and clean setting, and then deriving the (almost) exact eigenvalues of the hessian together with their multiplicities. Moreover, as the authors claimed in the paper, they use techniques from symmetric breaking and representation theory, which appear to be new to the field.

Weaknesses: The technical assumptions are a bit strong, e.g. Gaussian inputs, shallow nets with one trainable layer, and the target network needs to have orthogonal hidden layer's weight matrix. It seems to me that computation of the hessian crucially exploits the fact that the distribution of inputs is Gaussian. This may raise the question of whether the current techniques can be extended to other data distributions, or even the empirical loss which is closer to practice.

Correctness: This reviewer is not entirely familiar with the techniques used in the paper, and hence cannot fully tell the correctness of the main results, especially given that the proofs are nearly 20 pages long in the appendix. However, it looks like the obtained results are consistent with some prior works. One technical thing that I'm not quite sure in the paper is how they deal with the non-differentiability of ReLU, and the computation of the hessian in section C in the appendix (see questions below).

Clarity: In general the paper is clearly written. However, it's not entirely self-contained. For instance, the definitions of type-A, I, II local minima are not provided in the main paper. Also, it is not clear what's the definition of the hessian being used in the paper as the objective function in this case (defined in eq 1.1) is non-differentiable.

Relation to Prior Work: The discussion on related work is fine.

Reproducibility: Yes

Additional Feedback: 1. Can you please clarify what's the definition of the hessian used in your paper, and how do you deal with the non-differentiability of ReLU? 2. My question about the computation of the hessian in section C is as follows. For simplicity, suppose that the network has just one neuron, and the population loss is given by: F(w) = 0.5 E_x [\phi(<w,x>) - \phi(<v,x>)]^2, where v is the target weight vector. Suppose that w is given, and we want to compute the hessian of F at w. Then by chain rule the gradient of F is: Grad F(w) = E_x { [\phi(<w,x>) - \phi(<v,x>)] * \phi'(<w,x>) * x }. Similarly by chain rule, one obtains the hessian: Hess F(w) = E_x { \phi'(<w,x>) * \phi'(<w,x>) * xx' + [\phi(<w,x>) - \phi(<v,x>)] * \phi"(<w,x>) * xx' }. Since w is fixed, it holds that for almost all x, one has <w,x> is non-zero, and hence all the above derivatives are evaluated at differentiable points, and thus \phi"(<w,x>)=0 for a.a. x, meaning that the second term vanishes under the integral. Thus, one gets: Hess F(w) = E_x { \phi'(<w,x>) * \phi'(<w,x>) * xx' }. What puzzled me here is that this hessian is independent of the target v, whereas your formula given at lines 656-657 depends on v. Can you please clarify where's the mistake here? Overall, I think the paper contains some nice results on the hessian of the population loss for a simple network. I am not absolutely certain that the techniques used here can be extended to more practical settings, e.g. for the empirical loss and other data distributions. But I am happy to hear the feedback from the authors, and will adapt my score later. =========================== I have read the author's response. My main concern about the correctness of the claims/results remain, but I am willing to keep my current score as is assuming everything is correct.


Review 4

Summary and Contributions: This paper characterizes the Hessian spectrum of spurious minima in the two-layer ReLU network model, by utilizing group theory and symmetry properties in the model. The results give a full characterization of the Hessian spectrum for local minima. Numerical experiments are also provided to justify their results.

Strengths: Understanding the local geometric structure of neural network is an interesting topic, which could shed light on the understanding of optimization landscape for neural networks. The paper theoretically gives a detailed characterization of Hessian for local minima, which theoretically justifies previous empirical works in this area.

Weaknesses: While the paper gives a characterization of the Hessian spectrum for two-layer ReLU network, it seems a bit unclear whether the technique could generalize to the analysis for over-parameterized network or other data distribution, which might be closer to setting in practice.

Correctness: The claims and methods seem to be reasonable.

Clarity: The paper is well-written and provide several examples to help the reader to understand.

Relation to Prior Work: The authors discussed related works, and explained several differences between prior works and current paper.

Reproducibility: Yes

Additional Feedback: I was wondering whether the weights in top layer have to be positive (all 1), in order to apply current technique. Or there have some technical challenges to overcome. While the characterization for spectrum of Hessian H(w) is provided in this paper, I wonder if the current method could tell the structure of w itself. This might be able to give more intuition about why SGD favors some type of the local minima instead of others in practice. ============================================================== After rebuttal: After reading the rebuttal and other reviews, my concerns are addressed and I tend to keep my score. As also mentioned by other reviewers, the the proof in paper is very technical and involved, which I believe is new to most of people in the community. So, it would be nice if the paper could be further simplified so that it could be easier to understand for more people.

[Author Response · NeurIPS 2020]

We thank all the reviewers for their time and the positive feedback acknowledging the power of the methods to analytically characterize aspects of the Hessian which previously were only accessible numerically. The ReLU model we analyse has been the object of study of many works in recent years. We felt that pursuing generalizations in this work would have made the paper hard to read, especially in view of our introduction of novel mathematical techniques, largely unfamiliar in the field of machine learning. We put a lot of thought into how to present the new methods and appreciate the generally positive response of the reviewers to our efforts.

Other choices of distributions, activation functions and architectures are certainly of interest. Indeed, the methods described in the paper apply more broadly [23] and yield different spectral properties for the Hessian. Adding bias (R2) to the activation or addressing a non-fixed second-layer (R2,R4) increases the technical complexity but is quite tractable. The extension of the methods and results to multi-layers and over-parametrization is very much a topic of our current research. The methods and results we present follow in the tradition of mathematics and physics in that we start with a symmetric model, for which we can prove detailed analytic results, and subsequently break symmetry to get insight into the general theory (the results we obtain are robust under symmetry breaking perturbation, see [22]). We agree with the reviewers that a discussion regarding applicability to more general settings would improve the manuscript and we will revise accordingly.

**Reviewer #1** *Classification w.r.t number of classes.* Although the number of distinct dominating eigenvalues is of fixed cardinality, their total number (counting multiplicity) grows linearly with $k$ (in Sagun, Bottou and LeCun 2016, the data generating model is based on two Gaussians of varying degree of separability (Figure 9), which could reflect on the effective number of classes in our setting). **35**. The assumption that the distribution is orthogonally invariant, together with permutation symmetries, determine the invariants of the loss and objective function. Choosing a different activation, underlying distribution or architecture may lead to different groups of invariants. **69**. A fair request but difficult to satisfy completely as Theorem 2 involves precise asymptotic estimates on eigenvalues, in terms of $1/\sqrt{k}$, in a setting of symmetric maps and non-trivial representation theory. In the revision we will highlight the spectral computation for $\mathbf{W} = \mathbf{V}$ (Section 3.3), illustrating the role played by representation theory, and also include a new *toy (polynomial) model* in the appendix that illuminates other aspects of Theorem 2. **192**. Symmetry properties of the hessian are inherited from the symmetry (isotropy) of the associated critical point. The hessian spectrum is constant along the group orbit of the critical point.

**Reviewer #2** We directly address the generalization error and obtain an analytic description of spectral properties of the respective spurious minima. *Hessian at random weights.* As our intention was to study the spectrum at spurious minima, we view this as a virtue of the analysis. *Generalization and Training loss.* Following a recent series of works [15,17,18,19,21,22] and Hardt *et al.*, 2016; Hardt & Ma, 2016, we focus on the generalization error. Various properties of the training loss can be deduced by concentration of measure arguments, e.g., uniform convergence bounds for gradients (or higher-order derivatives) using generalized vector-valued Rademacher complexity (e.g., Foster *et al.* 2018, Mei *et al.* 2017).

**Reviewer #3** **1**. The hessian of the objective function at a critical point is a $k^2 \times k^2$ symmetric matrix. The objective function is real analytic outside of a thin (measure zero) subset of parameter space [24,22]. Real analyticity is crucial for the analysis and for the power series representation in $1/\sqrt{k}$ for the critical points [24]. **2**. For the single neuron case the reviewer considers, $w, v \in \mathbb{R}$ are non-zero scalars and the angle between $w, v$ is either 0 or $\pi$. A direct computation shows that the loss $\mathcal{L}(w,v) = (w-v)^2/4$ (resp. $(w^2 + v^2)/4$) if $wv > 0$ (resp. $wv < 0$) and so the *only* critical point is at $w = v$ where the Hessian is $1/2$—the angle zero case. The dependence is missed by differentiating under the integral sign. For $d = k > 1$, the explicit dependence of the hessian on the angle between parameters is given in the article (see also [15,22,24]). Some of the main implications for gradient-based methods based on our analysis: 1. Our spectral analysis *rigorously* establishes the spectrum of the Hessian at global and spurious minima is generically extremely skewed (indeed, highly ill-conditioned as $\kappa(\nabla^2 \mathcal{F}) = \Omega(\#\text{Neurons})$). This phenomenon challenges classical approaches to gradient-based methods w.r.t. the associated non-convex landscape [27,28] (reference to Chaudhari *et al.* 2017 and Lee *et al.* 2016 to be added). 2. Stability arguments imply that along the optimization process one should expect the formation of clusters of eigenvalues which drives most of the dynamics, as is indeed the case (note the follow-up discussion for Thm. 1, Section 2, and empirical corroboration of this phenomenon in Section F.1). 3. In terms of dynamical accessibility, our analysis shows that some minima are more likely to be detected by SGD than others. See the discussions under 'Flat minima conjecture and implicit bias'.

**Reviewer #4** The issue of which local minima appear is interesting—minima of types I and A did not appear in [22]. The answer appears to involve a mix of initialization (e.g. Xavier) and the detailed critical point structure of the objective function; this is a topic of current work. *Non-fixed second layer.* Please see the comments in paragraph 2.

[Meta-Review · NeurIPS 2020]

The paper considers the simple problem of a squared-loss for a two-layer ReLU network. By relying on certain invariance properties of the loss, the authors derive an analytical expression of the Hessian and its spectrum. The analysis, in the context of the specific problem at hand, is rather novel and although it only yields results on a small shallow network, it has some potential to yield some new research directions. Although I recommend acceptance, I should say that the paper is seen by the reviewers as quite technical and I think the paper would much benefit from a revision where more intuition (maybe a proof sketch) is given in the main paper.